# Quantifying the real-world impact of antibiotic use and genetic determinants of resistance on gonococcal dynamics

David Helekal[1], Tatum D. Mortimer [2], Aditi Mukherjee [1], Gabriella Gentile [3], Adriana Le Van[4,5], Sofia Blomqvist [1], Robert A. Nicholas[3,6], Ann E. Jerse[4], Samantha G. Palace [1] & Yonatan H. Grad [1] ✉

The dynamics of antimicrobial resistance in bacterial populations are influenced by the fitness impact of genetic determinants of resistance and antibiotic pressure. However, estimates of real-world fitness impact have been lacking. To address this gap, we developed a hierarchical Bayesian phylodynamic model to quantify contributions of resistance determinants to strain success in a 20-year collection of *Neisseria gonorrhoeae* isolates. Fitness contributions varied with antibiotic use, which over this period included ciprofloxacin, cefixime, ceftriaxone and azithromycin, and genetic pathways to phenotypically identical resistance conferred distinct fitness effects. These findings were supported by competition experiments both in vitro and in the mouse model of gonococcal infection. Quantifying these fitness contributions to lineage dynamics reveals opportunities for investigation into other genetic and environmental drivers of fitness. This work thus establishes a method for linking pathogen genomics and antibiotic use to define factors shaping ecological trends.

The prevalence of antimicrobial resistance (AMR) reflects competition in an ever-changing environment[1,2]. When an antibiotic is introduced into clinical use, the environment in which bacteria compete changes. Bacterial lineages that carry or acquire resistance to the newly introduced antibiotic are now better adapted to the new environment. This leads to an increase in the prevalence of resistance and therefore a fitness advantage associated with resistance. Once resistance to the antibiotic becomes sufficiently widespread, its use is often reduced in favour of another antibiotic for which resistance prevalence is low. This alters the fitness landscape, such that bacteria carrying alleles and genes that had previously conferred a fitness advantage in the context of the first antibiotic may now be less or equally fit compared to bacteria that do not carry such genes or alleles. As such, alleles and genes that had conferred increased fitness may become neutral or deleterious.

The complexity of the bacterial AMR fitness landscape is shaped by multiple factors. These include antibiotic pressure from drugs in current and past use as well as antibiotic resistance, which can often be achieved through multiple, at times interacting, genetic pathways (cf., macrolide resistance in *N. gonorrhoeae*[3,4] and fluoroquinolone resistance in *E. coli*[1]). Population structure[5,6], linkage between AMR determinants in a changing environment[1,6], linkage with sites under balancing selection[6,7], mutation–selection balance[8] and non-antibiotic pressures can also inform this landscape.

While the genetic determinants of AMR play a large role in the population expansion and contraction of drug resistant lineages[1], the fitness impact of these determinants can vary across genetic backgrounds and environments[9,10]. Efforts to quantify the fitness impact of individual genetic features across shifting patterns of antibiotic

[1]Department of Immunology and Infectious Diseases, Harvard T. H. Chan School of Public Health, Boston, MA, USA. [2]Department of Population Health, College of Veterinary Medicine, University of Georgia, Athens, GA, USA. [3]Department of Pharmacology, University of North Carolina at Chapel Hill, Chapel Hill, NC, USA. [4]Department of Microbiology and Immunology, Uniformed Services University of the Health Sciences, Bethesda, MD, USA. [5]Henry M. Jackson Foundation for the Advancement of Military Medicine Inc, Bethesda, MD, USA. [6]Departments of Microbiology and Immunology, University of North Carolina at Chapel Hill, Chapel Hill, NC, USA. ✉e-mail: ygrad@hsph.harvard.edu

use in real-world data have been fraught with many challenges, such as limited availability of data on antimicrobial use, both for targeted treatment and accounting for bystander exposure[11]; the influence of other factors, such as pressure from host immunity, on overall pathogen fitness; the frequent co-occurrence of multiple antibiotic resistance determinants in drug resistant strains; and host factors, such as population structures, and host behaviour, including differences in access to diagnostics and treatment[5,6]. In addition, for many pathogens, we lack longitudinal datasets of sufficient size, duration and systematic collection to enable inference about fitness.

Here we overcame these challenges to define the fitness contributions of AMR determinants in response to changes in treatment using data from *N. gonorrhoeae* in the USA. *N. gonorrhoeae* is an obligate human pathogen that causes the sexually transmitted infection gonorrhoea; infection does not elicit a protective immune response[12,13]. A collection of over 5,000 specimens from 20 years (2000–2019) of the CDC's Gonococcal Isolate Surveillance Project (GISP), the CDC's sentinel surveillance programme for antibiotic resistant gonorrhoea, has been sequenced and has undergone resistance phenotyping, with metadata including the demographics of the infected individuals[3,14–17]. Data on primary treatment in the USA over this period and going back to 1988 (Extended Data Fig. 1)[18] have been reported by the CDC and reflect changes in first-line therapy, from fluoroquinolones to cephalosporins plus macrolides, and among the cephalosporins, from the oral cefixime to intramuscular ceftriaxone[19]. To combine sequencing data with resistance covariates, we used phylodynamic modelling. Building on previous efforts[20–24], we deployed a framework that can accommodate multiple lineages, multiple pathways and multiple time-varying covariates to quantitatively estimate the resistance determinant-specific fitness costs in circulating *N. gonorrhoeae* and their interaction with antibiotic pressures. We analysed the period from 1993, the year fluoroquinolones were recommended as first-line therapy for *N. gonorrhoeae* in the USA to 2019, the year in which the most recent sample in this dataset was collected (Extended Data Fig. 1).

## Results

### Defining *N. gonorrhoeae* drug resistant lineages

We first sought to define AMR-linked lineages from the GISP specimens. On epidemic timescales[25], *N. gonorrhoeae* maintains a lineage structure largely shaped by antimicrobials and sexual networks[26]. The treatment for *N. gonorrhoeae* infections in the USA over the past 20 years has been defined by three main drug classes: fluoroquinolones, third-generation cephalosporins and macrolides (Extended Data Fig. 1). We used ancestral state reconstruction for the major AMR determinants for these antibiotics to identify clusters of specimens that had not changed state since descending from their most recent common ancestors (MRCA) (Fig. 1a and Table 1). We refined the classification by requiring that the MRCA was no earlier than 1980. As the three drug classes under study entered use after this date, this cut-off limits the analysis to a time frame over which ancestral state reconstruction is likely to remain accurate, while helping separate lineages that acquired the same resistance pattern independently. We focused on lineages that have at least 30 specimens, reasoning that a minimum cut-off helps avoid the inclusion of small outbreak clusters that could potentially produce unreliable estimates ('Lineage assignment and phylogenetic reconstruction' in Methods).

This definition led to the identification of 29 lineages across the dataset (Fig. 1a), along with the corresponding distribution of determinants (Table 1). Most lineages (21/29) have at least one AMR determinant, with multiple pathways to resistance for a given antibiotic present across lineages. Lineages 22 and 23, for example, carry the mosaic *penA* 34 allele, and the remaining 27 lineages all carry the Penicillin Binding Protein 2 (PBP2; encoded by *penA*) substitution A516G, each of which increases resistance to cephalosporins[27,28]. While we did not use phenotypic resistance data in the lineage assignment or

subsequent analysis, we noted that the resulting lineage assignment (Fig. 1a) was congruent with the distribution of phenotypic resistance to ciprofloxacin, cefixime and azithromycin (Fig. 1b). This provided us with a form of validation for the lineage assignment approach.

The related lineages 20–22 share many resistance determinants but have differing estimates of their effective population sizes through time, $Ne(t)$, and illustrate the dynamics that emerge when juxtaposing $Ne(t)$ with antibiotic use and resistance (Fig. 2). This cluster of lineages contains a previously described and the largest mosaic *penA* 34-carrying lineage, lineage 22 (ref. 3), along with its two sister lineages. Lineage 20, the oldest lineage in this cluster, grew during the fluoroquinolone era and decreased afterwards[23]. In this lineage, nearly all descendants sampled after the recommendation of ceftriaxone plus azithromycin dual therapy had acquired a new resistance determinant or lost an existing one. In one sublineage, this change included replacement of a resistance-conferring *gyrA* allele (encoding 91F, 95G) with the wild-type allele (91S, 95D), resulting in phenotypic susceptibility (Fig. 2, blue bar). Another sublineage changed *mtrR* promoter alleles (Fig. 2, yellow bar). Determinants at the *mtrR* promoter are associated with resistance to a wide range of antibiotics[29] including macrolides[30]. Yet another sublineage acquired azithromycin resistance through C2611T substitution in 23S rRNA (Fig. 2, red bar). Furthermore, the descendants of lineage 20 appeared to switch sexual networks: most recent isolates were from heterosexuals, whereas past isolates were from men who have sex with men (Supplementary Fig. 1). Lineage 21 expanded around the 2010 switch in recommended treatment to dual therapy, with azithromycin plus ceftriaxone median lineage MRCA time of 2011.46 and 95% credible interval of [2009.66, 2012.66]; see Supplementary Fig. 2 for credible intervals indicating timing uncertainty for lineage ancestor nodes. The effective population size for the mosaic *penA* 34-carrying lineage 22 grew during the fluoroquinolone period and after, but decreased with the introduction of azithromycin and ceftriaxone dual therapy.

Together, these patterns of lineage expansion and contraction indicated a relationship among the antibiotics recommended for treatment, genetic determinants of resistance and lineage success. However, while for lineages 20 and 22 the pattern of expansion and contraction matches our expectations based on their resistance profile, it was not clear from a simple inspection what could explain the dynamics of lineage 21.

### Hierarchical Bayesian phylodynamic modelling reveals a changing fitness landscape

We next sought to quantify the fitness contributions of the genetic determinants of resistance, how these varied over time and whether this was consistent across the phylogeny. For each determinant, we estimated a set of regression coefficients, one for each of the antibiotic classes to which it conferred resistance, along with an intercept term. These modelled the effect of a given resistance determinant on the effective population size growth rate of lineages that carry that determinant as a function of the reported treatments (Supplementary Table 1). To account for the impact of lineage background and overdispersion of lineage growth rates due to effects not included in the model, we also included lineage-specific residual terms, lineage-specific background terms and a global mean growth-rate term in a hierarchical manner. We noted that as the treatment data are percentages summing to 1 and thus are not full rank, we selected ceftriaxone 250 mg as the baseline for all estimated treatment use effects.

This model formulation allowed us to answer three main questions. First, did the relative fitness of a given resistance determinant change as a function of the pattern of antibiotic use? Second, what is the fitness cost or benefit through time associated with a particular resistance allele compared to its susceptible counterpart? Third, how much of a lineage's trajectory is explained by the fitness contributions of the resistance determinants? To answer these questions, we first

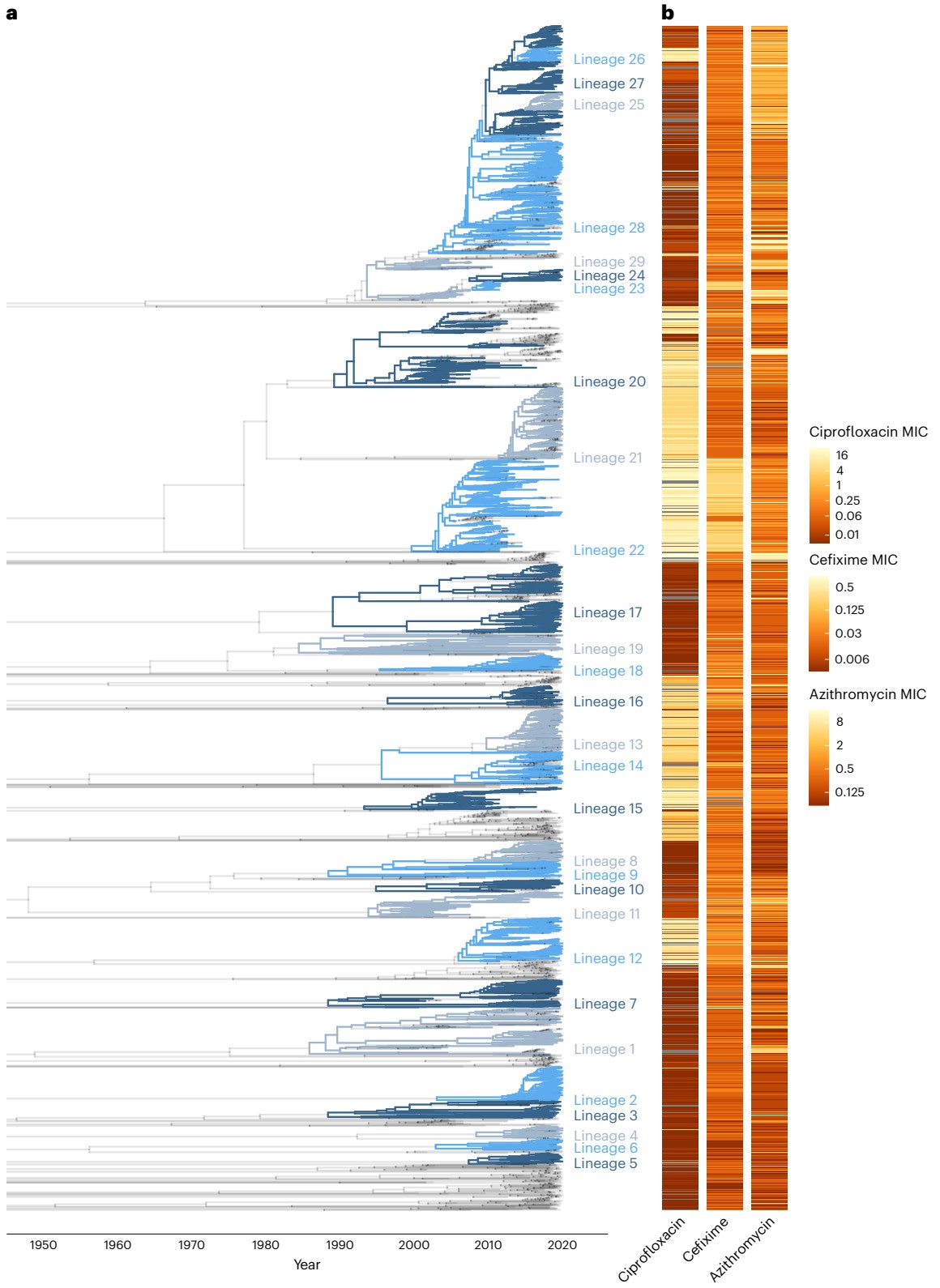

**Fig. 1 | Lineage assignment based on AMR determinants. a**, The phylogenetic tree annotated according to the lineage assignment of each node. Grey nodes in the phylogenetic tree denote lines of descent that are not in an assigned lineage. Lineage numbering was determined by post-order traversal of the tree. Colours are for legibility purposes only. **b**, The distribution of phenotypic resistance for ciprofloxacin, cefixime and azithromycin. Grey represents missing values. MICs are reported in units of µg ml⁻¹. MICs are truncated for ciprofloxacin above 32 µg ml⁻¹ and below 0.005 µg ml⁻¹, for cefixime above 1 µg ml⁻¹ and below 0.003 µg ml⁻¹, and for azithromycin MICs above 16 µg ml⁻¹ and below 0.06 µg ml⁻¹.

**Table 1 | Distribution of AMR determinants across lineages**

| Lineage | gyrA | | parC | | | ponA | penA | | | mtr | | | | 23S rRNA |
|---|---|---|---|---|---|---|---|---|---|---|---|---|---|---|
| | S91 | D95 | D86 | D87 | E91 | L421 | A501 | G542 | Type | mtrC | mtrD | mtr promoter | mtrR | C2611 |
| 1 | S | D | D | S | E | L | A | G | non-m | non-m | non-m | non-m | **LOF** | C |
| 2 | S | D | D | S | E | L | A | G | non-m | non-m | non-m | non-m | non-m | C |
| 3 | S | D | D | S | E | L | A | G | non-m | non-m | non-m | non-m | non-m | C |
| 4 | S | D | D | S | E | L | A | G | non-m | non-m | non-m | non-m | non-m | C |
| 5 | S | D | D | S | E | L | A | G | non-m | non-m | non-m | non-m | non-m | C |
| 6 | S | D | D | S | E | L | A | G | non-m | non-m | non-m | non-m | non-m | C |
| 7 | S | D | D | S | E | **P** | A | G | non-m | non-m | non-m | **A-del** | non-m | C |
| 8 | S | D | D | S | E | L | A | G | non-m | non-m | non-m | non-m | non-m | C |
| 9 | S | D | D | S | E | **P** | A | G | non-m | non-m | non-m | non-m | non-m | C |
| 10 | S | D | D | S | E | **P** | A | G | non-m | non-m | non-m | **A-del** | non-m | C |
| 11 | S | D | D | S | E | **P** | A | G | non-m | non-m | non-m | **A-del** | non-m | C |
| 12 | **F** | **G** | D | S | **G** | **P** | **T** | G | non-m | non-m | non-m | **A-del** | non-m | C |
| 13 | **F** | **A** | D | **R** | E | L | A | G | non-m | non-m | non-m | non-m | **LOF** | C |
| 14 | **F** | **A** | **N** | S | E | L | A | G | non-m | non-m | non-m | non-m | **LOF** | C |
| 15 | **F** | **G** | D | **R** | E | **P** | A | G | non-m | non-m | non-m | **A-del** | non-m | C |
| 16 | **F** | **G** | **N** | S | E | **P** | **V** | G | non-m | non-m | non-m | **A-del** | non-m | C |
| 17 | S | D | D | S | E | **P** | A | G | non-m | non-m | non-m | **A-del** | non-m | C |
| 18 | S | D | D | S | E | L | A | G | non-m | non-m | non-m | non-m | non-m | C |
| 19 | S | D | D | S | E | **P** | A | G | non-m | non-m | non-m | non-m | non-m | C |
| 20 | **F** | **G** | D | **R** | E | **P** | A | **S** | non-m | non-m | non-m | **A-del** | non-m | C |
| 21 | **F** | **A** | D | **R** | E | **P** | A | **S** | non-m | non-m | non-m | non-m | non-m | C |
| 22 | **F** | **G** | D | **R** | E | **P** | - | - | **mosaic** | non-m | non-m | **A-del** | non-m | C |
| 23 | S | D | D | S | E | L | - | - | **mosaic** | non-m | non-m | non-m | non-m | C |
| 24 | S | D | D | S | E | L | A | G | non-m | non-m | non-m | non-m | non-m | C |
| 25 | S | D | D | S | E | L | A | G | non-m | non-m | **mosaic** | **mosaic** | non-m | C |
| 26 | **F** | **A** | **N** | S | **E** | L | A | G | non-m | **mosaic** | **mosaic** | **mosaic** | non-m | C |
| 27 | S | D | D | S | E | L | A | G | non-m | **mosaic** | **mosaic** | **mosaic** | non-m | C |
| 28 | S | D | D | S | E | L | A | G | non-m | non-m | **mosaic** | non-m | non-m | C |
| 29 | S | D | D | S | E | L | A | G | non-m | non-m | non-m | non-m | non-m | **T** |

Bolded text indicates genes and loci associated with resistance. For *gyrA*, *parC* and *ponA*, polymorphisms occur at several key amino acid positions. The *penA* and *mtr* loci exhibit complex patterns of polymorphism that include interspecies mosaicism as well as individual amino acid variations. 'non-m' denotes non-mosaic alleles. 'A-del' denotes A-deletion in the *mtrR* promoter[29]. 'LOF' denotes loss-of-function, such as due to premature stop codons. The only determinant in 23S rRNA that appears frequently in our dataset is the C2611T substitution, where T indicates at least one and up to four copies of 23S rRNA C2611T. Genes and loci associated with fluoroquinolone resistance: *gyrA*, *parC*, *mtr*. Genes and loci associated with cephalosporin resistance: *ponA*, *penA*, *mtr*. Genes and loci associated with macrolide resistance: *mtr*, 23S rRNA.

calculated the predicted effect of individual determinants on the growth rate of lineages, finding that several resistance determinants had a strong impact (defined by the 95% posterior credible interval excluding [−0.1, 0.1]) on lineage dynamics (Supplementary Table 2).

**Fluoroquinolones and the fitness impact of *gyrA* determinants of resistance**

*gyrA* is the main fluoroquinolone-resistance determining gene in *N. gonorrhoeae*, with alleles of *parC* also contributing to resistance[30]. Our modelling revealed several phenomena among lineages encoding ParC 86D/87R/91E. First, lineages carrying GyrA 91F/95G with this *parC* allele experienced a growth rate increase during the period of recommended fluoroquinolone treatment for gonorrhoea (Fig. 3a and Supplementary Fig. 3). However, for GyrA 91F/95G, there was too much uncertainty to determine its absolute effect on the growth rate of lineages carrying it during the fluoroquinolone period compared to the baseline GyrA 91S/95D type within the wild-type ParC 86D/87S/91E context (Fig. 3a). After fluoroquinolones were no longer

recommended, GyrA 91F/95G appeared weakly deleterious when combined with ParC 86D/87R/91E allele compared to wild-type (Fig. 3a and Supplementary Table 2). Second, lineages carrying GyrA 91F/95A had distinctly higher growth rates than the GyrA 91S/95D susceptible allele after the period in which fluoroquinolones were used for treatment (Fig. 3a and Supplementary Table 2). Third, lineages carrying GyrA 91F/95A had a relative growth rate advantage over GyrA 91F/95G after the end of fluoroquinolone era (Extended Data Fig. 2). Most lineages carrying GyrA 91F/95A expanded only after 2007 when fluoroquinolones were no longer recommended, increasing the uncertainty in estimates of the effect that fluoroquinolone use had on these lineages. It is also worth noting that we were unable to make meaningful inferences about the impact of GyrA 91F/95A and GyrA 91F/95G across all *parC* contexts (Fig. 3a; 'Mean'). This may be due either to heterogeneity of fitness effects across *parC* contexts or to the lack of sufficient polyphyly to inform this estimate.

To investigate whether the degree of resistance provided by GyrA 91F/95G differed from that provided by GyrA 91F/95A in lineages

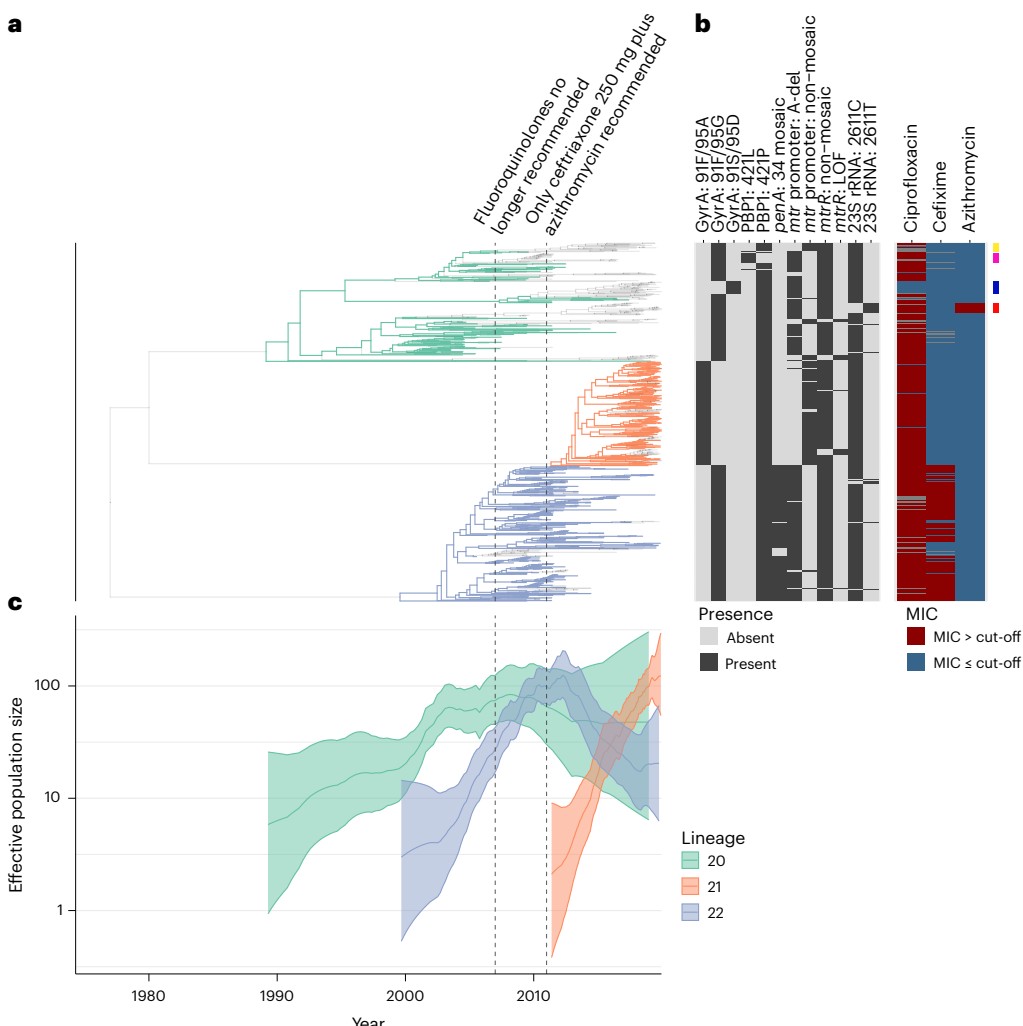

**Fig. 2 | A cluster of phylogenetically related lineages with evidence of adaptation in response to changes in antibiotic use. a**, The phylogeny for lineages 20–22. The grey transparent tips correspond to isolates that have diverged from the ancestral determinant combination of the parental lineage. Ancestral types for lineage 20: GyrA 91F/95G, PBP1 421P, *mtr* promoter A-del, 23S rRNA 2611C. Ancestral types for lineage 21: GyrA 91F/95A, PBP1 421P, *mtr* promoter non-mosaic, 23S rRNA 2611C. Ancestral types for lineage 22: GyrA 91F/95G, PBP1 421P, *penA* 34 mosaic, *mtr* promoter non-mosaic, 23S rRNA 2611C. **b**, The presence and absence of relevant determinants and antibiotic resistance

phenotypes above or below each drug's cut-off (CIP: 1 μg ml⁻¹; CFX: 0.125 μg ml⁻¹; AZI: 4 μg ml⁻¹). Coloured bars on the right: yellow bar highlights a cluster of isolates that changed *mtr* promoter alleles; magenta bar highlights a reversion of resistance-conferring PBP1 421P mutation to susceptible 421L; blue bar highlights a cluster of isolates that reverted to a fluoroquinolone-susceptible *gyrA* allele; red bar highlights a cluster of isolates that acquired azithromycin resistance. **c**, Median effective population size trajectories for each of the lineages with 95% credible intervals as estimated by PHYLODYN[87].

containing the ParC 86D/87R/91E allele, we fitted a linear model to log₂-transformed ciprofloxacin (MIC) while accounting for determinants at the *mtrCDE* operon. There was no significant difference in log₂-transformed ciprofloxacin MICs (GyrA 91F/95G coefficient = 0.183, two-sided *t*-test *P* > 0.242; see Methods for details).

Given these results, we tested whether GyrA 91F/95A contributes to a growth rate advantage over GyrA 91F/95G in vitro. Both sets of mutations increased the ciprofloxacin MIC by ≥128-fold over the susceptible *gyrA* allele (Supplementary Table 3). In an in vitro competition assay between GyrA 91F/95A and GyrA 91F/95G isogenic strains in the GCGS0481 strain background with ParC 86D/87R/91E, GyrA 91F/95A conferred a fitness benefit (Fig. 3b,c): the competitive index (CI) after 8 h of competition for GCGS0481 kanamycin-labelled GyrA 91F/95A versus GyrA 91F/95G was 1.76 (*P* = 0.006), consistent with the reciprocal competition, in which the CI of kanamycin-labelled GyrA 95F/95G versus GyrA 91F/95A was 0.53 (*P* = 0.0009). Both GyrA 91F/95G and GyrA 91F/95A strains were less fit in vitro than the susceptible parental strain (Supplementary Fig. 4). After 8 h of competition, the

CI of kanamycin-labelled GyrA 91F/95A versus GyrA 91S/95D was 0.67 (*P* = 0.0015), and for kanamycin-labelled GyrA 91F/95G versus GyrA 91S/95D, it was 0.52 (*P* = 0.01) (Supplementary Fig. 4). Consistent with this, the CI after 8 h of competition of GCGS0481 kanamycin-labelled GyrA 91S/95D versus GyrA 91F/95A was 1.45 (*P* = 0.0023), and for kanamycin-labelled GyrA 91S/95D versus GyrA 91F/95G, it was 1.72 (*P* = 0.007) (Supplementary Fig. 4).

GyrA 91F/95A within the ParC 86N/87S/91E context conferred a growth rate advantage after 2007, when fluoroquinolones were no longer in use, compared to the baseline type that does not carry any of the resistance determinants studied (Supplementary Table 2 and Supplementary Fig. 5).

## Beta lactam resistance and the fitness impact of resistance-conferring *penA* alleles and mutations

Variants in the *penA* gene, which encodes PBP2, contribute to resistance to cephalosporins as well as other beta lactams, with mosaic *penA* alleles being the major determinants of resistance to cephalosporins[14,31,32].

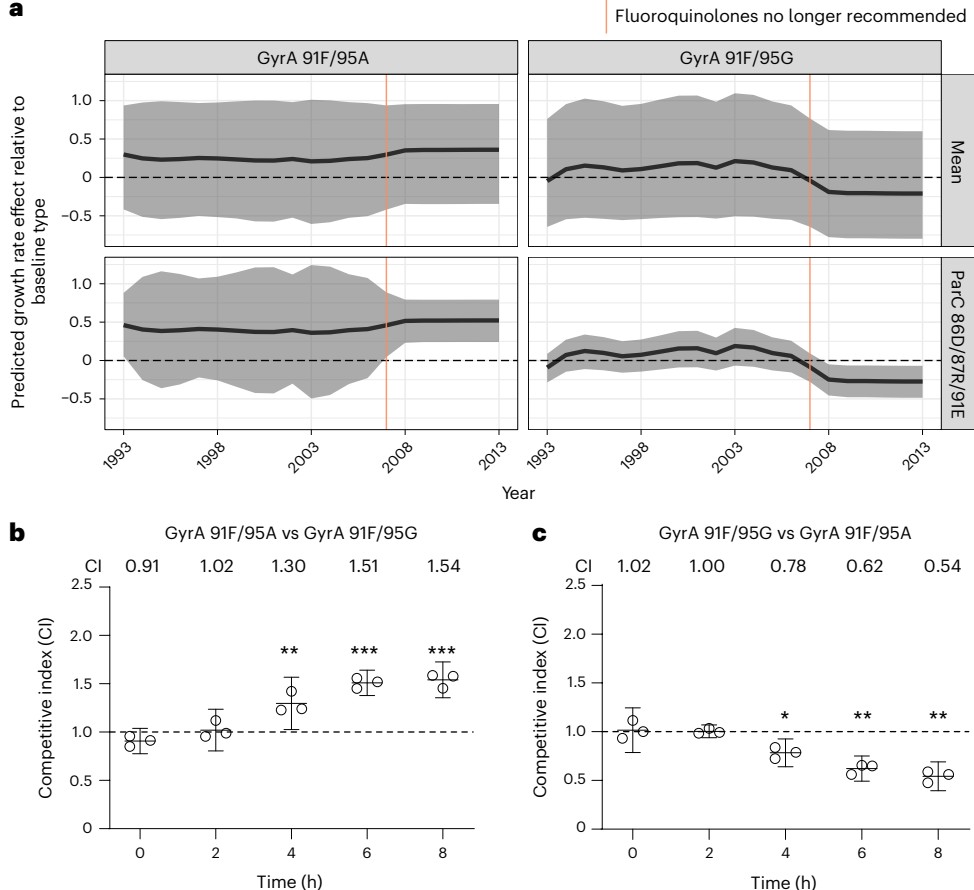

**Fig. 3 | Summary of results for fitness impact of selected GyrA determinants.**
**a**, The predicted absolute effect on lineage growth rate of selected GyrA
determinants across all *parC* contexts (top, 'Mean') or within the ParC
86D/87R/91E context (bottom) based on past fluoroquinolone use patterns.
The predicted effect is an absolute effect as computed compared to the baseline
GyrA 91S/95D type within the wild-type ParC 86D/87S/91E context. The shaded
region denotes the 95% posterior credible interval around the posterior median,
depicted by the bold black line. Dashed line denotes no predicted growth
rate effect relative to baseline allele. **b**,**c**, In vitro competition in liquid culture

between kanamycin-labelled GyrA 91F/95A and unlabelled GCGS0481 GyrA
91F/95G (**b**) and kanamycin-labelled GyrA 91F/95G and unlabelled GCGS0481
GyrA 91F/95A (**c**). The competitive index (CI) is shown for the kanamycin-labelled
strain relative to the unlabelled strain. Statistical significance (at 2, 4, 6 and 8 h:
$P = 0.89, 0.01, 0.002, 0.006$ (**b**); $P = 0.14, 0.005, 0.002, 0.0009$ (**c**)). $N = 3$ per
timepoint, representative of 3 independent experiments. Error bars represent
mean with 95% CI. Statistical significance of CI measurements was assessed using
unpaired two-sided Student's *t*-test (*$P \leq 0.05$, **$P \leq 0.01$ and ***$P \leq 0.005$).

Most of the cephalosporin resistance determinants in our dataset
appeared in 1–3 lineages each (Table 1), which limited our ability to esti-
mate their impact on growth rates (Supplementary Table 2 and Fig. 6).
However, we estimated a major beneficial effect of mosaic *penA* 34 on
growth rates when cefixime and ceftriaxone 125 mg were widely used,
as well as the subsequent loss of this beneficial effect when treatment
with cephalosporins other than ceftriaxone 250 mg declined (Fig. 4a
and Supplementary Table 2). Similarly, carriage of PBP2 501T was associ-
ated with a large relative decrease in fitness when ceftriaxone 250 mg
became the sole recommended treatment (Supplementary Table 2
and Fig. 6). However, the absolute effect for PBP2 501T compared to
wild-type cannot be identified, as PBP2 501T appeared only in a single
lineage that carries a unique GyrA/ParC combination. The decrease
in the predicted growth rate effect for both mosaic *penA* 34 and PBP2
501T started in 2008 and aligns with a shift in primary treatment with
cephalosporins to ceftriaxone 250 mg, even before the guidelines
changed in 2012 (Extended Data Fig. 1).

PBP2 501V was associated with a weak increase in fitness after the
switch in treatment to ceftriaxone 250 mg (Supplementary Table 2 and
Fig. 7). Both the PBP2 501V and 501T have wide credible intervals for
their absolute effects compared to the baseline type that does not carry
any of the resistance determinants studied (Supplementary Table 2

and Fig. 6). These alleles occur in a single lineage each, with both line-
ages carrying unique *gyrA*/*parC* combinations (Table 1), making the
intercept term for these determinants unidentifiable.

### Beta lactams and the fitness impact of *ponA* L421P

The *ponA* gene encodes Penicillin Binding Protein 1 (PBP1). Variants in
*ponA* contribute to resistance to penicillin[33], while also being associated
with an increase in cephalosporin MICs[27].

The PBP1 421P variant was associated with a weak disadvan-
tage compared to the baseline type that does not carry any of the
resistance determinants throughout the study period (Fig. 4a and
Supplementary Table 2). However, there was no discernible change
to its fitness effect as the antibiotic treatment composition changed
(Supplementary Table 2 and Fig. 10), consistent with PBP1 421P being
primarily associated with an increase in penicillin MICs and only a mild
increase in cephalosporin MICs[27]. This is consistent with the observa-
tion of reversions from the resistance-associated PBP1 421P to the
wild-type PBP1 421L.

To test whether the PBP1 421P allele confers a fitness cost in the
absence of antibiotic treatment, we competed the wild-type PBP1 421L
against PBP1 421P in two genetic backgrounds: the penicillin suscep-
tible FA19 strain that normally carries the wild-type PBP1 421L and the

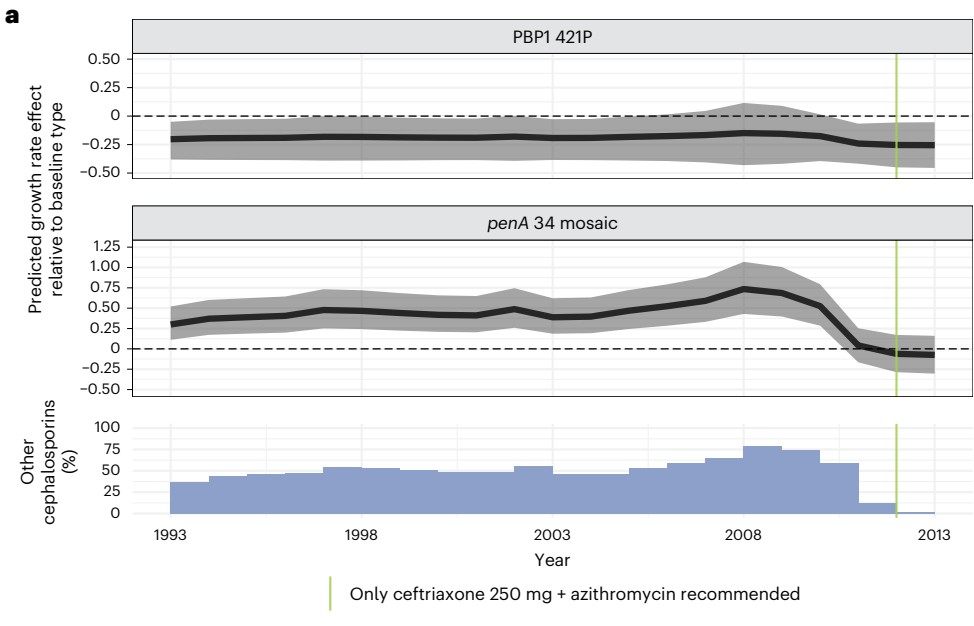

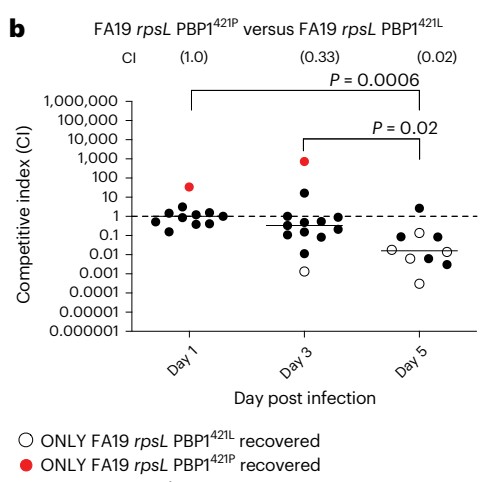

**Fig. 4 | Summary of results for mosaic PBP1 421P and *penA* 34. a**, Top: the predicted absolute growth rate effect for mosaic *penA* 34 compared to baseline. The predicted effect was computed on the basis of reported treatments. The shaded region denotes the 95% credible interval around the posterior median, depicted by the bold black line. Dashed line denotes no predicted growth rate effect relative to the baseline type that does not carry any of the resistance determinants studied. Middle: the predicted absolute growth rate effect for PBP1 421P on lineage growth rate. The predicted effect was computed on the basis of reported treatments. Shaded region denotes the 95% posterior credible interval around the median, depicted by the bold black line. Dashed line denotes no predicted growth rate effect relative to the baseline type that does not carry any of the resistance determinants studied. Bottom: the use of cephalosporins other than ceftriaxone 250 mg as a percentage of primary treatment. Other cephalosporins consist mainly of ceftriaxone 125 mg, cefixime and other unclassified cephalosporins. **b**, In vivo murine competition assay between FA19 *rpsL* PBP1 421P versus FA19 *rpsL* PBP1 421L. Competitive index (CI) was measured at 1, 3 and 5 days post infection. Where only one strain was detected, the CI was imputed at the limit of detection (Methods). $N = 2 \times 7$ per timepoint, equating to 2 independent experiments. Each point represents the CI measurement for an individual mouse. The horizontal lines indicate the geometric mean. Statistical significance of CI measurements was assessed using two-sided Mann–Whitney test.

penicillin-resistant FA6140 strain[34] that normally carries PBP1 421P. The PBP1 421L and PBP1 421P derivatives of each strain had similar fitness during in vitro growth (Supplementary Fig. 9). Because the predicted fitness effect of the PBP1 421P variant is small and may not be detectable on the timeline of the in vitro studies, we also performed competition assays of these strains in the female murine infection model to measure relative fitness in the presence of physiologically relevant stressors and over a longer period. The PBP1 421P allele incurred an in vivo fitness cost to both FA19 and FA6140 when in competition with the isogenic PBP 421L strain: 5 days post infection, the competitive index for the FA19 *rpsL* PBP1 421P versus FA19 *rpsL* PBP1 421L mutant was 0.02 ($P = 0.0006$) (Fig. 4b), and the competitive index for FA6140 *rpsL* PBP1 421L versus FA6140 *rpsL* PBP1 421P was 3.46 ($P = 0.03$) (Supplementary Fig. 8).

## Azithromycin use and the fitness impact of variants in the *mtr* locus

*mtrCDE* encodes an efflux pump that modulates resistance to a wide range of antibiotics in *N. gonorrhoeae*[29], including to macrolides[30], and it is regulated by its transcriptional repressor, MtrR. Of particular relevance are mosaic *mtrC*, *mtrD* and *mtrR* promoter as these are associated with azithromycin resistance[3,35]. While our modelling recovered a growth rate increase associated with the carriage of mosaic *mtrR* promoter and the mosaic *mtrD* compared to wild-type baseline during the azithromycin co-treatment era (Supplementary Table 2 and Fig. 11), the fact that mosaic *mtrR* promoter only occurred on mosaic *mtrD* backgrounds in our dataset (Table 1) raises the concern that the estimated effects of the *mtrR* promoter and the mosaic *mtrD* may be only weakly

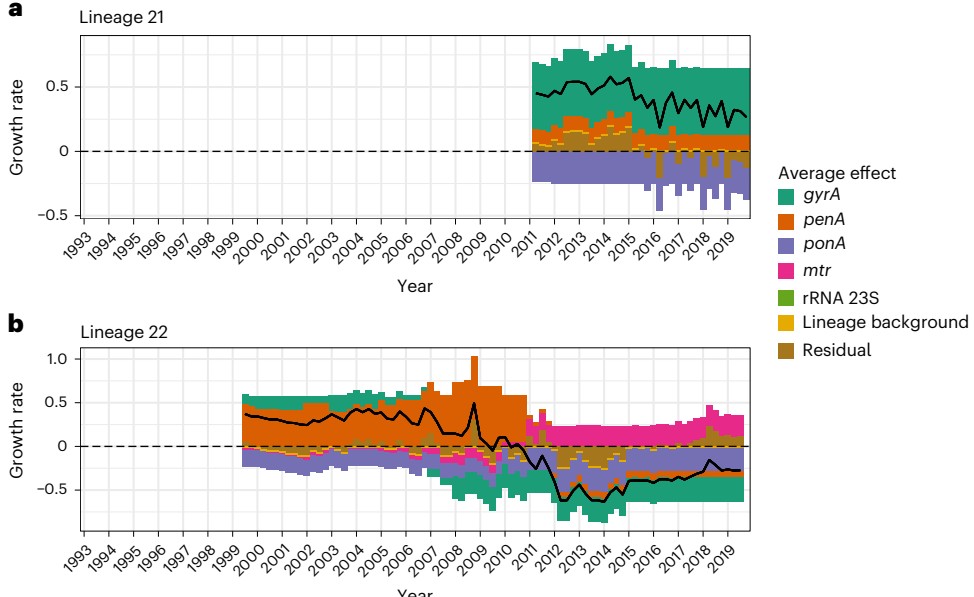

**Fig. 5 | Growth rate effect summary for lineages 21 and 22. a**, Summary of average growth rate effects for lineage 21. **b**, Summary of average growth rate effects for lineage 22. The black solid line represents the total average effect.

The dashed horizontal line indicates zero. Posterior summaries with uncertainty estimates for epochs spanning 1993–2007, 2007–2010, 2010–2012 and 2012–2019 available in Supplementary Table 1.

identified. As such, we focused on the combined effect of mosaic *mtrR* promoter, mosaic *mtrC* and mosaic *mtrD*. The predicted combined effect had large uncertainty, limiting interpretation, with clear support for a growth rate benefit only in 2010–2011 (Extended Data Fig. 3).

The *mtrR* promoter A deletion in the 13-bp inverted repeat—a determinant implicated in an increase in resistance to a wide range of antibiotics including macrolides[29]—was associated with a weak increase in growth rate after the switch to ceftriaxone 250 mg and azithromycin. This increase led to a weak advantage in growth rate after 2012 compared to the baseline type that does not carry any of the resistance determinants studied (Supplementary Table 2 and Figs. 11 and 12).

### How much of the lineage growth trajectory is explained by resistance determinants?

To quantify how much the set of resistance determinants and lineage background explains each lineage's growth rate over time, for each lineage we visualized the average growth rate effect of individual resistance determinants, along with lineage residual effect and lineage background effect, and summarized the total effect in the four treatment recommendation periods in the study period (Fig. 5 and Supplementary Figs. 13–19).

This per-lineage analysis revealed the shifting contributions of resistance determinants to individual lineage growth dynamics, provided examples in which fitness costs of one determinant are counterbalanced by the fitness benefits of another, and identified lineages with dynamics unexplained by these determinants. Lineage 22 carried a combination of GyrA 91F/95G along with mosaic *penA* 34. Despite carrying GyrA 91F/95G, which was associated with a fitness cost after fluoroquinolones were no longer recommended in 2007 (Fig. 3a and Supplementary Table 2), the growth of this lineage peaked in 2010, reflecting the fitness benefit of the mosaic *penA* 34 (Fig. 5a). The shift in use to cephalosporins plus azithromycin in 2010 was accompanied by a fitness benefit from *mtr* variants; however, the loss of fitness benefit from the mosaic *penA* 34 after the shift to ceftriaxone 250 mg plus azithromycin as the only recommended treatment in 2012 led to the accumulation of fitness costs from other resistance determinants and an overall negative growth rate.

The per-lineage analysis also addressed the question of what drove the growth of lineage 21, indicating that its expansion post 2010 was driven primarily by the presence of GyrA 91F/95A (Fig. 5b).

While the fitness contributions of the set of resistance determinants in our model accounted for much of the lineage dynamics, some dynamics remained unexplained. We first sought to highlight lineages for which a manual inspection would be beneficial. To do so, we computed the number of years in which the absolute value of the average of the sum of the residual and lineage background terms exceeded a threshold of 0.1 for each lineage. We chose a threshold of 0.1 as it represents ~10% growth or decline in a given year, arguing that this is large enough to be epidemiologically meaningful while maintaining sensitivity. In 11/29 lineages (lineages 2–4, 11, 13, 15, 16, 23, 26–28), there was at least one such year, and in six of those (lineages 2–4, 11, 15, 27), there were at least three such years.

To investigate this pattern, we examined lineages 2, 3 and 4. Lineage 2 carried none of the determinants we included in our model and underwent substantial growth starting in 2012 and peaking in 2018. We revisited the resistance phenotypes and genotypes for this lineage and noted that the isolates in lineage 2 carry *tetM*, which confers high-level resistance to tetracycline-class antibiotics[36]. We did not include *tetM* in our model because we lacked data on the extent of tetracycline-class antibiotics use for *N. gonorrhoeae* treatment and for syndromic treatment of known or presumed chlamydial co-infection, and because none of the other lineages carry *tetM*. Lineages 3 and 4 had high residual effects for at least 3 years and carried none of the resistance determinants in our model. The large and consistently positive residual effects (Supplementary Fig. 13) thus point to factors other than the antibiotic pressures examined here in shaping these two lineages' success.

## Discussion

Antibiotic exposure selects for resistant strains over their susceptible counterparts, whereas in the absence of antibiotics the resistant strains may suffer a fitness cost. Here we used *N. gonorrhoeae* population genomics from large-scale surveillance data, detailed understanding of the genetics underlying antibiotic resistance, and data on antibiotic treatment to quantify the contribution of and the interactions among

antibiotic resistance determinants and how these shaped *N. gonorrhoeae* dynamics in the USA.

Models of antibiotic use–resistance relationships typically treat all phenotypically resistant strains as if they have the same fitness costs and benefits[37]. However, our findings for GyrA 91F/95G and GyrA 91F/95A suggest that a single amino acid difference in a resistance determinant may result in markedly different dynamics. Several potential explanations exist for this phenomenon. One possibility is that the reduced fitness cost of GyrA 91F/95A facilitates an overall fitness benefit in the presence of bystander exposure to fluoroquinolones, whereas GyrA 91F/95G is simply too costly to provide a net benefit from bystander fluoroquinolone exposure. Another possibility is a fitness benefit irrespective of treatment exposure in vivo. Studies have characterized an in vivo advantage of *N. gonorrhoeae* fluoroquinolone-resistant *gyrA* mutants compared to those carrying wild-type GyrA 91S/95D in a mouse model[38] in the absence of fluoroquinolone administration. The marked difference in fitness between GyrA mutants is also consistent with in vitro estimates of fitness differences between resistant *gyrA* mutants in *E. coli*[9]. These results underscore the importance of accounting for pathways to resistance when analysing and modelling antimicrobial resistance dynamics.

The carriage of GyrA 91F/95G was associated with a decrease in lineage fitness after fluoroquinolones were no longer recommended, and persistence of sublineages illuminate *N. gonorrhoeae*'s strategies for responding to this fitness change, including reversion to a susceptible allele and acquiring resistance to the new treatment. While the number of isolates and sublineages limited quantification of these phenomena, they suggest responses to pressures from both antibiotics and host environments.

Similarly, for mosaic *penA* 34, we saw clear evidence that this allele conferred a large growth rate advantage when cephalosporins other than ceftriaxone 250 mg were recommended, as compared to the baseline type that does not carry any of the resistance determinants studied. We estimated that this advantage was rapidly lost after the switch to ceftriaxone 250 mg, consistent with the observed decline of cefixime resistance[39].

PBP1 421P had a small fitness defect across the study period, a finding supported by our competition experiments in the murine model of gonorrhoea. PBP1 421P mainly provides penicillin resistance, with only a relatively modest increase in cephalosporin MICs[27], but the absence of a fitness benefit suggests that this resistance phenotype was insufficient even in the context of cephalosporin use to confer an advantage. We conclude that the PBP1 421P allele is likely a relic of the decades when penicillin was the treatment for gonorrhoea.

We estimated a large lineage growth rate benefit with the carriage of mosaic *mtrR* promoters once azithromycin co-treatment was introduced in 2010. This is consistent with the rapid expansion of lineages carrying mosaic *mtrR* promoters in Europe[40]. The trajectory of the largest lineage carrying mosaic *mtrR* promoter, lineage 29, plateaued around 2015 (Supplementary Fig. 20), as does the overall prevalence of azithromycin resistance (Extended Data Fig. 1). Possible explanations include a drop in azithromycin use[41] and sexual network-dependent fitness of the *mtrCDE* mosaics[42].

Investigating the fitness contributions at the level of individual lineages allowed us to interrogate how much of each lineage's growth trajectory can be explained by the combination of resistance determinants and changes in treatment. Several lineages did not carry any of the determinants and displayed a consistent trend in their residual terms. In the case of lineage 2 (Supplementary Fig. 14), we identified the presence of *tetM*, which confers resistance to tetracycline-class antibiotics. This suggests that for some lineages, there may be a substantial impact of bystander exposure to antibiotics not used directly to treat gonorrhoea on their fitness trajectories[41,43]. A similar phenomenon may explain the dynamics of lineages 3 and 4. Incorporating population-wide antimicrobial use may enable quantification of the

impact of bystander exposure. Lineages that displayed large residual effects despite carrying resistance determinants included in our model may reflect the impact of lineage background, environment pressures or bystander exposure, suggesting avenues for further investigation. With improved sampling and individual-level prescription data, birth–death-sampling models[44,45] may prove useful.

Our results demonstrate how the expanding collection of microbial genomic data together with antibiotic-prescribing data and phylodynamic modelling can be used to explain microbial ecological dynamics and quantify the fitness contributions of genetic elements in their changing in vivo environments. While we focused on *N. gonorrhoeae* and AMR, these methods could be more broadly applied to other microbes and pressures, aiding in efforts to understand how combinations of genetic elements inform strain fitness across antimicrobial exposure and other environmental pressures.

## Methods

### Genomic analysis
We collected publicly available genomic data, minimum inhibitory concentrations (MICs) and demographic data from GISP isolates (*n* = 5,367) sequenced between years 2000 and 2019[3,14–17]. De novo assembly was performed using SPAdes (v.3.12.0)[46] with the '–careful flag', and reference-based mapping to NCCP11945 (NC_011035.1) was done using BWA-MEM (v.0.7.17)[47]. We used Pilon v.1.23 to call variants (minimum mapping quality 20, minimum coverage 10×)[48] after marking duplicate reads with Picard v.2.20.1 (https://broadinstitute.github.io/picard/) and sorting reads with samtools (v.1.17)[49]. We generated pseudogenomes by incorporating variants supported by at least 90% of reads and sites with ambiguous alleles into the reference genome sequence. We mapped reads to a single copy of the locus encoding the 23S rRNA and called variants using the same procedures[50]. We identified resistance-associated alleles from de novo assemblies and pseudogenomes. Likewise, we identified the presence of single nucleotide variants (for example, mutations in *gyrA*, *parC*, *ponA* and *penA*) and the copy number of resistance-associated variants in 23S rRNA from variant calls. To determine the presence or absence of genes, mosaic alleles, promoter variants and small insertions or deletions, we used the results of blastn (v.2.9.0)[51] searches of assemblies for resistance-associated genes. We typed mosaic *penA* alleles according to the nomenclature in the NG-STAR database[52]. We defined mosaic *mtr* alleles as those with <95% identity to the *mtr* operon encoded by FA1090 (NC_002946.2). We defined *mtr* alleles as loss-of-function (LOF) if frameshifts or nonsense mutations led to the translated peptide being less than 80% of the length of the translated reference allele. We chose the 80% value as a conservative cut-off based on findings that the majority of *mtrR* alleles with premature stop codons produce peptides that are 23%–69% of the expected peptide length[53]. Likewise, in *mtrC*, the 2-bp deletion leading to a premature stop codon produces a peptide that is 35% of the expected length[53].

Before phylogenetic reconstruction, we filtered assembled genomes on the basis of the following criteria: (1) The total assembly length was longer than 1,900,000 bp and less than 2,300,000 bp. (2) Reference coverage was more than 30%. (3) Percentage of reads mapped to reference was at least 70%. (4) Less than 12% of positions were missing in pseudogenomes. This resulted in *n* = 4,573 retained samples.

### Lineage assignment and phylogenetic reconstruction
We used GUBBINS (3.4.3)[54] to estimate recombining regions and IQTREE (2.4.0)[55] for phylogenetic reconstruction. The molecular clock model was *GTR + G + ASC* as selected by MODELFINDER[56].

We estimated the dates of ancestral nodes using the resulting tree with BACTDATING[57] under the additive relaxed clock model[58]. The estimated substitution rate was $5.46 \times 10^{-6}$ substitutions per site per year with 95% credible interval of $(5.29 \times 10^{-6}, 5.65 \times 10^{-6})$ substitutions per genome per year, while the root date was estimated to be 1645 with 95%

credible interval of (1621, 1667). These values are largely in agreement with previous estimates[26]. We estimated ancestral states for all determinants under study apart from the 23S rRNA as the joint maximum-likelihood estimate under the F81 model[59] using PASTML[60]. In the case of *penA*, the ancestral state reconstruction was performed using allele types and the resulting reconstruction was then mapped to *penA* determinants for subsequent analysis and lineage calling. For the 23S rRNA, we used maximum-parsimony ancestral reconstruction based on the DELTRAN algorithm[61] as implemented in PASTML[60]. We chose this approach because the C2611T substitution is usually present in four copies, making reverse mutation unlikely, and DELTRAN prioritizes parallel mutation[61]. Furthermore, the 23S rRNA C2611T variant does not display a clonal pattern of inheritance apart from one exception (Supplementary Fig. 21).

We excluded samples with missing values in any of the determinants from the analysis before ancestral state reconstruction, leaving ($n$ = 4,464) samples. We defined a subset of tips as a lineage if it was the maximal subset such that there was no change in ancestral state in any of the loci across the unique path from each tip to the most recent common ancestor of the subset and the timing of the most recent common ancestor was estimated to no earlier than 1980 with at least 99% posterior probability. We then defined included lineages in the analysis if they contained at least 30 tips.

## Lineage-based hierarchical phylodynamic model: overview and rationale

Our aim was to study how interactions among six resistance-associated genes and operons: *gyrA*, *parC*, *ponA*, *penA*, the *mtr* operon and the 23S rRNA, and the major antimicrobial classes used as primary treatment for gonorrhoea between 1993–2019 (Supplementary Table 1; refs. 27,62) affected the success and failure of resistant *N. gonorrhoeae* lineages in the USA. For *gyrA*, we considered alleles given by codons 91 and 95 and for *parC*, combinations of codons at positions 86, 87 and 91. The determinants at *parC* act as mutations modulating the impact of *gyrA* resistance mutations[30]. Consequently, we used partial pooling to estimate the effects of *gyrA* determinants across different *parC* contexts. For *ponA*, we considered the L and P variants encoded at codon 421. For each of the loci that make up the *mtr* operon (*mtrC*, *mtrD*, *mtrR* and the *mtr* promoter), we considered whether the locus was non-mosaic, mosaic, affected by a loss-of-function mutation, and whether there was an A-deletion in the *mtr* promoter. For *penA*, we considered variants at each site listed by NG-STAR *penA* allele types[52], along with whether the *penA* allele was mosaic. Within the lineages derived from our dataset, we only observed variation at codon sites 501 and 542, and the presence of the mosaic *penA* 34 allele. The mosaic *penA* 34 allele carries variants at other sites; however, since these variants do not occur on other backgrounds in the dataset, we could not estimate their contributions. For 23S rRNA, we considered the presence of at least one copy carrying the C2611T substitution. While the 23S substitution A2059G is associated with a more dramatic increase in azithromycin resistance, we did not include it in any analysis as it appeared in fewer than 20 isolates. We did not include determinants at the *porB* locus despite this locus contributing to resistance[27], as these determinants are rapidly lost and re-acquired with a limited amount of vertical inheritance. This was evident from phylogenetic analysis of *N. gonorrhoeae* resistance[17] and our own analysis (Supplementary Fig. 22).

We used phylodynamic modelling[63] to mitigate the impact of inconsistent sampling on reconstruction of lineage ecology. Because coalescent phylodynamic modelling can be less sensitive than traditional incidence-based modelling to violations of sampling assumptions[64], it can accommodate the overrepresentation of antibiotic resistant specimens in collections of sequenced isolates[3,14,15].

The data used in the statistical model consisted of (1) $L$ genealogies $G = \{\mathbf{g}_i\}_{1 \le i \le L}$, each corresponding to a particular AMR-linked lineage (Fig. 1a); (2) resistance determinant presence by lineage (Table 1); and

(3) treatment data from GISP clinics (Extended Data Fig. 1). As our aim was to quantify the impact of individual AMR determinants on lineage success and failure, we estimated the growth rate of the lineage-specific effective population size $r(t) = \dot{N}e(t)/Ne(t)$ [20,23]. We extended previous work[20,23] to a multiple lineage, multiple treatment, multiple AMR determinant scenario by constructing a hierarchical Bayesian regression model that accounted for intrinsic variation among lineages. We formulated the growth rate of the effective population size as a hierarchical linear model to estimate how much of lineage growth and decline could be explained as a function of the interaction between AMR determinants and the pattern of antimicrobial use. Disentangling the contributions of individual AMR determinants from external factors required accounting for the overall epidemic dynamics, for which we included a global trend term shared by all lineages; the effect of lineage background on baseline fitness, for which we included lineage-specific terms; and the overdispersion in the growth and decline of individual lineages that occurs due to factors unaccounted for.

The growth rate of the effective population size serves as a proxy for lineage success and can be used to solve for the effective population size (equation 1). While the effective population size is not necessarily directly proportional to incidence (it is a nonlinear function of incidence and prevalence[64,65]), if fitness benefits are small in comparison to the per capita transmission rate $\beta(t)$, or if $\beta(t)$ is approximately constant, then the growth rate of the effective population size will approximately match the growth of the epidemic[20]. The effective population size can then be linked to individual genealogies via the coalescent likelihood (equation 2). The key quantity of interest was the marginal impact of individual determinants on lineage growth rates. This is formulated in equations 4 and 5.

Coalescent-based phylodynamic approaches condition on sampling and are therefore more robust to sampling bias across individual lineages[64]. The sampling bias present in the data[3,14,15] is a form of ascertainment bias for macrolide and cephalosporin resistant phenotypes. The lineages are defined on the basis of resistance determinants, and thus the resistance phenotype is approximately identical within a lineage. Note that the lineages are typically paraphyletic and exclude isolates that show evidence of a change in resistance determinant type. This means that the ascertainment bias translates to different sampling intensity between lineages, but not within a lineage. We reasoned that most of the impact of the ascertainment bias should be mitigated as the coalescent likelihood of a given lineage is conditional on sampling ('Lineage-based hierarchical phylodynamic model: construction of the likelihood' in Methods). We therefore worked within the coalescent framework as opposed to a birth–death-sampling process framework to mitigate the impact of sampling being artificially enriched for resistant phenotypes.

While the effective population size relates to census size only through a complex and potentially nonlinear relationship, this relationship can be made explicit under mild assumptions[64,66,67]. Phylodynamic approaches in the analysis of AMR have been limited to scenarios that either considered an interaction between a single population and covariate, such as treatment usage data[20], or to a simplistic scenario with multiple populations and only a single covariate[23]. Here we studied multiple sites, following previous work[22,24], but we expanded on those efforts by introducing covariates in the form of antimicrobial use data to incorporate the impact of a changing environment.

## Lineage-based hierarchical phylodynamic model: construction of the likelihood

Our aim was to study to what extent the growth and decline of individual lineages can be explained through the interaction of the motifs a given lineage carries at each of the loci in $S$ and the changes in antimicrobial use over time. Since we could not observe the population size directly, we instead modelled the effective population sizes from the genealogies in $G$ and used these estimates to inform past population

dynamics using phylodynamic modelling[63,67,68]. In particular, for a given lineage, we aimed to estimate the growth rates of the corresponding effective population size $r(t) = N\mathrm{e}(t)/N\mathrm{e}(t)$ [20,23]. From $r(t)$, $N\mathrm{e}(t)$ can be recovered as

$$N\mathrm{e}(t) = N\mathrm{e}(0) \exp \int_0^t r(\tau) \, d\tau \tag{1}$$

and related to the respective genealogy **g** using the coalescent likelihood[69],

$$p(\mathbf{g}|\lambda(t)) = \exp\left(-\int_{-\infty}^{\infty} \mathbb{1}\,[A(t) \geq 2] \binom{A(t)}{2} \lambda(t)\,dt\right) \prod_{i=1}^{n-1} \lambda(c_i) \tag{2}$$

where $\lambda(t) = 1/N\mathrm{e}(t)$, $A(t)$ is the number of active lineages present in the genealogy at time $t$, also referred to as the block counting process[70], and $c_i$ are the times of internal nodes corresponding to coalescent events.

Since neither the integral in equation 1 nor the one in equation 2 is tractable, we discretized time as a regular grid of size $N$, denoting the end points of this discretization as $t_1 < \cdots < t_i < \cdots < t_N$. We approximated $r(t)$ as a piecewise constant function on this regular grid and denoted this approximate quantity as $r_{t_i}$.

We defined $\Delta_t = t_i - t_{i-1}$. We then approximated $r(t)$ as well as $N\mathrm{e}(t)$ as a piecewise constant function. We measured time in reverse with $t_1$ corresponding to present and $t_N$ to the beginning of the first year for which treatment data were available. This approximation was originally introduced as a part of the skygrid approach[71].

For a lineage $l$, the discretized version of $N\mathrm{e}^{(l)}(t)$, denoted $\widetilde{N\mathrm{e}}^{(l)}(t)$ is then

$$\widetilde{N\mathrm{e}}^{(l)}(t) = N_0^{(l)} \exp\left(\sum_{i=1}^{N} r_{t_i}^{(l)} \Delta_t 1[t_{i-1} < t]\right) \tag{3}$$

## Lineage-based hierarchical phylodynamic model: detailed description of the statistical model

To model the fitness differences between lineages, we characterized lineage-specific growth rates $r_{t_i}^{(l)}$ as a function of the determinants a lineage carries and past antibiotic use. As in SKYGROWTH[20], our aim was to model the growth and decline in the effective population size $r(t) = N\mathrm{e}(t)$ as a function of covariates, while accounting for overdispersion. Expanding on past approaches[20,23], we modelled multiple covariates interacting with lineages through multiple resistance-linked loci, along with random effects to account for overdispersion and the impact of lineage backgrounds.

Using $G = \{\mathbf{g}_i\}_{1 \leq i \leq L}$, we denoted each of the $L$ genealogies corresponding to the $L$ lineages under study. We considered 4 genomic regions associated with resistance and denoted these with $S$. $S$ consisted of an indicator for the *gyrA* locus, the *penA* locus, the *ponA* locus, 23S rRNA, as well as an indicator for the *mtrCDE* operon. For each lineage, we kept track of AMR determinants present in each of the regions in $S$. Each region in $S$ then interacts with the proportion at which a subset of the 4 treatment classes was used. We denoted the set of these treatment classes as $D$. $D$ consisted of indicators for fluoroquinolones, cephalosporins other than ceftriaxone 250 mg, and azithromycin co-treatment. For each treatment class, the data consisted of the proportion of cases treated with that class as primary treatment.

Using $m(s) : s \in S$, we denoted the set of determinant indices associated with the region $s$ that can be found in that dataset we studied. For example, for the *gyrA* locus, this consisted of indices for 91F/95G and 91F/95A.

Using $I(s) \subseteq D : s \in S$, we denoted the subset of treatment class indices that the region $s$ interacts with (Supplementary Table 1). For example, for $s$ corresponding to the *penA* locus, $I(s)$ consists

of the index for cephalosporins other than ceftriaxone 250 mg. $u_t(d) \in [0,1] : d \in D$ denoted the proportion of cases treated with treatment $d$ at time $t$. $X_s(j,l) \in \{0,1\}$ was an indicator function equal to 1 if the determinant $j$ at region $s$ was present in lineage $l$, and 0 otherwise.

We denoted the set of regions that depend on known or suspected compensatory mutations as $S^c \subseteq S$. For each $s \in S^c$, we defined $G(s)$ to be the set of indices corresponding to the types of compensatory mutation. $g_s(l) \in G(s)$ was the type of compensatory mutation associated with region $s$ that is present in lineage $l$. In our case, $S^c$ consisted of just one index for *gyrA*. For *gyrA*, $G(s)$ consisted of indices corresponding to three different *parC* types: 86D/87S/91G, 86D/87R/91E, 86N/87S/91E.

Using this notation, we introduced terms that characterize the mean effect of AMR determinants on lineage $N\mathrm{e}(t)$ growth rates. For determinant a $j \in m(s)$ belonging to region $s$, we denoted the effect at time $t_i$ as $f_{t_i}^{s,j}$:

$$f_{t_i}^{s,j} = \alpha_j^s + \sum_{d \in I(s)} \beta_{j,d}^s u_{t_i}(d) \tag{4}$$

This can be understood as a linear model characterizing the impact of a given determinant on the $N\mathrm{e}(t)$ growth rate of a lineage as a function of past treatment composition compared to the baseline. The coefficients $\alpha_j^s$ represent the change in intercept for determinant $j$ in region $s$, and the coefficients $\beta_{j,d}^s$ represent the change in the growth rate associated with the presence of determinant $j$ in region $s$ when the proportion of primary treatment corresponding to antimicrobial class $d$ changes. For regions that depend on compensatory mutations, these coefficients represent the average coefficient across all compensatory mutation backgrounds.

For those determinants that depend on compensatory mutations, we defined the effect of the compensatory mutation background as

$$\tilde{f}_{t_i}^{s,j}|k = \left(\tilde{\alpha}_j^s|k\right) + \sum_{d \in I(s)} \left(\tilde{\beta}_{j,d}^s|k\right) u_{t_i}(d)\ k \in G(s) \tag{5}$$

The term $\tilde{\alpha}_j^s|k$ represents the change in the intercept term for determinant $j$ on background indexed by $k$ compared to the the determinant's mean intercept $\alpha_j^s$. Analogously, $\tilde{\beta}_{j,d}^s|k$ represents the impact of the compensatory background indexed by $k$ on the antimicrobial–determinant interaction coefficients.

We modelled the growth rate of lineage $l$ at time $t_i$, denoted by $r_{t_i}^{(l)}$ as:

$$
\begin{aligned}
r_{t_i}^{(l)} = {}& \mu_{t_i} + (\epsilon_{t_i}|l) + (a|l) \\
& + \sum_{d \in D} \beta_d u_{t_i}(d) \\
& + \sum_{s \in S} \sum_{j \in m(s)} \chi_s(j,l) f_{t_i}^{s,j} \\
& + \sum_{s \in S^c} \sum_{j \in m(s)} \chi_s(j,l) \left(\tilde{f}_{t_i}^{s,j}|g_s(l)\right)
\end{aligned}
\tag{6}
$$

The first line corresponds to the global trend $\mu_{t_i}$, the lineage background intercept $a|l$ and the lineage residual term $\epsilon_{t_i}|l$ that accounts for overdispersion in the lineage trajectory.

The second line corresponds to the impact of changing treatment policy on the growth rate of lineages with only the baseline determinants. This impact is represented by coefficients $\beta_d$.

The third line contains the mean effects of the determinants carried by a lineage on its growth rate.

The final line accounts for the impact of compensatory mutation background on the effect of determinants present in the corresponding regions.

The last step was to characterize the previous model that was used for the parameters included in equation 6. Note that the choice

of priors required a choice of a time scale. We therefore set the unit of time to 1 year.

We modelled the global trend $\mu_{t_i}$ as a Gaussian random walk with independent, stationary increments and an unknown marginal standard deviation $\tau_\mu$, itself equipped with a half-normal prior:

$$\mu_{t_i} - \mu_{t_{i-1}} \sim \mathcal{N}\left(0, \sqrt{\Delta_t}\tau_\mu\right) \tag{7}$$

$$\tau_\mu \sim \mathcal{N}(0,1)1_{[0,\infty)} \tag{8}$$

$$\mu_{t_1} \sim \mathcal{N}(0,1) \tag{9}$$

We modelled the lineage-specific residual terms $\epsilon_{t_i}|l$ as:

$$\epsilon_{t_i}|l \sim \mathcal{N}\left(0, \tau_\epsilon/\sqrt{\Delta_t}\right) \tag{10}$$

$$\tau_\epsilon \sim \mathcal{N}(0,1)1_{[0,\infty)} \tag{11}$$

This can be understood as a multilevel model component accounting for temporal trends in lineage trajectories not explained by either the motifs that we focus on or the global trend.

The hierarchical prior on the lineage intercept terms was:

$$a|l \sim \mathcal{N}(0, \sigma_a) \tag{12}$$

$$\sigma_a \sim \mathcal{N}(0,1)1_{[0,\infty)} \tag{13}$$

For the baseline coefficients treatment impact coefficients $\beta_d$, we used standard normal priors, which corresponds to a weakly informative choice:

$$\beta_d \sim \mathcal{N}(0,1) \tag{14}$$

We used a similar strategy for the site and motif specific interaction coefficients $\beta^s_{j,d}$. Motifs at loci associated with resistance to several classes of antimicrobials (Supplementary Table 1) will typically increase or decrease resistance to all such classes at once. To account for this correlation for sites that do not have compensatory mutations, we modelled $\beta^s_{j,d}$ for a fixed $s,j$ as correlated:

$$\beta^s_{j,d} \sim \mathrm{MVN}(0, \phi I + (1-\phi)\mathbf{1}) \tag{15}$$

$$\phi \sim U([0,1]) \tag{16}$$

Where MVN stands for a multivariate normal distribution parametrized using a mean and a square root of the covariance matrix, **1** represents the matrix with unit entries, $I$ is the identity matrix, and $U([0,1])$ is the uniform distribution on the unit interval. $\phi$ is a pooling factor that determines the level of correlation between the individual antimicrobial specific coefficients.

We modelled the effect of compensatory mutation background with the following prior

$$\widetilde{\alpha}^s_j|k \sim \mathcal{N}(0, \sigma^\alpha_s) \tag{17}$$

$$\sigma^\alpha_s \sim \mathcal{N}(0,1)1_{[0,\infty)} \tag{18}$$

and

$$\widetilde{\beta}^s_{j,d}|k \sim \mathcal{N}\left(0, \sigma^\beta_s\right) \tag{19}$$

$$\sigma^\beta_s \sim \mathcal{N}(0,1)1_{[0,\infty)} \tag{20}$$

This hierarchical prior enables us to estimate the overall impact of compensatory mutations. For the initial lineage effective population sizes $N^{(l)}_0$, we used hierarchical log-normal distributions

$$\mu_{Ne} \sim \mathcal{N}(4,4) \tag{21}$$

$$\sigma_{Ne} \sim \mathcal{N}(0,1)1_{[0,\infty)} \tag{22}$$

$$\log N^{(l)}_0 \sim \mathcal{N}(\mu_{Ne}, \sigma_{Ne}) \tag{23}$$

## Lineage-based hierarchical phylodynamic model: implementation

The model was implemented in the STAN probabilistic programming language[72] and R language (v.4.4.0)[73]. Sampling was performed using Hamiltonian Monte Carlo as implemented in STAN[72]. Four chains were run in parallel for 1,000 sampling iterations each. All chains terminated under 2 h on a laptop equipped with an Apple M3 Pro CPU and 36 GB of RAM. For all model parameters, the bulk effective sample size (bulk-ESS) was always at least 500 and the $\hat{R}$ statistic always lower than 1.05 (ref. [74]).

## Lineage-based hierarchical phylodynamic model: assumptions and limitations

To quantify the impact of a determinant on lineage dynamics, the determinant must have a large effect and appear on multiple lineage backgrounds. Even as we captured the dominant effects of resistance determinants, there is still substantial uncertainty in the fitness landscape of *N. gonorrhoeae*. Reducing this uncertainty requires either a larger number of sequences than used here, more representative sampling, or both. Improved sampling may enable use of birth–death-sampling processes[44,45]. Larger sample sizes would enable more robust estimates under more complex models that could, for example, accommodate time-varying relationships between prescribing data and the growth rate effect of resistance determinants.

We used a two-step approach for the phylodynamic analysis, in keeping with the common practice for bacteria[20,23,24], whereby a dated phylogeny is reconstructed first and then used as a fixed input for downstream phylodynamic analysis. This approach neglects the propagation of uncertainty from the phylogenetic reconstruction and phylogenetic dating through to the phylodynamic analysis while potentially reducing temporal signal due to over- or underregularization by the model used for reconstructing the dated phylogeny. Relaxing this approximation would require the development of a scalable approach to jointly infer the ancestral dates of multiple lineage trees along with the parameters of the phylodynamic model. Furthermore, we defined fixed lineages, and this may result in fragmentation of otherwise linked lineages and exclusion of lineages smaller than the threshold size.

Deviations from the model are captured by the residual terms. Non-parametric methods such as splines or Gaussian processes could be used to model the relationship between treatment composition, time and growth rate effect of determinants, thus enabling more precise attribution of fitness fluctuations.

The approach presented is only applicable for determinants that give rise to lineage-like dynamics. This effectively means that the estimates for the determinants are valid for sufficiently compatible genetic backgrounds where any putative fitness cost is not too large. If the fitness costs were large, the observed dynamics would probably resemble mutation–selection balance in the case of strong mutation and strong negative selection[8]. Likewise, the requirement for lineage-like dynamics effectively conditions for determinants to be present on compatible genetic backgrounds, as it is unlikely that a clone would give rise to a major lineage in the absence of compatibility between the genetic background and the resistance determinant.

We ignored spatial heterogeneity in transmission and treatment. Relaxing this assumption would require access to spatially resolved treatment data as well as denser sampling. As the data collection is overall sparse with just over 4,400 sequences retained for phylodynamic analysis, compared to over 640,000 reported cases of gonorrhoea in 2022[18], heterogeneity within the USA in terms of transmission and treatment is unlikely to impact the results.

Importations from outside of the USA may distort the results. Due to the focus on only major lineages and the size of the *N. gonorrhoeae* epidemic in the USA, we do not expect this to play a major role, and any remaining effects of importation should be compensated for by the residual overdispersion terms in the statistical model.

## Comparison of the impact of GyrA 91F/95G versus GyrA 91F/95A on ciprofloxacin MICs

To investigate possible differences between the levels of fluoroquinolone resistance, we selected all isolates that carried ParC 86D/87R/91E and GyrA 91F/95G or GyrA 91F/95A. For each of these isolates, we $\log_2$-transformed the corresponding ciprofloxacin MIC, removing any isolates that were either missing a ciprofloxacin MIC value or had a reported MIC of 0. To assess the MIC difference between GyrA 91F/95G and GyrA 91F/95A on ParC 86D/87R/91E background, we then fitted a linear model using the 'lm' function in R (v.4.4.0)[73], while controlling for determinants at the *mtrCDE* operon. The model used was standard linear regression with categorical predictors as specified by the following regression formula:

$$\log_2 \text{MIC} \sim gyrA + mtrC + mtrD + mtrR + mtrR \text{ promoter} \quad (24)$$

## *gyrA* mutants competition assay: *N. gonorrhoeae* culture conditions

*N. gonorrhoeae* was cultured on GCB agar (Difco) supplemented with Kellogg's supplement (GCB-K) at 37 °C with 5% $CO_2$[75]. We performed pairwise competition experiments in liquid GCP medium containing 15 g l$^{-1}$ proteose peptone 3 (Thermo Fisher), 1 g l$^{-1}$ soluble starch, 1 g l$^{-1}$ KH$_2$PO$_4$, 4 g l$^{-1}$ K$_2$HPO$_4$ and 5 g l$^{-1}$ NaCl (Sigma-Aldrich) with Kellogg's supplement[76].

## *gyrA* mutants competition assay: generation of isogenic *N. gonorrhoeae* strains and antibiotic susceptibility testing

Antibiotic susceptibility testing for ciprofloxacin was performed on GCB-K agar via Etest (BioMerieux). All MIC results represent the mean of three independent experiments. Strains, plasmids and primers used for *gyrA* mutant experiments are listed in Supplementary Tables 3–6. All isogenic *N. gonorrhoeae* strains were generated in a ciprofloxacin-resistant clinical isolate, GCGS0481, which carries GyrA 91F/95G and ParC 87R. To clone a GyrA 91S/95D fragment with a chloramphenicol resistant cassette (CMR), pAM_3 plasmid was constructed using Gibson assembly in a pUC19 (ref. 77) backbone. The GyrA 91S/95D fragment was amplified from pDRE77 (ref. 78) using the primer pair AM_7 and AM_8 and the chloramphenicol cassette from pKH37 (ref. 79) using the primer pair AM_9 and AM_10. Fragments were amplified using Phusion high-fidelity DNA polymerase (NEB), checked for appropriate size by gel electrophoresis, column purified (Qiagen PCR purification kit), assembled with Gibson Master Mix (NEB) and transformed into chemically competent DH5α *E. coli* (Invitrogen). Individual colonies were selected on LB agar supplemented with 20 µg ml$^{-1}$ chloramphenicol and grown overnight at 37 °C. Plasmids were isolated using Miniprep kit (Qiagen) according to manufacturer instructions, and sequences were confirmed by Sanger sequencing. For the insertion of the GyrA 91S/95D allele into *N. gonorrhoeae* GCGS0481, the isolate was grown overnight on a GCB-K plate at 37 °C with 5% $CO_2$. After 16–20 h, the strain was scraped and suspended in 0.3 M sucrose (Sigma-Aldrich), electroporated with 200 ng of pAM_3 plasmid and rescued with GCP

medium supplemented with Kellogg's for 30 min. The transformants were then plated on non-selective GCB-K agar plates for 4–6 h, followed by selection on GCB-K plates supplemented with 4.5 µg ml$^{-1}$ chloramphenicol. Finally, individual colonies were re-streaked on non-selective GCB-K agar plates and the *gyrA* allele checked by Sanger sequencing. For cloning of GyrA 91F/95G and GyrA 91F/95A, fragments of *gyrA* were amplified using primers AM_5 (F) and AM_6 (R) from the genomic DNA of clinical *N. gonorrhoeae* isolates GCGS0481 and NY0842, respectively. Electroporation was done as described above, and individual colonies were selected on GCB-K plates supplemented with 2 µg ml$^{-1}$ ciprofloxacin. For all transformations performed, transformations without DNA were used as negative controls. Genomic DNA from each strain was purified using an Invitrogen PureLink Genomic DNA mini kit (K182001), prepared for sequencing using Oxford Nanopore Technologies Native Barcoding Kit 24 V14 (SQK-NBD114.24) and sequenced on an Oxford Nanopore Technologies R10.4.1 flowcell followed by base calling with dorado v.0.8.1 with super accuracy. Base-called reads were uploaded to the NCBI SRA and are available at PRJNA1281689.

## *gyrA* mutants competition assay: competitive fitness measurement of GyrA variants

GCGS0481 GyrA 91S/95D, GyrA 91F/95G and GyrA 91F/95A containing the CMR cassette were transformed with pDR53, a kanamycin cassette (KanR) derivative of pDR1 (ref. 80) (constructed using the primer pair DR_395 and DR_396). The resulting transformants were selected on GCB-K agar supplemented with 70 µg ml$^{-1}$ kanamycin. Colony PCR was performed to screen the kanamycin positive clones using the primer pair DR_62 and DR_63 (Supplementary Table 6). During the pairwise competition experiments, the competitive paired strains from overnight cultured plates (one kanamycin-sensitive and one kanamycin-resistant strain) were mixed and co-cultured (at a ratio of 1:1 by optical density) in antibiotic-free GCP media with Kellogg's supplement for 8 h. At each timepoint, cultures were serially diluted, and the same volume was plated on both GCB-K agar and GCB-K agar supplemented with 70 µg ml$^{-1}$ kanamycin. Finally, dilutions on both plates were quantified and the competitive index (CI) was calculated at each timepoint by counting colony-forming units (c.f.u.s). The CI value at any timepoint was calculated as $(R_t/S_t)/(R_0/S_0)$, where $R_t$ and $S_t$ are the proportions of kanamycin-resistant and kanamycin-sensitive strains, respectively, at any timepoint, and $R_0$ and $S_0$ are the proportions of kanamycin-resistant and kanamycin-sensitive strains at time 0. Statistical analysis of CI measurements was performed using unpaired two-sided Student's *t*-test.

## *ponA* mutants competition assay: generation of strains and plasmids

Strains, plasmids and primers used for *ponA* mutant experiments are listed in Supplementary Tables 7–9. Strains used in these in vivo and in vitro competitive fitness experiments were made streptomycin resistant to facilitate mouse infection studies[81]. To achieve this, the *rpsL1* allele from FA1090, which encodes a mutant form of ribosomal protein S12 that confers streptomycin resistance[82], was transformed into the penicillin-resistant clinical isolate FA6140 (ref. 34) and the antibiotic-susceptible laboratory strain FA19 (ref. 83) by allelic exchange[84]. Clones that acquired *rpsL1* were selected on GCB agar plates containing 100 µg ml$^{-1}$ streptomycin and verified by Sanger sequencing. FA19 rpsL and FA6140 rpsL were transformed with a pUC plasmid containing a portion of the *ponA* gene starting at 831 bp and harbouring either 421L or 421P, the *aad1* resistance cassette ($\Omega$) conferring spectinomycin/streptomycin resistance, and 531 bp of sequence downstream of *ponA* to facilitate recombination. Transformants were selected on GCB agar plates containing 25 µg ml$^{-1}$ spectinomycin and verified by Sanger sequencing. All gonococcal strains were propagated on solid GCB agar containing Kellogg's supplements I and II[75] for 18–20 h at 37 °C in a 5% $CO_2$-enriched atmosphere.

## *ponA* mutants competition assay: in vitro growth curve experiments

Non-piliated bacterial colonies were collected and inoculated into GCB supplemented with Kellogg's supplements I and II[75]. Cultures were shaken at 180 r.p.m. at 37 °C in a 5% $CO_2$-enriched atmosphere, and bacterial growth was assessed by measuring the optical density $(OD)_{600}$ at hourly intervals for a total of 8 h. Experiments were repeated in biological triplicate, and statistical significance between individual strains was determined using a repeated-measures two-way analysis of variance (ANOVA) with Tukey's multiple comparisons.

## *ponA* mutants: competitive murine infection experiments

Animal experiments were conducted at the Uniformed Services University of the Health Sciences according to the guidelines of the Association for the Assessment and Accreditation of Laboratory Animal Care under protocol MIC-23-488 that was approved by the University's Institutional Animal Care and Use Committee. Mice were housed under standard environmental conditions of 68–79 °F (20–26 °C), a humidity range of 30–70% and a 12-h light/dark cycle (0600–1800). Female BALB/c mice (6–8 weeks old; National Cancer Institute) were treated with water-soluble 17$\beta$-estradiol and antibiotics to increase susceptibility to *N. gonorrhoeae*[85]. Groups of female BALB/c mice were inoculated vaginally with 20 µl of a PBS suspension containing similar numbers of wild-type FA19 *rpsL* PBP1 421L and isogenic FA19 *rpsL* PBP1 421P; or wild-type FA6140 *rpsL* PBP1 421P and isogenic FA6140 *rpsL* PBP1 421L c.f.u.s (total dose, $10^6$ c.f.u.s; 7 mice per group). Vaginal swabs were collected on days 1, 3 and 5 post inoculation and suspended in 1.0 ml GCB. Vaginal swab suspensions and inocula were cultured quantitatively on GCB agar with streptomycin (100 µg ml$^{-1}$) for total c.f.u. counts and GCB agar with streptomycin (100 µg ml$^{-1}$) and spectinomycin (25 µg ml$^{-1}$) for FA19 *rpsL* PBP1 421P or FA6140 *rpsL* PBP1 421L c.f.u. counts. Results were expressed as the competitive index (CI) by counting c.f.u.s. The CI value at any time-point was calculated as $(R_t/S_t)/(R_0/S_0)$, where $R_t$ and $S_t$ are the proportions of spectinomycin-resistant and spectinomycin-sensitive strains, respectively, at any timepoint, and $R_0$ and $S_0$ are the proportions of spectinomycin-resistant and spectinomycin-sensitive strains at time 0.

If only one strain was recovered from an infected mouse in a swab, the limit of detection of 1 c.f.u. was assigned to the strain that was not recovered, and the CI was calculated and plotted at this limit of detection. When swabs yielded no culturable *N. gonorrhoeae*, the corresponding data point was omitted from the CI plot. FA19 competition experiments were performed in biological duplicate (total 7 mice per group). FA6140 competition experiments were performed once (total 7 mice per group). Statistical analysis of CI values was performed using Mann–Whitney test.

### Reporting summary

Further information on research design is available in the Nature Portfolio Reporting Summary linked to this article.

## Data availability

The data necessary to reproduce the statistical analysis, along with the metadata for the isolates analysed, are available in Zenodo at https://doi.org/10.5281/zenodo.18001481 (ref. 86). The accession numbers for the sequenced isolates used in this analysis are available as part of the metadata table included in the repository.

Base-called nanopore reads for generated GyrA mutants were uploaded to the NCBI SRA and are available under accession ID PRJNA1281689. Source data are provided with this paper.

## Code availability

The code necessary to reproduce the analysis in this paper is available in Zenodo at https://doi.org/10.5281/zenodo.18001481 (ref. 86).

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

## Acknowledgements

We thank D. H. F. Rubin for the pDR53 plasmid. This work was supported by NIH R01 AI132606 to Y.H.G. and R01 AI153521 to Y.H.G., A.E.J. and R.A.N.

## Author contributions

D.H. and Y.H.G. conceptualized the study. D.H. and T.D.M. designed and performed the computational analysis. A.M., G.G., A.L.V. and S.B. performed the competition assays. D.H. and Y.H.G. wrote the original draft. Y.H.G. acquired funding. D.H., Y.H.G., T.D.M., A.M., S.G.P., R.A.N. and A.E.J. discussed the results and contributed to writing, reviewing and editing of the paper.

## Competing interests

The authors declare no competing interests.

## Additional information

**Extended data** is available for this paper at https://doi.org/10.1038/s41564-025-02235-w.

**Correspondence and requests for materials** should be addressed to Yonatan H. Grad.

**Reviewer recognition** *Nature Microbiology* thanks the anonymous reviewer(s) for their contribution to the peer review of this work. Peer reviewer reports are available.

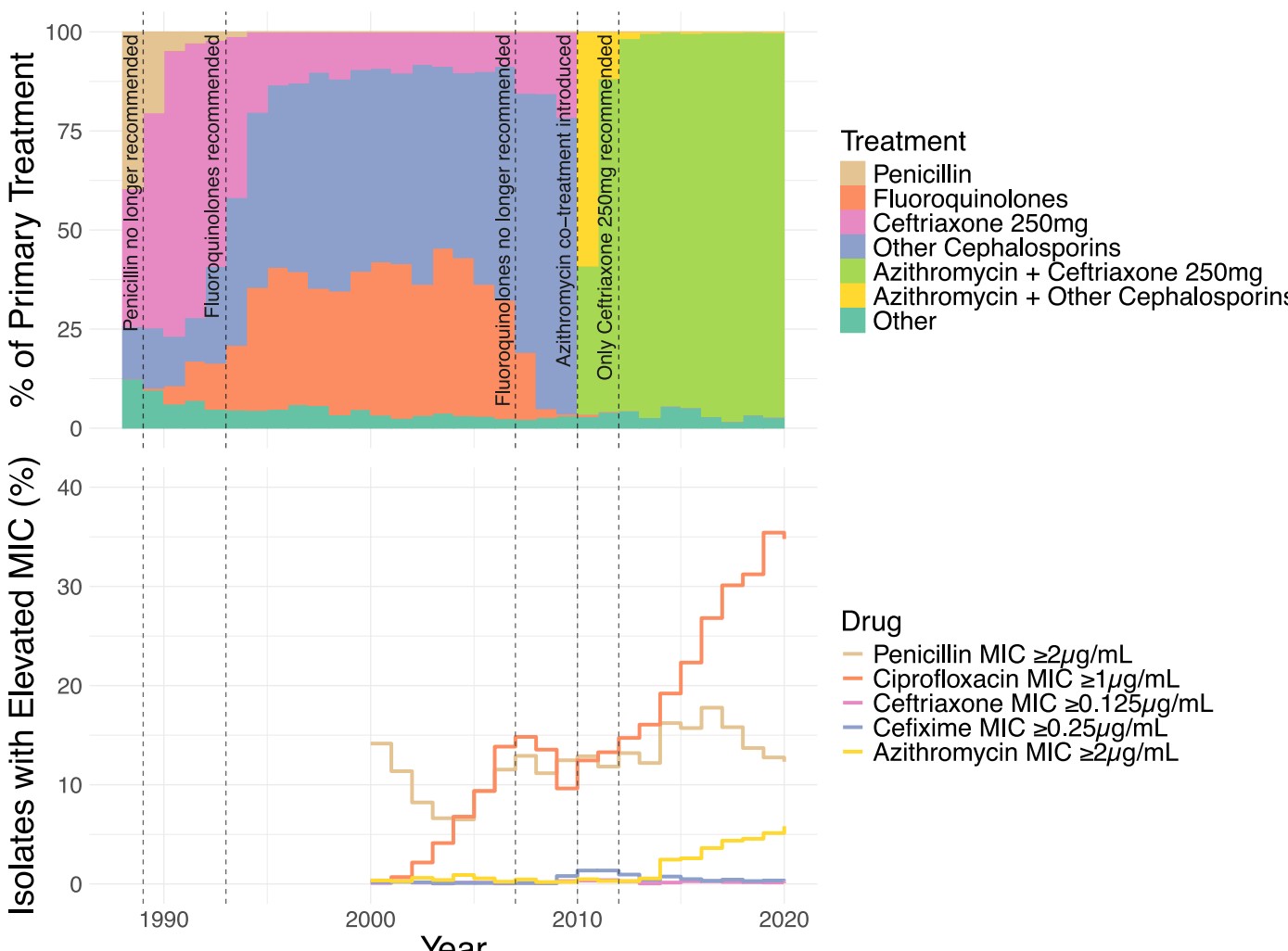

**Extended Data Fig. 1 | Gonococcal Isolate Surveillance Project treatment and resistance trends.** Data from *CDC's 2022 STI Surveillance Report*[18]. Top panel: Past treatment composition. The fluoroquinolone category consists of ciprofloxacin and ofloxacin. Other cephalosporins consists of cefixime, ceftriaxone 125 mg, and otherwise unspecified cephalosporins. Bottom panel:

Prevalence of resistance over time. Note that resistance data is only available up from year 2000 onwards, and treatment data from 1988 onwards. CDC data does not report usage rates for azithromycin, however azithromycin co-treatment has been recommended with all cephalosporins since 2010 until 2020[18]. We therefore impute it to be equal to the total proportion for all cephalosporins past 2010.

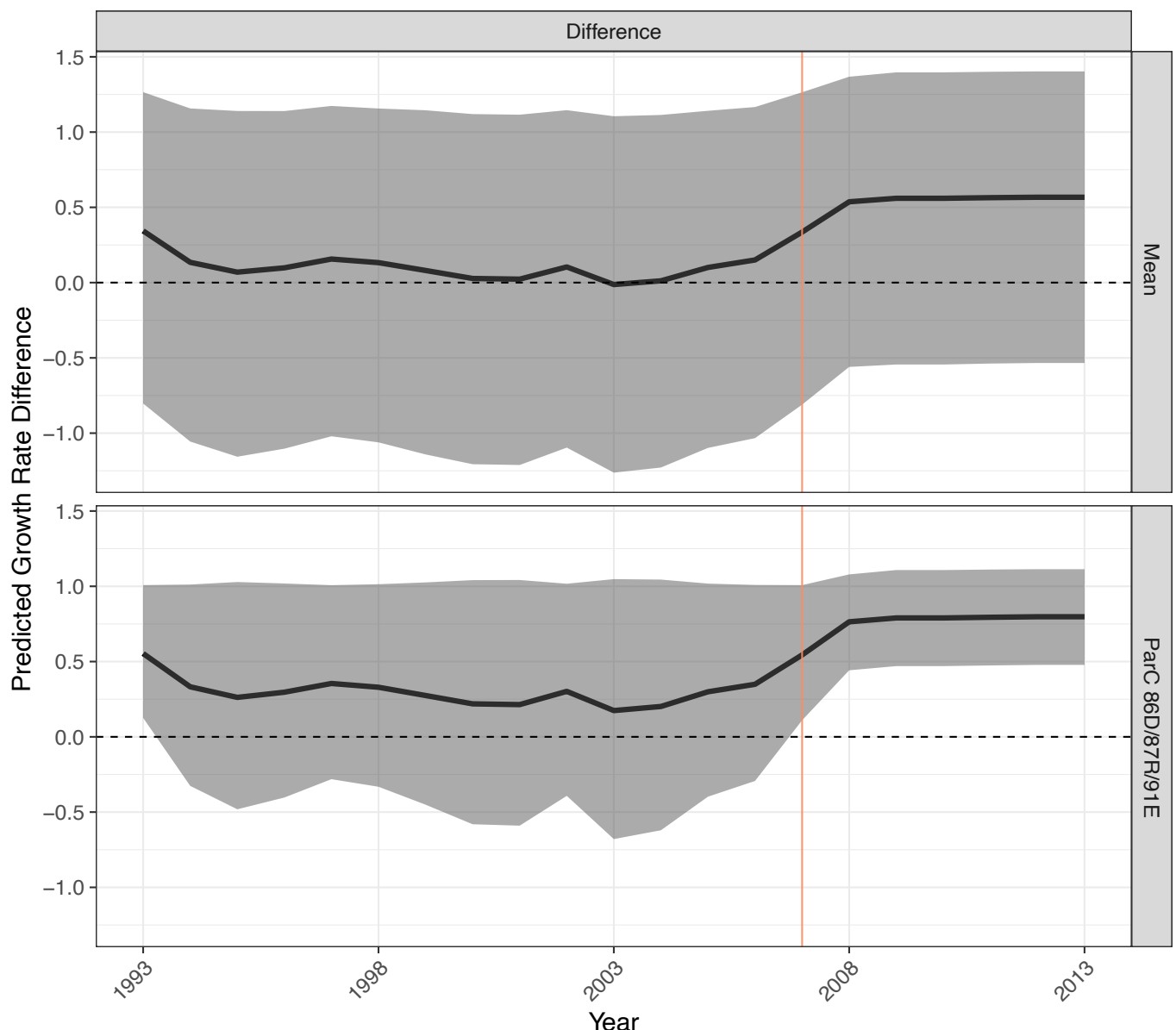

**Extended Data Fig. 2 | The difference between the predicted absolute growth rate effect as a function of fluoroquinolone usage of GyrA 91 F/95 G and GyrA 91 F/95 A averaged across all *parC* backgrounds and on ParC 86D/87 R/91E background.** The shaded region denotes the 95% posterior credible interval around the posterior median, depicted by the bold black line. Dashed line denotes 0.

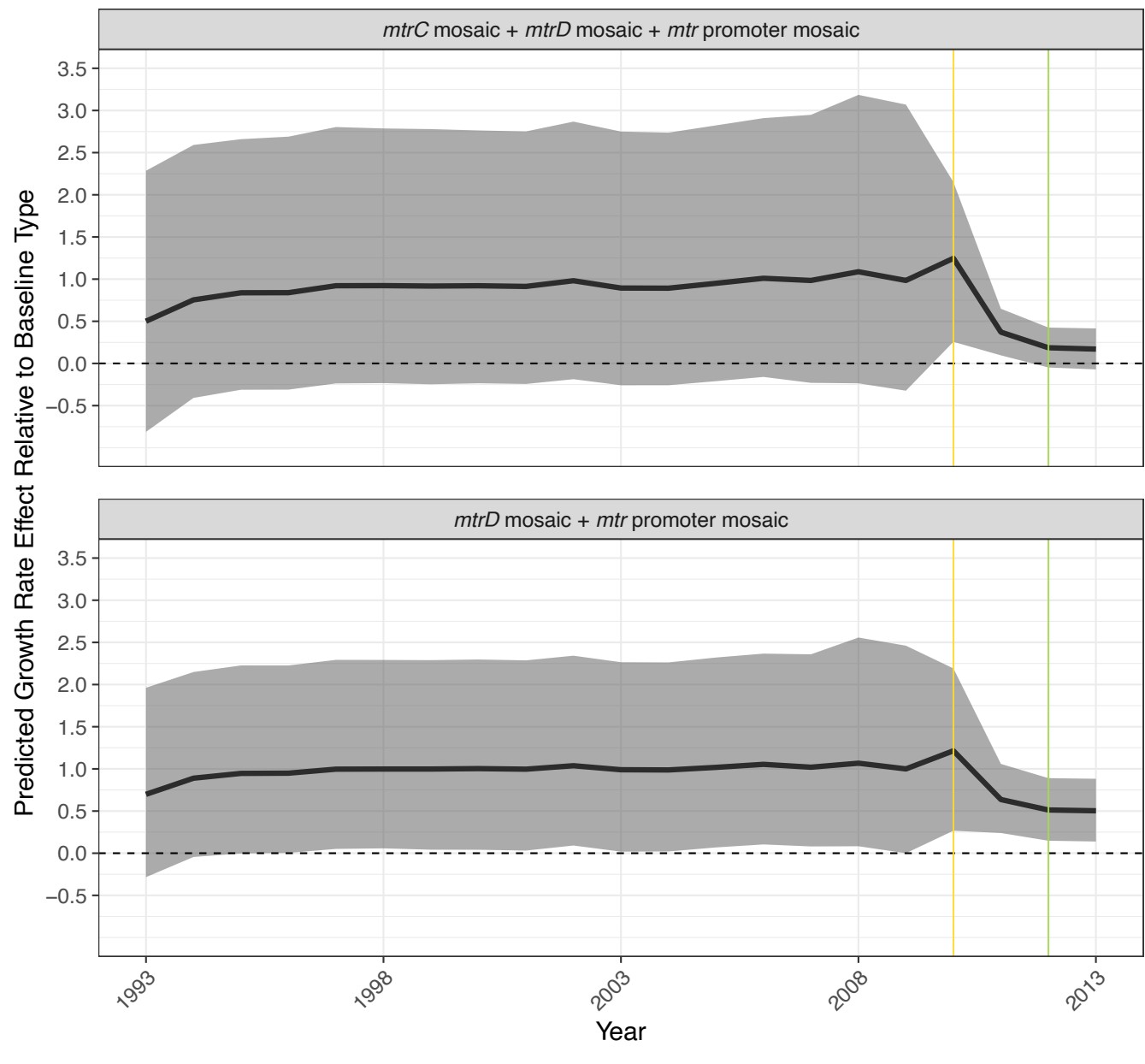

**Extended Data Fig. 3 | Predicted absolute effect for the total impact of the two most common combinations of determinants at the *mtrCDE* locus.** The predicted effect was computed on the basis of reported treatments. The shaded region denotes the 95% credible interval around the posterior median, depicted by the bold black line. Dashed line denotes no predicted growth rate effect relative to the baseline type that does not carry any of the resistance determinants studied. Azithromycin co-treatment refers to the guidance recommending the co-administration of azithromycin and any recommended cephalosporin.

# Reporting Summary

## Statistics

For all statistical analyses, confirm that the following items are present in the figure legend, table legend, main text, or Methods section.

| n/a | Confirmed | |
|---|---|---|
| ☐ | ☒ | The exact sample size (*n*) for each experimental group/condition, given as a discrete number and unit of measurement |
| ☐ | ☒ | A statement on whether measurements were taken from distinct samples or whether the same sample was measured repeatedly |
| ☐ | ☒ | The statistical test(s) used AND whether they are one- or two-sided *Only common tests should be described solely by name; describe more complex techniques in the Methods section.* |
| ☐ | ☒ | A description of all covariates tested |
| ☐ | ☒ | A description of any assumptions or corrections, such as tests of normality and adjustment for multiple comparisons |
| ☐ | ☒ | A full description of the statistical parameters including central tendency (e.g. means) or other basic estimates (e.g. regression coefficient) AND variation (e.g. standard deviation) or associated estimates of uncertainty (e.g. confidence intervals) |
| ☐ | ☒ | For null hypothesis testing, the test statistic (e.g. *F*, *t*, *r*) with confidence intervals, effect sizes, degrees of freedom and *P* value noted *Give P values as exact values whenever suitable.* |
| ☐ | ☒ | For Bayesian analysis, information on the choice of priors and Markov chain Monte Carlo settings |
| ☒ | ☐ | For hierarchical and complex designs, identification of the appropriate level for tests and full reporting of outcomes |
| ☒ | ☐ | Estimates of effect sizes (e.g. Cohen's *d*, Pearson's *r*), indicating how they were calculated |

*Our web collection on statistics for biologists contains articles on many of the points above.*

## Software and code

Policy information about availability of computer code

| Data collection | No software was used for data collection |
|---|---|
| Data analysis | Analysis of experimental data was carried out in Graphpad Prism. Open-source software required for genomic analysis is described in Materials and Methods. Software required to reproduce the computational analysis is available at https://github.com/gradlab/GC_AMR_Lineages<br>Software used for genomic analysis:<br>- Gubbins 3.4.3<br>- IQTree 2.4.0<br>- SPAdes 3.12.0<br>- BWA-MEM 0.7.17<br>- Pilon 1.23<br>- samtools 1.17<br>- blastn 2.9.0<br>- BactDating<br>- PastML web version |

For manuscripts utilizing custom algorithms or software that are central to the research but not yet described in published literature, software must be made available to editors and reviewers. We strongly encourage code deposition in a community repository (e.g. GitHub). See the Nature Portfolio guidelines for submitting code & software for further information.

## Data

Policy information about availability of data

All manuscripts must include a data availability statement. This statement should provide the following information, where applicable:
- Accession codes, unique identifiers, or web links for publicly available datasets
- A description of any restrictions on data availability
- For clinical datasets or third party data, please ensure that the statement adheres to our policy

Genomic data, metadata, and accession numbers are publicly available in previously published work and listed in the github repository https://github.com/gradlab/GC_AMR_Lineages. Base-called Nanopore reads for generated GyrA mutants were deposited to NCBI SRA under PRJNA1281689.

## Research involving human participants, their data, or biological material

Policy information about studies with human participants or human data. See also policy information about sex, gender (identity/presentation), and sexual orientation and race, ethnicity and racism.

| | |
|---|---|
| Reporting on sex and gender | Not applicable |
| Reporting on race, ethnicity, or other socially relevant groupings | Not applicable |
| Population characteristics | Not applicable |
| Recruitment | Not applicable |
| Ethics oversight | Not applicable |

Note that full information on the approval of the study protocol must also be provided in the manuscript.

# Field-specific reporting

Please select the one below that is the best fit for your research. If you are not sure, read the appropriate sections before making your selection.

☒ Life sciences    ☐ Behavioural & social sciences    ☐ Ecological, evolutionary & environmental sciences

For a reference copy of the document with all sections, see nature.com/documents/nr-reporting-summary-flat.pdf

# Life sciences study design

All studies must disclose on these points even when the disclosure is negative.

| | |
|---|---|
| Sample size | Sample sizes were determined based on experimental feasibility and cost considerations |
| Data exclusions | No data was excluded |
| Replication | All experiments were replicated |
| Randomization | No randomization was performed  as the comparisons are between isogenic controls |
| Blinding | No blinding was performed as the comparisons are between isogenic controls |

# Reporting for specific materials, systems and methods

We require information from authors about some types of materials, experimental systems and methods used in many studies. Here, indicate whether each material, system or method listed is relevant to your study. If you are not sure if a list item applies to your research, read the appropriate section before selecting a response.

## Materials & experimental systems

| n/a | Involved in the study |
|---|---|
| ☒ ☐ | Antibodies |
| ☒ ☐ | Eukaryotic cell lines |
| ☒ ☐ | Palaeontology and archaeology |
| ☐ ☒ | Animals and other organisms |
| ☒ ☐ | Clinical data |
| ☒ ☐ | Dual use research of concern |
| ☒ ☐ | Plants |

## Methods

| n/a | Involved in the study |
|---|---|
| ☒ ☐ | ChIP-seq |
| ☒ ☐ | Flow cytometry |
| ☒ ☐ | MRI-based neuroimaging |

# Animals and other research organisms

Policy information about studies involving animals; ARRIVE guidelines recommended for reporting animal research, and Sex and Gender in Research

| Laboratory animals | BALB/c mice, female, 6 to 8 weeks old, National Cancer Institute<br>Humidity: 30-70%<br>Temperature: 68-79°F or 20-26°C<br>Lighting conditions: Lights on 0600; lights off: 1800 |
|---|---|
| Wild animals | Not applicable |
| Reporting on sex | Female only, animal model is of intra-vaginal infection |
| Field-collected samples | Not applicable |
| Ethics oversight | Uniformed Services University of the Health Sciences Institutional Animal Care and Use Committee |

Note that full information on the approval of the study protocol must also be provided in the manuscript.

# Plants

| Seed stocks | Not applicable |
|---|---|
| Novel plant genotypes | Not applicable |
| Authentication | Not applicable |

