## [Peer Review File · Nature Microbiology]

Quantifying the real-world impact of antibiotic use and genetic determinants of resistance on gonococcal dynamics

Corresponding Author: Professor Yonatan Grad

Version 0:

Decision Letter:

9th December 2024

Dear Yonatan,

Thank you very much for your enquiry about submitting your manuscript to Nature Microbiology. It certainly sounds interesting, and we would be happy to consider it for publication. However, I'm sure you'll understand that we cannot make a firm decision about whether to send the paper out to review until we have carefully read the full paper (and appropriate background literature).

In order to submit your complete manuscript to Nature Microbiology, please use the link below:

Link Redacted

If you have any questions, please feel free to contact me.

Version 1:

Reviewer comments:

Reviewer #1

(Remarks to the Author)

In the manuscript titled "Quantifying the impact of antibiotic use and genetic determinants of resistance on bacterial lineage dynamics" – the authors seek to address the real-world impacts of antimicrobial resistance in *Neisseria gonorrhoeae* on bacterial fitness. The authors do this using phylodynamic modeling to determine fitness contributions to lineage dynamics. This manuscript appears to establish a method linking pathogen genomics and antimicrobial resistance in a clinically relevant organism.

In regards to the in vitro and in vivo work, the data is presented well and accurately, and complements the stated conclusions derived from the modeling work. The methods chosen for genomic analysis, lineage assignment and phylogenetic reconstruction appear appropriate. My critiques lie mainly with accessibility/reproducibility of the code provided.

(Remarks on code availability)

I have minor critiques related to improving the accessibility of the data analysis to interested readership:

1. In regards, to the code provided on github – the code is clearly organized and runs well. Using the provided data, I was able to replicate all of the included figures. However, I would highly recommend that the author provide the version used for all of the R packages and dependencies used in the directory, which are not listed in the manuscript. Specifically, phylodyn, which is no longer available on CRAN, has multiple archived versions. I was able to use the code provided by downloading through package through github, rather than CRAN. This is essential information for replicating the figures.
2. In regard to the genomic analysis, lineage assignment, and phylogenetic reconstruction, the github repository would be strengthened by including command-line scripts used for upstream analysis, as well as a more complete computational environment description. Additionally, providing the project accession numbers for the sequenced GISP isolates (in addition to the cited manuscript currently present) would aid in replication of this analysis.
3. It would also be helpful if the authors provided information on the type of system used for this analysis (local vs HPC cluster), the operating system, and RAM/CPU specs, ect as many of these programs can be memory intensive.

Reviewer #2

(Remarks to the Author)

Thank you for asking me to review this paper by Helekal et al. It is an extremely well written manuscript, that integrates large-scale genomic data across nearly 30 years of *N. gonorrhoeae* evolution with well-considered molecular microbiological experiments and Bayesian phylodynamic modelling. The findings highlight that small genetic differences in resistance determinants (e.g., single amino acid changes) can produce large fitness effects, moving beyond simple resistant vs. susceptible frameworks. The authors support their modelling inferences with laboratory experiments, adding causal weight to the observations. The authors also link genomic lineage trends to documented treatment shifts, providing useful historical context for putative changes in lineage dynamics. This work builds on similar high quality work from this group, and adds considerably to the body of knowledge about NG resistance emergence and evolution.

Comments:

Introduction:

- Line 48: The term fitness is not explicitly defined. It is used interchangeably to describe both within-host and population-level effects. The authors should define explicitly what they mean by fitness, given that it is used repeatedly throughout the paper.
- Lines 61-70: the authors should consider mentioning other confounding pressures on AMR epidemiology, beyond bacteria factors and antibiotic use. This is particularly relevant for *N. gonorrhoeae*, and they include things such as host sexual network structure, social determinants (e.g., syndemics; access to testing and treatment; diagnostic pressure; travel).

Material and Methods

- The genomic analysis is pretty standard and robust.
- Lines 810-812: What was the justification for the 80% LOF cutoff?
- Lines 841-842: What was the justification for including at least 30 tips? Would this exclude those emerging but epidemiologically important lineages or those with recombination-induced diversity in resistance loci?
- Lines 904-906: I wasn't quite clear here – I think the authors are defining lineages as monophyletic groups with stable AMR determinants and assuming no within-lineage recombination. This may not be entirely valid for *N. gonorrhoeae*.
- Line 1119-1120: Do the authors know if introduction of the kanamycin cassette introduces a fitness cost?
- Did the authors check for introduction of secondary mutations during the in vitro and in vivo work?

Results

- Line 114: I suggest modifying this to also note that the effect of sexual networks (composition; behaviours within networks) has an impact on lineages structure.
- Lines 157-158: Do the authors have any epidemiological / demographic information on patients in lineage 21 to help explain its dynamics? Were there mutations elsewhere in the genome that may explain its emergence?
- Lines 153-155: The authors make an implicit assumption that changes in treatment guidelines temporally align with changes in $N_e(t)$, which is probably OK, but in practice, changes to guidelines are often delayed or inconsistent across different settings.
- Lines 188-203: gyrA – did the GyrA 91F/95A lineages that expanded after FQs were no longer recommended have any other mutations that could encourage co-selection?
- Lines 229-230: Why did the authors only take the competition experiments out to 8 hours? Ideally it would be good to do longer term passage; this may also reveal compensatory mutations.
- Lines 228-232: The differential fitness of 91F/95A and 91F/95G should be addressed; although the mutations confer similar high MICs, their fitness difference may indicate non-equivalence in biological cost or compensation. Again, it would be important to confirm there are no secondary mutations that have been introduced during the mutagenesis process which may contribute to observed phenotype.
- Lines 371-372: I note the authors excluded tetM because of limited treatment data, but this omits an important contributor to the success of Lineage 2.
- Lines 366-367: The choice of a 10% threshold for identifying meaningful residual deviations should be justified or sensitivity-tested, as it shapes the number of lineages flagged as unexplained.

Discussion

General points

- The Discussion underplays other ecological or epidemiological factors (e.g. sexual networks, host immunity, transmission dynamics, diagnostic practices) that may have shaped lineage success. In addition, use of antibiotics for other indications (which this team has previously explored) has not been mentioned – this could obviously impact indirectly on NG lineage structure.
- Some statements imply causal links between antibiotic policies and lineage dynamics, but these are based on temporal correlations without direct causal tests. I think the tone could be slightly modified to frame treatment-lineage relationships as associations or correlations, unless there is direct mechanistic evidence otherwise.
- My major comment on the Discussion is that it wasn't clear how the findings might inform AMR surveillance, treatment guideline development, or targeted interventions. A few sentences in the closing paragraphs illustrating the direct applicability of this work would greatly help readers fully understand its utility and potential impact.

(Remarks on code availability)

I read through the phylodynamic models in detail. While this is not my area of expertise, the rationale for use of the phylodynamic

framework seemed methodologically sound.

Reviewer #3

(Remarks to the Author)

The manuscript "Quantifying the impact of antibiotic use and genetic determinants of resistance on bacterial lineage dynamics" studies the contributions of resistance determinants to strain success in a large collection of *Neisseria gonorrhoeae* isolates. The manuscript is well written and is of great interest to the field, as it explores the fitness effects of different AMR-conferring mutations and validates some of the candidate mutations with experiments.

I was asked to assess the mathematical modelling and phylogenetics-related aspects of the study, so I will focus my review on those points.

Main remarks

I find the approach of the authors – modelling the effective population size of the different AMR clades – clever and of interest for the field. However, I have a few concerns:

- My main concern is the fact that they first reconstruct a time-resolved phylogeny using Bactdating, which assumes a constant effective population size, and then model the changes in effective population size on this tree, for particular clades. To me, these two modelling assumptions contradict each other. Have the authors tested how these model choices impact their analysis? If the initial time-resolved phylogeny was reconstructed using a varying effective population size, would the results change? Using a skygrid/skyline model in the time-tree reconstruction step would then not contradict the modelling of the changes in effective population size in the second part of the analysis. I realise that skygrid/skyline models are not implemented in Bactdating at present, but this point should be at the very least discussed in depth as a limitation of the study.

- The authors use a two-step approach to perform their analysis (tree reconstruction and then N_e modelling). I might have missed it, but did the authors consider the uncertainty in the reconstructed timed phylogeny from Bactdating? (i.e. the posterior of trees generated by Bactdating) In Bactdating, the tree topology is fixed, but the branch lengths can have wide credible intervals.

- The authors base their study on a bacterial time-resolved phylogeny, yet they do not present the clock signal (root-to-tip regression) or the substitution rate estimated during the tree reconstruction. This is important to assess the reliability of the phylogenetic signal and compare the tree obtained to the existing literature on *N. gonorrhoeae*.

- The hierarchical phylodynamic model developed by the authors is highly parameterized, taking into account "multiple lineages, multiple pathways, and multiple time-varying covariates". While it is highly valuable to be able to build such a model to test hypotheses, I wonder what a model comparison analysis, removing in turn some parameters (genetic markers/antibiotics), would yield. Are all the parameters critical to explain the N_e changes of the multiple lineages? How much does the model currently explain the fitness dynamics?

Other remarks:

- The dataset comprises isolates from 2000 to 2019, but most figures show dynamics estimated before this – can the author discuss why they chose this? For clarity, it would be good to add a mark on the figures showing for which time period there is sequence data.

- Figure 1: what are the colors indicating?

- Figure 2A: The legend indicates "The grey transparent tips correspond to isolates that have diverged from the ancestral determinant combination of the parental lineage.". It would be good to mention what is the ancestral determinant combination to increase the readability of the figure.

- Line 817: "4; Less than 12% of positions were missing in pseudogenomes", how did the authors come up with this threshold?

(Remarks on code availability)

I successfully ran the code provided on the GitHub repository. I did not go through all the codes, but only `run_model.R` file, and the `stan` model briefly. I was able to reproduce the main figures. It is nicely organised and easy to run.

However, the code is sparsely commented, which can make it hard to use for future applications on other datasets. It would be of particular interest to have a description of the data format used in the model. Additionally, it would be good to have a clear list of packages and versions that are needed to run the code.

Reviewer #4

(Remarks to the Author)

This is a wonderful, though dense, piece of work. It is exactly what is needed for the field of AMR and microbial population dynamics. The authors have done an incredible amount of work to integrate laboratory phenotypes with public health scale population dynamic modelling, leveraging genomic surveillance data to guide exquisitely targeted confirmatory experimentation for better understanding AMR and the influence of genetic context (in a WHO AMR priority pathogen). Moreover, they have made an admirably succinct narration for what is a very complex, interdisciplinary study. I have several probing questions below which mostly reflect my enthusiasm for the manuscript and subject area, rather than major criticisms that detract from the overall quality of the work. However, I do feel some require further explanation (and potentially further analyses) could improve the manuscript.

Line 51 – "neutral or deleterious" perhaps – deleterious assumes there is still a fitness cost associated with AMR carriage (which is often ameliorated under sustained selection)

Line 78 – "demographics of the infected individuals" Any info on ab use by those individuals?

Figure 1. I feel like this could be more informative for evaluating the study - perhaps truncating to 1980 to allow better visualisation of the relevant time period and room to show uncertainty (in both topology and ASR node support) in the tree structure and maybe some raw AMR data (not sure why this was separated into Table 1 and hard to assess how well the ASR has worked for lineage assignment as is). It's hard to get a feel for co-existence/circulation of the lineages in its current form (since presumably real world competition dynamics are what we are trying to capture/test in the lab here).

Line 131 – are the phenotypes the same?

e.g. in Table S3 – resistances are presented as ≥ 32 – is there a higher MIC for one set of mutations, was this information available? Or were absolute MIC laboratory phenotypes at least determined and compared for the competing strains? [and same question for other antimicrobial class phenotypes and allelic variants]

Table 1- maybe add the corresponding antimicrobial classes for the uninitiated?

Figure 2 is compelling, but what does this look like across the tree? If they are partitioned into resistant and non-resistant (or determinant-containing vs not) irrespective of lineage? Do the population dynamics shown in this select set of sub-lineages hold?

It's unclear why further detail (rightmost bars) are only provided for Lineage 20?

Figure 3 - The modelling results are borderline on their own with wide uncertainty and little convincing variation from the baseline, but these estimates have natural limitations borne from the actual evolutionary dynamics and polyphyly of phenotypes in the bacterial populations. The results are rescued convincingly by the juxtaposed laboratory data in panel B which confirms the relative fitness benefits of these alleles in vitro.

Figure 4 – This decrease in fitness appears to be mostly driven by the descent of lineage 22 (which is most of the mosaic allele containing isolates according to Table 1 and Figure 1 – e.g. Lineage 23 appears relatively small) Is there anything to be learned from the Lineage 23 dynamics that would support this? And the descent of Lineage 22 is really a clonal replacement (of 22 by 21) at the time of treatment recommendation change. So, is the 'mysterious' expansion of Lineage 21 now meant to be explained by the decreased fitness of the mosaic penA allele? Again, these are limitations of the natural dynamics, but it needs exploring – why no head-to-head laboratory competition for these alleles?

Figure 5 – This is a nice negative control for confirming the observed changes are the influence of fluctuating treatment guidance, but it's not explicit why this progressed to mouse experiments whereas the fluoroquinolone determinants did not. Is this because of the negative in vitro growth results? And why wasn't it observed in vitro? Just a time effect? Could it be influenced by different starting ratios? it would have been nice to see the contrast with the gyrA/parC work discussed more e.g. with Line 433 – 442.

Figure 6 – given the need to collapse AZM genetic determinants, I wonder whether the model have more signal if different phenotype bins (e.g. bounds of MIC value) were compared here. I'm unsure of the underlying MIC data distribution, and of course the relative influence of genotype and phenotype is not the main focus of the paper, but it would be interesting. Also perhaps specify 'co-treatment' with what for the AZM in legend

Figure 7 – again Abx class (both for years of rec and corresponding genes) could be added to the Table for clarity/readability. It's not intuitively clear from the text what AMR total is – is this having phenotypic resistance against the recommended treatment? Does the tailing off of a fitness benefit suggest there are other resistance alleles emerging? Or is this the influence of tetracycline use and non-target abxs?

L393 – 416 – what about the possibility of different (non-binary) resistance phenotypes?

L428 – again, this is potentially driven by clonal replacement perhaps consider/discuss differences in cefixime/ceftriaxone phenotypes for the different alleles here

L433- L442 – again, as above, some discussion on why this effect wasn't detectable in vitro would add value

L466 – 'incorporating population-wide antimicrobial use would ...' (I assume individual level use data was not associated with this surveillance program?)

L477 – 'reducing the uncertainty' ... the inherent bacterial population dynamics may also be a natural limiter here (see above) e.g. lack of polyphyly for some elements, etc

L494 – re: ignoring spatial separation – for at least one analysis transmission network (i.e. MSM or MSW) was considered and could act as a proxy for spatial separation throughout – why was this not more systematically considered across lineages? Also, is there much/anything known about the spatial structure of gonno in the US that would support ignoring this?

(Remarks on code availability)

I have had a look though and it appears comprehensive, is well commented, and having read through the methods, much of it is drawn from previous manuscripts/analyses so I'm confident of reproducibility here.

Decision Letter:

7th May 2025

Dear Yonatan,

Thank you for your patience while your manuscript "Quantifying the impact of antibiotic use and genetic determinants of resistance on bacterial lineage dynamics" was under peer-review at Nature Microbiology. It has now been seen by 4 referees, whose expertise and comments you will find at the end of this email. Although they find your work of some potential interest, they have raised a number of concerns that will need to be addressed before we can consider publication of the work in Nature Microbiology.

In particular, referee #1 has some suggestions on how to improve accessibility/reproducibility of the code. Referee #2 asks to provide more clinical and public health context for the findings, and acknowledge the likely contribution of non-AMR associated factors to shaping *N. gonorrhoeae* lineages. This referee also says it would be important to confirm that there are no secondary mutations that have been introduced during the mutagenesis process. Referee #3 has some concerns regarding the reliance of the paper on a timed phylogeny which was built using different assumption than the one used in the model. The referee says that the timed-tree should ideally be reconstructed again using a varying N_e , but also states that given this option is not available in the program used (Bactdating), it might not be computationally feasible. Nonetheless, as the referee suggest, it should be presented as a limitation of the study. Referee #4 says that Figure 2 is compelling, but asks what this looks like across the tree. Further, regarding Figure 6, the referee asks whether the model would have more signal if different phenotype bins (e.g. bounds of MIC value) were compared here.

Should further experimental data allow you to address these criticisms, we would be happy to look at a revised manuscript.

Please include a data availability statement as a separate section after Methods but before references, under the heading "Data Availability". This section should inform readers about the availability of the data used to support the conclusions of your study. This information includes accession codes to public repositories (data banks for protein, DNA or RNA sequences, microarray, proteomics data etc...), references to source data published alongside the paper, unique identifiers such as URLs to data repository entries, or data set DOIs, and any other statement about data availability. At a minimum, you should include the following statement: "The data that support the findings of this study are available from the corresponding author upon request", mentioning any restrictions on availability. If DOIs are provided, we also strongly encourage including these in the Reference list (authors, title, publisher (repository name), identifier, year). For more guidance on how to write this section please see: <http://www.nature.com/authors/policies/data/data-availability-statements-data-citations.pdf>

* If you have not done so already we suggest that you begin to revise your manuscript so that it conforms to our Article format instructions at <http://www.nature.com/nmicrobiol/info/final-submission>. Refer also to any guidelines provided in this letter.

When submitting the revised version of your manuscript, please pay close attention to our [href="https://www.nature.com/nature-portfolio/editorial-policies/image-integrity">Digital Image Integrity Guidelines](https://www.nature.com/nature-portfolio/editorial-policies/image-integrity) and to the following points below:

EXTENDED DATA FIGURES

Link Redacted

Note: This url links to your confidential homepage and associated information about manuscripts you may have submitted or be reviewing for us. If you wish to forward this e-mail to co-authors, please delete this link to your homepage first.

Nature Microbiology is committed to improving transparency in authorship. As part of our efforts in this direction, we are now requesting that all authors identified as 'corresponding author' on published papers create and link their Open Researcher and Contributor Identifier (ORCID) with their account on the Manuscript Tracking System (MTS), prior to acceptance. This applies to primary research papers only. ORCID helps the scientific community achieve unambiguous attribution of all scholarly contributions. You can create and link your ORCID from the home page of the MTS by clicking on 'Modify my Springer Nature account'. For more information please visit www.springernature.com/orcid.

If you wish to submit a suitably revised manuscript we would hope to receive it within 3 months. If you cannot send it within this time, please let us know.

Yours sincerely,

Reviewer Expertise:

Referee #1: Neisseria gonorrhoeae, AMR
Referee #2: AMR, population genomics
Referee #3: Mathematical modelling, phylodynamics
Referee #4: AMR, population genomics

Reviewer Comments:

Reviewer #1 (Remarks to the Author):

In the manuscript titled "Quantifying the impact of antibiotic use and genetic determinants of resistance on bacterial lineage dynamics" – the authors seek to address the real-world impacts of antimicrobial resistance in *Neisseria gonorrhoeae* on bacterial fitness. The authors do this using phylodynamic modeling to determine fitness contributions to lineage dynamics. This manuscript appears to establish a method linking pathogen genomics and antimicrobial resistance in a clinically relevant organism.

In regards to the in vitro and in vivo work, the data is presented well and accurately, and complements the stated conclusions derived from the modeling work. The methods chosen for genomic analysis, lineage assignment and phylogenetic reconstruction appear appropriate. My critiques lie mainly with accessibility/reproducibility of the code provided.

Reviewer #1 (Remarks on code availability):

I have minor critiques related to improving the accessibility of the data analysis to interested readership:
1. In regards, to the code provided on github – the code is clearly organized and runs well. Using the provided data, I was able to replicate all of the included figures. However, I would highly recommend that the author provide the version used for all of the R packages and dependencies used in the directory, which are not listed in the manuscript. Specifically, phylodyn, which is no longer available on CRAN, has multiple archived versions. I was able to use the code provided by downloading through package through github, rather than CRAN. This is essential information for replicating the figures.

2. In regard to the genomic analysis, lineage assignment, and phylogenetic reconstruction, the github repository would be strengthened by including command-line scripts used for upstream analysis, as well as a more complete computational environment description. Additionally, providing the project accession numbers for the sequenced GISP isolates (in addition to the cited manuscript currently present) would aid in replication of this analysis.
3. It would also be helpful if the authors provided information on the type of system used for this analysis (local vs HPC cluster), the operating system, and RAM/CPU specs, ect as many of these programs can be memory intensive.

Reviewer #2 (Remarks to the Author):

Thank you for asking me to review this paper by Helekal et al. It is an extremely well written manuscript, that integrates large-scale genomic data across nearly 30 years of *N. gonorrhoeae* evolution with well-considered molecular microbiological experiments and Bayesian phylodynamic modelling. The findings highlight that small genetic differences in resistance determinants (e.g., single amino acid changes) can produce large fitness effects, moving beyond simple resistant vs. susceptible frameworks. The authors support their modelling inferences with laboratory experiments, adding causal weight to the observations. The authors also link genomic lineage trends to documented treatment shifts, providing useful historical context for putative changes in lineage dynamics. This work builds on similar high quality work from this group, and adds considerably to the body of knowledge about NG resistance emergence and evolution.

Comments:

Introduction:

- Line 48: The term fitness is not explicitly defined. It is used interchangeably to describe both within-host and population-level effects. The authors should define explicitly what they mean by fitness, given that it is used repeatedly throughout the paper.
- Lines 61-70: the authors should consider mentioning other confounding pressures on AMR epidemiology, beyond bacteraemia factors and antibiotic use. This is particularly relevant for *N. gonorrhoeae*, and they include things such as host sexual network structure, social determinants (e.g., syndemics; access to testing and treatment; diagnostic pressure; travel).

Material and Methods

- The genomic analysis is pretty standard and robust.
- Lines 810-812: What was the justification for the 80% LOF cutoff?
- Lines 841-842: What was the justification for including at least 30 tips? Would this exclude those emerging but epidemiologically important lineages or those with recombination-induced diversity in resistance loci?
- Lines 904-906: I wasn't quite clear here – I think the authors are defining lineages as monophyletic groups with stable AMR determinants and assuming no within-lineage recombination. This may not be entirely valid for *N. gonorrhoeae*.
- Line 1119-1120: Do the authors know if introduction of the kanamycin cassette introduces a fitness cost?
- Did the authors check for introduction of secondary mutations during the in vitro and in vivo work?

Results

- Line 114: I suggest modifying this to also note that the effect of sexual networks (composition; behaviours within networks) has an impact on lineages structure.
- Lines 157-158: Do the authors have any epidemiological / demographic information on patients in lineage 21 to help explain its dynamics? Were there mutations elsewhere in the genome that may explain its emergence?
- Lines 153-155: The authors make an implicit assumption that changes in treatment guidelines temporally align with changes in $N_e(t)$, which is probably OK, but in practice, changes to guidelines are often delayed or inconsistent across different settings.
- Lines 188-203: gyrA – did the GyrA 91F/95A lineages that expanded after FQs were no longer recommended have any other mutations that could encourage co-selection?
- Lines 229-230: Why did the authors only take the competition experiments out to 8 hours? Ideally it would be good to do longer term passage; this may also reveal compensatory mutations.
- Lines 228-232: The differential fitness of 91F/95A and 91F/95G should be addressed; although the mutations confer similar high MICs, their fitness difference may indicate non-equivalence in biological cost or compensation. Again, it would be important to confirm there are no secondary mutations that have been introduced during the mutagenesis process which may contribute to observed phenotype.
- Lines 371-372: I note the authors excluded tetM because of limited treatment data, but this omits an important contributor to the success of Lineage 2.
- Lines 366-367: The choice of a 10% threshold for identifying meaningful residual deviations should be justified or sensitivity-tested, as it shapes the number of lineages flagged as unexplained.

Discussion

General points

- The Discussion underplays other ecological or epidemiological factors (e.g. sexual networks, host immunity, transmission dynamics, diagnostic practices) that may have shaped lineage success. In addition, use of antibiotics for other indications (which this team has previously explored) has not been mentioned – this could obviously impact indirectly on NG lineage structure.
- Some statements imply causal links between antibiotic policies and lineage dynamics, but these are based on temporal correlations without direct causal tests. I think the tone could be slightly modified to frame treatment-lineage relationships as associations or correlations, unless there is direct mechanistic evidence otherwise.

- My major comment on the Discussion is that it wasn't clear how the findings might inform AMR surveillance, treatment guideline development, or targeted interventions. A few sentences in the closing paragraphs illustrating the direct applicability of this work would greatly help readers fully understand its utility and potential impact.

Reviewer #2 (Remarks on code availability):

I read through the phylodynamic models in detail. While this is not my area of expertise, the rationale for use of the phylodynamic framework seemed methodologically sound.

Reviewer #3 (Remarks to the Author):

The manuscript "Quantifying the impact of antibiotic use and genetic determinants of resistance on bacterial lineage dynamics" studies the contributions of resistance determinants to strain success in a large collection of *Neisseria gonorrhoeae* isolates. The manuscript is well written and is of great interest to the field, as it explores the fitness effects of different AMR-conferring mutations and validates some of the candidate mutations with experiments.

I was asked to assess the mathematical modelling and phylogenetics-related aspects of the study, so I will focus my review on those points.

Main remarks

I find the approach of the authors – modelling the effective population size of the different AMR clades – clever and of interest for the field. However, I have a few concerns:

- My main concern is the fact that they first reconstruct a time-resolved phylogeny using Bactdating, which assumes a constant effective population size, and then model the changes in effective population size on this tree, for particular clades. To me, these two modelling assumptions contradict each other. Have the authors tested how these model choices impact their analysis? If the initial time-resolved phylogeny was reconstructed using a varying effective population size, would the results change? Using a skygrid/skyline model in the time-tree reconstruction step would then not contradict the modelling of the changes in effective population size in the second part of the analysis. I realise that skygrid/skyline models are not implemented in Bactdating at present, but this point should be at the very least discussed in depth as a limitation of the study.
- The authors use a two-step approach to perform their analysis (tree reconstruction and then N_e modelling). I might have missed it, but did the authors consider the uncertainty in the reconstructed timed phylogeny from Bactdating? (i.e. the posterior of trees generated by Bactdating) In Bactdating, the tree topology is fixed, but the branch lengths can have wide credible intervals.
- The authors base their study on a bacterial time-resolved phylogeny, yet they do not present the clock signal (root-to-tip regression) or the substitution rate estimated during the tree reconstruction. This is important to assess the reliability of the phylogenetic signal and compare the tree obtained to the existing literature on *N. gonorrhoeae*.
- The hierarchical phylodynamic model developed by the authors is highly parameterized, taking into account "multiple lineages, multiple pathways, and multiple time-varying covariates". While it is highly valuable to be able to build such a model to test hypotheses, I wonder what a model comparison analysis, removing in turn some parameters (genetic markers/antibiotics), would yield. Are all the parameters critical to explain the N_e changes of the multiple lineages? How much does the model currently explain the fitness dynamics?

Other remarks:

- The dataset comprises isolates from 2000 to 2019, but most figures show dynamics estimated before this – can the author discuss why they chose this? For clarity, it would be good to add a mark on the figures showing for which time period there is sequence data.
- Figure 1: what are the colors indicating?
- Figure 2A: The legend indicates "The grey transparent tips correspond to isolates that have diverged from the ancestral determinant combination of the parental lineage." It would be good to mention what is the ancestral determinant combination to increase the readability of the figure.
- Line 817: "4; Less than 12% of positions were missing in pseudogenomes", how did the authors come up with this threshold?

Reviewer #3 (Remarks on code availability):

I successfully ran the code provided on the GitHub repository. I did not go through all the codes, but only run_model.R file, and the stan model briefly. I was able to reproduce the main figures. It is nicely organised and easy to run.

However, the code is sparsely commented, which can make it hard to use for future applications on other datasets. It would be of particular interest to have a description of the data format used in the model. Additionally, it would be good to have a clear list of packages and versions that are needed to run the code.

Reviewer #4 (Remarks to the Author):

This is a wonderful, though dense, piece of work. It is exactly what is needed for the field of AMR and microbial population dynamics. The authors have done an incredible amount of work to integrate laboratory phenotypes with public health scale population dynamic modelling, leveraging genomic surveillance data to guide exquisitely targeted confirmatory experimentation

for better understanding AMR and the influence of genetic context (in a WHO AMR priority pathogen). Moreover, they have made an admirably succinct narration for what is a very complex, interdisciplinary study. I have several probing questions below which mostly reflect my enthusiasm for the manuscript and subject area, rather than major criticisms that detract from the overall quality of the work. However, I do feel some require further explanation (and potentially further analyses) could improve the manuscript.

Line 51 – “neutral or deleterious” perhaps – deleterious assumes there is still a fitness cost associated with AMR carriage (which is often ameliorated under sustained selection)

Line 78 – “demographics of the infected individuals” Any info on ab use by those individuals?

Figure 1. I feel like this could be more informative for evaluating the study - perhaps truncating to 1980 to allow better visualisation of the relevant time period and room to show uncertainty (in both topology and ASR node support) in the tree structure and maybe some raw AMR data (not sure why this was separated into Table 1 and hard to assess how well the ASR has worked for lineage assignment as is). It's hard to get a feel for co-existence/circulation of the lineages in its current form (since presumably real world competition dynamics are what we are trying to capture/test in the lab here).

Line 131 – are the phenotypes the same?

e.g. in Table S3 – resistances are presented as ≥ 32 – is there a higher MIC for one set of mutations, was this information available? Or were absolute MIC laboratory phenotypes at least determined and compared for the competing strains? [and same question for other antimicrobial class phenotypes and allelic variants]

Table 1- maybe add the corresponding antimicrobial classes for the uninitiated?

Figure 2 is compelling, but what does this look like across the tree? If they are partitioned into resistant and non-resistant (or determinant-containing vs not) irrespective of lineage? Do the population dynamics shown in this select set of sub-lineages hold?

It's unclear why further detail (rightmost bars) are only provided for Lineage 20?

Figure 3 - The modelling results are borderline on their own with wide uncertainty and little convincing variation from the baseline, but these estimates have natural limitations borne from the actual evolutionary dynamics and polyphyly of phenotypes in the bacterial populations. The results are rescued convincingly by the juxtaposed laboratory data in panel B which confirms the relative fitness benefits of these alleles in vitro.

Figure 4 – This decrease in fitness appears to be mostly driven by the descent of lineage 22 (which is most of the mosaic allele containing isolates according to Table 1 and Figure 1 – e.g. Lineage 23 appears relatively small) Is there anything to be learned from the Lineage 23 dynamics that would support this? And the descent of Lineage 22 is really a clonal replacement (of 22 by 21) at the time of treatment recommendation change. So, is the ‘mysterious’ expansion of Lineage 21 now meant to be explained by the decreased fitness of the mosaic penA allele? Again, these are limitations of the natural dynamics, but it needs exploring – why no head-to-head laboratory competition for these alleles?

Figure 5 – This is a nice negative control for confirming the observed changes are the influence of fluctuating treatment guidance, but it's not explicit why this progressed to mouse experiments whereas the fluoroquinolone determinants did not. Is this because of the negative in vitro growth results? And why wasn't it observed in vitro? Just a time effect? Could it be influenced by different starting ratios? it would have been nice to see the contrast with the gyrA/parC work discussed more e.g. with Line 433 – 442.

Figure 6 – given the need to collapse AZM genetic determinants, I wonder whether the model have more signal if different phenotype bins (e.g. bounds of MIC value) were compared here. I'm unsure of the underlying MIC data distribution, and of course the relative influence of genotype and phenotype is not the main focus of the paper, but it would be interesting. Also perhaps specify ‘co-treatment’ with what for the AZM in legend

Figure 7 – again Abx class (both for years of rec and corresponding genes) could be added to the Table for clarity/readability. It's not intuitively clear from the text what AMR total is – is this having phenotypic resistance against the recommended treatment? Does the tailing off of a fitness benefit suggest there are other resistance alleles emerging? Or is this the influence of tetracycline use and non-target abxs?

L393 – 416 – what about the possibility of different (non-binary) resistance phenotypes?

L428 – again, this is potentially driven by clonal replacement perhaps consider/discuss differences in cefixime/ceftriaxone phenotypes for the different alleles here

L433- L442 – again, as above, some discussion on why this effect wasn't detectable in vitro would add value

L466 – ‘incorporating population-wide antimicrobial use would ...’ (I assume individual level use data was not associated with this surveillance program?)

L477 – ‘reducing the uncertainty’ ... the inherent bacterial population dynamics may also be a natural limiter here (see above) e.g. lack of polyphyly for some elements, etc

L494 – re: ignoring spatial separation – for at least one analysis transmission network (i.e. MSM or MSW) was considered and could act as a proxy for spatial separation throughout – why was this not more systematically considered across lineages? Also, is there much/anything known about the spatial structure of gonno in the US that would support ignoring this?

Reviewer #4 (Remarks on code availability):

I have had a look though and it appears comprehensive, is well commented, and having read through the methods, much of it is drawn from previous manuscripts/analyses so I'm confident or reproducibility here.

Version 2:

Reviewer comments:

Reviewer #1

(Remarks to the Author)

In general the authors have thoroughly and thoughtfully addressed the reviewer comments. The new Snakemake pipeline on github is well documented and will be extremely helpful for reproducibility. My only remaining minor comment, is that although the authors now specify where to retrieve packages and dependencies, they still do not specify the version used for each. As both Reviewer 1 and 3 requested this, I think this should be addressed as it is important for reproducibility.

(Remarks on code availability)

The code is well documented and runs well. I was able to reproduce the figures.

Reviewer #3

(Remarks to the Author)

Thank you for your careful consideration of the reviewers' remarks, including mine. The changes made to the manuscript add precision to both the results and the discussion. It remains a great, well-written study and is of large interest to the field.

Last two remarks:

- I agree with the authors that two-step approaches, while not ideal, are currently necessary when considering large datasets such as theirs. Thank you for adding this point in the discussion. However, given that the authors' results focus on the timings of fitness changes with respect to antibiotic usage/implementation, it would be good to have special care with the use of the "one" timed tree.

Some examples from the text: "Lineage 21 expanded after the 2010 switch" Line 172, "with clear support for a growth rate benefit only in 2010-2011" : Line 348

It is unclear how much the results from a single timed tree reflects what would be obtained from the full BactDating posterior (which already does not take into account topology uncertainty). I understand that using the full posterior of trees is tricky; however, the authors could replicate their analysis on a few trees to analyse how consistent their results are. A difference of a few years in some node dating could impact their conclusions. I would also encourage the authors to plot a phylogeny with confidence intervals on node dates in the supplementary material.

- Clock signal: Thank you for providing the estimates of the substitution rate and root time. I note that the reference added [PMID 31358980] estimated $3.74E-06$ [$3.39E-06 - 4.07E-06$] substitutions/site/year, which is ~20% lower than the one they estimated. Such a difference could be due to the fact that the datasets are different. As in the article the authors cited, it would be good to present a root-to-tip regression to show how much signal there is in the authors' dataset (see Fig S2 in 31358980). Again, as there is uncertainty in the estimated rate, it would be sensible to analyse/provide a sense of the impact of these on the results (see point above).

(Remarks on code availability)

Reviewer #4

(Remarks to the Author)

Thank you to the authors for providing comprehensive responses to my queries.

(Remarks on code availability)

Decision Letter:

22nd September 2025

Dear Yonatan,

Thank you for your patience while your manuscript "Quantifying the impact of antibiotic use and genetic determinants of resistance on bacterial lineage dynamics" was under peer-review at Nature Microbiology. It has now been seen by 3 of the 4 previous referees (referees #1, #3 and #4), whose expertise and comments you will find at the end of this email. You will see from their comments below that while they find your work of interest, some important points are raised. We are very interested in the possibility of publishing your study in Nature Microbiology, but would like to consider your response to these concerns in the form of a revised manuscript before we make a final decision on publication.

In particular, you will see that referee #3 still has a couple of concerns regarding the phylogenetic analysis and suggests some additional analyses. The rest of the referees' reports are clear and the remaining issues should be straightforward to address.

If you have not done so already please begin to revise your manuscript so that it conforms to our Article format instructions at <http://www.nature.com/nmicrobiol/info/final-submission/>

The usual length limit for a Nature Microbiology Article is six display items (figures or tables) and 4,000 words. We have some flexibility, and can allow a revised manuscript at 4,500 words, but please consider this a firm upper limit. There is a trade-off of ~250 words per display item, so if you need more space, you could move a Figure or Table to Supplementary Information.

Some reduction could be achieved by focusing any introductory material and moving it to the start of your opening 'bold' paragraph, whose function is to outline the background to your work, describe in a sentence your new observations, and explain your main conclusions. The discussion should also be limited. Methods should be described in a separate section following the discussion, we do not place a word limit on Methods.

Nature Microbiology titles should give a sense of the main new findings of a manuscript, and should not contain punctuation. Please keep in mind that we strongly discourage active verbs in titles, and that they should ideally fit within 90 characters each (including spaces).

Please include a data availability statement as a separate section after Methods but before references, under the heading "Data Availability". This section should inform readers about the availability of the data used to support the conclusions of your study. This information includes accession codes to public repositories (data banks for protein, DNA or RNA sequences, microarray, proteomics data etc...), references to source data published alongside the paper, unique identifiers such as URLs to data repository entries, or data set DOIs, and any other statement about data availability. At a minimum, you should include the following statement: "The data that support the findings of this study are available from the corresponding author upon request", mentioning any restrictions on availability. If DOIs are provided, we also strongly encourage including these in the Reference list (authors, title, publisher (repository name), identifier, year). For more guidance on how to write this section please see: <http://www.nature.com/authors/policies/data/data-availability-statements-data-citations.pdf>

To improve the accessibility of your paper to readers from other research areas, please pay particular attention to the wording of the paper's opening bold paragraph, which serves both as an introduction and as a brief, non-technical summary in about 150 words. If, however, you require one or two extra sentences to explain your work clearly, please include them even if the paragraph is over-length as a result. The opening paragraph should not contain references. Because scientists from other sub-disciplines will be interested in your results and their implications, it is important to explain essential but specialised terms concisely. We suggest you show your summary paragraph to colleagues in other fields to uncover any problematic concepts.

If your paper is accepted for publication, we will edit your display items electronically so they conform to our house style and will reproduce clearly in print. If necessary, we will re-size figures to fit single or double column width. If your figures contain several parts, the parts should form a neat rectangle when assembled. Choosing the right electronic format at this stage will speed up the processing of your paper and give the best possible results in print. We would like the figures to be supplied as vector files - EPS, PDF, AI or postscript (PS) file formats (not raster or bitmap files), preferably generated with vector-graphics software (Adobe Illustrator for example). Please try to ensure that all figures are non-flattened and fully editable. All images should be at least 300 dpi resolution (when figures are scaled to approximately the size that they are to be printed at) and in RGB colour format. Please do not submit Jpeg or flattened TIFF files. Please see also 'Guidelines for Electronic Submission of Figures' at the end of this letter for further detail.

Figure legends must provide a brief description of the figure and the symbols used, within 350 words, including definitions of any error bars employed in the figures.

When submitting the revised version of your manuscript, please pay close attention to our [href="https://www.nature.com/nature-](https://www.nature.com/nature-)

research/editorial-policies/image-integrity">Digital Image Integrity Guidelines. and to the following points below:

EXTENDED DATA FIGURES

Please include a statement before the acknowledgements naming the author to whom correspondence and requests for materials should be addressed.

Finally, we require authors to include a statement of their individual contributions to the paper -- such as experimental work, project planning, data analysis, etc. -- immediately after the acknowledgements. The statement should be short, and refer to authors by their initials. For details please see the Authorship section of our joint Editorial policies at http://www.nature.com/authors/editorial_policies/authorship.html

* include a point-by-point response to any editorial suggestions and to our referees. Please include your response to the editorial suggestions in your cover letter, and please upload your response to the referees as a separate document.

* ensure it complies with our format requirements for Letters as set out in our guide to authors at www.nature.com/nmicrobiol/info/gta/

* resubmit electronically if possible using the link below to access your home page:

Link Redacted

*This url links to your confidential homepage and associated information about manuscripts you may have submitted or be reviewing for us. If you wish to forward this e-mail to co-authors, please delete this link to your homepage first.

Please ensure that all correspondence is marked with your Nature Microbiology reference number in the subject line.

Nature Microbiology is committed to improving transparency in authorship. As part of our efforts in this direction, we are now requesting that all authors identified as 'corresponding author' on published papers create and link their Open Researcher and Contributor Identifier (ORCID) with their account on the Manuscript Tracking System (MTS), prior to acceptance. This applies to primary research papers only. ORCID helps the scientific community achieve unambiguous attribution of all scholarly contributions. You can create and link your ORCID from the home page of the MTS by clicking on 'Modify my Springer Nature account'. For more information please visit www.springernature.com/orcid.

We hope to receive your revised paper within three weeks. If you cannot send it within this time, please let us know.

Yours sincerely,

Reviewer Expertise:

Referee #1: Neisseria gonorrhoeae, AMR

Referee #3: Mathematical modelling, phylodynamics

Referee #4: AMR, population genomics

Reviewers Comments:

Reviewer #1 (Remarks to the Author):

In general the authors have thoroughly and thoughtfully addressed the reviewer comments. The new Snakemake pipeline on github is well documented and will be extremely helpful for reproducibility. My only remaining minor comment, is that although the authors now specify where to retrieve packages and dependencies, they still do not specify the version used for each. As both Reviewer 1 and 3 requested this, I think this should be addressed as it is important for reproducibility.

Reviewer #1 (Remarks on code availability):

The code is well documented and runs well. I was able to reproduce the figures.

Reviewer #3 (Remarks to the Author):

Thank you for your careful consideration of the reviewers' remarks, including mine. The changes made to the manuscript add precision to both the results and the discussion. It remains a great, well-written study and is of large interest to the field.

Last two remarks:

- I agree with the authors that two-step approaches, while not ideal, are currently necessary when considering large datasets such as theirs. Thank you for adding this point in the discussion. However, given that the authors' results focus on the timings of fitness changes with respect to antibiotic usage/implementation, it would be good to have special care with the use of the "one" timed tree.

Some examples from the text: "Lineage 21 expanded after the 2010 switch" Line 172, "with clear support for a growth rate benefit only in 2010-2011" : Line 348

It is unclear how much the results from a single timed tree reflects what would be obtained from the full BactDating posterior (which already does not take into account topology uncertainty). I understand that using the full posterior of trees is tricky; however, the authors could replicate their analysis on a few trees to analyse how consistent their results are. A difference of a few years in some node dating could impact their conclusions. I would also encourage the authors to plot a phylogeny with confidence intervals on node dates in the supplementary material.

- Clock signal: Thank you for providing the estimates of the substitution rate and root time. I note that the reference added [PMID 31358980] estimated $3.74E-06$ [$3.39E-06 - 4.07E-06$] substitutions/site/year, which is ~20% lower than the one they estimated. Such a difference could be due to the fact that the datasets are different. As in the article the authors cited, it would be good to present a root-to-tip regression to show how much signal there is in the authors' dataset (see Fig S2 in 31358980). Again, as there is uncertainty in the estimated rate, it would be sensible to analyse/provide a sense of the impact of these on the results (see point above).

Reviewer #4 (Remarks to the Author):

Thank you to the authors for providing comprehensive responses to my queries.

Version 3:

Reviewer comments:

Reviewer #3

(Remarks to the Author)

I thank the authors for carefully addressing my comments and for the detailed clarifications provided.

Although I agree that the use of a single fixed tree reflects a limitation of currently available methods, this should not preclude at least some exploration of tree uncertainty. I appreciate the inclusion of the new Figure S3 in the supplementary material and find the authors' reasoning and explanations satisfactory.

While there remains scope for further methodological improvement in future work, I believe the current version appropriately acknowledges the relevant limitations. I have no further comments.

(Remarks on code availability)

Decision Letter:

Our ref: NMICROBIOL-24113701C

30th October 2025

Dear Yonatan,

Thank you for submitting your revised manuscript "Quantifying the impact of antibiotic use and genetic determinants of resistance on bacterial lineage dynamics" (NMICROBIOL-24113701C). It has now been seen by the original referees (referee #3) and their comments are below. The reviewers find that the paper has improved in revision, and therefore we'll be happy in principle to publish it in Nature Microbiology, pending minor revisions to comply with our editorial and formatting guidelines.

Thank you again for your interest in Nature Microbiology. Please do not hesitate to contact me if you have any questions.

Sincerely,

Reviewer #3 (Remarks to the Author):

I thank the authors for carefully addressing my comments and for the detailed clarifications provided.

Although I agree that the use of a single fixed tree reflects a limitation of currently available methods, this should not preclude at least some exploration of tree uncertainty. I appreciate the inclusion of the new Figure S3 in the supplementary material and find the authors' reasoning and explanations satisfactory.

While there remains scope for further methodological improvement in future work, I believe the current version appropriately acknowledges the relevant limitations. I have no further comments.

Version 4:

Decision Letter:

8th December 2025

Dear Yonatan,

I am pleased to accept your Article "Quantifying the real-world impact of antibiotic use and genetic determinants of resistance on gonococcal dynamics" for publication in Nature Microbiology. Thank you for having chosen to submit your work to us and many congratulations.

After the grant of rights is completed, you will receive a link to your electronic proof via email with a request to make any corrections within 48 hours. If, when you receive your proof, you cannot meet this deadline, please inform us at rjsproduction@springernature.com immediately. You will not receive your proofs until the publishing agreement has been received through our system.

Due to the importance of these deadlines, we ask you please us know now whether you will be difficult to contact over the next

month. If this is the case, we ask you provide us with the contact information (email, phone and fax) of someone who will be able to check the proofs on your behalf, and who will be available to address any last-minute problems.

Authors may need to take specific actions to achieve compliance with funder and institutional open access mandates. If your research is supported by a funder that requires immediate open access (e.g. according to [Plan S principles](https://www.springernature.com/gp/open-science/plan-s-compliance) or the [NIH public access policy](https://www.springernature.com/gp/open-science/us-federal-agency-compliance)) then you should select the gold OA route, and we will direct you to the compliant route where possible. Because authors warrant under our subscription licensing terms that they haven't committed to licensing any version of their article under a licence inconsistent with the terms of our agreement – including the applicable embargo period – publication under the subscription model isn't suitable for authors whose funders require no embargo.

Congratulations once again and I look forward to seeing the article published.

With kind regards,

P.S. Click on the following link if you would like to recommend Nature Microbiology to your librarian <http://www.nature.com/subscriptions/recommend.html#forms>

** Visit the Springer Nature Editorial and Publishing website at http://editorial-jobs.springernature.com?utm_source=ejP_NMicro_email&utm_medium=ejP_NMicro_email&utm_campaign=ejp_NMicro for more information about our career opportunities. If you have any questions please click [here](mailto:editorial.publishing.jobs@springernature.com).

Editor

Thank you for your patience while your manuscript "Quantifying the impact of antibiotic use and genetic determinants of resistance on bacterial lineage dynamics" was under peer-review at Nature Microbiology. It has now been seen by 4 referees, whose expertise and comments you will find at the end of this email. Although they find your work of some potential interest, they have raised a number of concerns that will need to be addressed before we can consider publication of the work in Nature Microbiology.

In particular, referee #1 has some suggestions on how to improve accessibility/reproducibility of the code. Referee #2 asks to provide more clinical and public health context for the findings, and acknowledge the likely contribution of non-AMR associated factors to shaping *N. gonorrhoeae* lineages. This referee also says it would be important to confirm that there are no secondary mutations that have been introduced during the mutagenesis process. Referee #3 has some concerns regarding the reliance of the paper on a timed phylogeny which was built using different assumption than the one used in the model. The referee says that the timed-tree should ideally be reconstructed again using a varying N_e , but also states that given this option is not available in the program used (Bactdating), it might not be computationally feasible. Nonetheless, as the referee suggest, it should be presented as a limitation of the study. Referee #4 says that Figure 2 is compelling, but asks what this looks like across the tree. Further, regarding Figure 6, the referee asks whether the model would have more signal if different phenotype bins (e.g. bounds of MIC value) were compared here.

Reviewer Expertise:

Referee #1: *Neisseria gonorrhoeae*, AMR

Referee #2: AMR, population genomics

Referee #3: Mathematical modelling, phylodynamics

Referee #4: AMR, population genomics

Reviewer #1

(Remarks to the Author):

In the manuscript titled "Quantifying the impact of antibiotic use and genetic determinants of resistance on bacterial lineage dynamics" – the authors seek to address the real-world impacts of antimicrobial resistance in *Neisseria gonorrhoeae* on bacterial fitness. The authors do this using phylodynamic modeling to determine fitness contributions to lineage dynamics. This manuscript appears to establish a method linking pathogen genomics and antimicrobial resistance in a clinically relevant organism.

In regards to the in vitro and in vivo work, the data is presented well and accurately, and complements the stated conclusions derived from the modeling work. The methods chosen for genomic analysis, lineage assignment and phylogenetic reconstruction appear appropriate. My critiques lie mainly with accessibility/reproducibility of the code provided.

(Remarks on code availability):

I have minor critiques related to improving the accessibility of the data analysis to interested readership:

- 1. In regards, to the code provided on github – the code is clearly organized and runs well. Using the provided data, I was able to replicate all of the included figures. However, I would highly recommend that the author provide the version used for all of the R packages and dependencies used in the directory, which are not listed in the manuscript. Specifically, phylodyn, which is no longer available on CRAN, has multiple archived versions. I was able to use the code provided by downloading through**

package through github, rather than CRAN. This is essential information for replicating the figures.

We updated the README file within the github repository to list all the required dependencies and where they can be installed from.

- 2. In regard to the genomic analysis, lineage assignment, and phylogenetic reconstruction, the github repository would be strengthened by including command-line scripts used for upstream analysis, as well as a more complete computational environment description. Additionally, providing the project accession numbers for the sequenced GISP isolates (in addition to the cited manuscript currently present) would aid in replication of this analysis.**

The accession numbers are included in the metadata spreadsheet in the github repository. We reworded the data availability statement to make this clear (**Lines 943-946**): “The code and data necessary to reproduce the statistical analysis, along with the metadata for the isolates analyzed, are available at: https://github.com/gradlab/GC_AMR_Lineages. The accession numbers for the sequenced isolates used in this analysis are available as a part of the metadata table included in the repository.”

We also added a subdirectory called “snakemake_pipeline,” which contains the assembly and variant calling pipeline.

- 3. It would also be helpful if the authors provided information on the type of system used for this analysis (local vs HPC cluster), the operating system, and RAM/CPU specs, ect as many of these programs can be memory intensive.**

We added the following sentence to the methods section (**Lines 1250-1251**): “All chains terminated under two hours on a laptop equipped with an Apple M3 Pro CPU and 36GB of RAM.”

Reviewer #2
(Remarks to the Author):

Thank you for asking me to review this paper by Helekal et al. It is an extremely well written manuscript, that integrates large-scale genomic data across nearly 30 years of *N. gonorrhoeae* evolution with well-considered molecular microbiological experiments and Bayesian phylodynamic modelling. The findings highlight that small genetic differences in resistance determinants (e.g., single amino acid changes) can produce large fitness effects, moving beyond simple resistant vs. susceptible frameworks. The authors support their modelling inferences with laboratory experiments, adding causal weight to the observations. The authors also link genomic lineage trends to documented treatment shifts, providing useful historical context for putative changes in lineage dynamics. This work builds on similar high quality work from this group, and adds considerably to the body of knowledge about NG resistance emergence and evolution.

Comments:
Introduction:

- 1. Line 48: The term fitness is not explicitly defined. It is used interchangeably to describe both within-host and population-level effects. The authors should define explicitly what they mean by fitness, given that it is used repeatedly throughout the paper.**

Throughout the manuscript, the fitness advantage (or disadvantage) associated with the carriage of a specific genetic determinant refers to a larger increase or decrease in the prevalence of pathogens carrying the determinant than those that do not. We have reworded the first paragraph of the introduction to reflect this (**Lines 46-56**): “The prevalence of antimicrobial resistance (AMR) reflects competition in an ever-changing environment (1, 2). When an antibiotic is introduced into clinical use, the environment in which bacteria compete changes. Bacterial lineages that carry or acquire resistance to the newly introduced antibiotic are now better adapted to the new environment. This leads to an increase in the prevalence of resistance and therefore a fitness advantage associated with resistance. Once resistance to the antibiotic becomes sufficiently widespread, its use is often reduced in favor of another antibiotic for which resistance prevalence is low. This alters the fitness landscape, such that bacteria carrying alleles and genes that had previously conferred a fitness advantage in the context of the first antibiotic may now be less or equally fit compared to bacteria that do not carry such genes or alleles. As such, alleles and genes that had conferred increased fitness may become neutral or deleterious.”

We further clarified a few sentences throughout the manuscript to make it explicit when a fitness advantage refers to *in vitro* or *in vivo* conditions.

Lines 288-289: “Both GyrA 91F/95G and GyrA 91F/95A strains were less fit *in vitro* than the susceptible parental strain (figure S5).”

Lines 353-355: “, the penicillin susceptible FA19 strain that normally carries the wild-type PBP1 421L and the penicillin resistant FA6140 strain (34) that normally carries PBP1 421P, to test whether the isolates that carry PBP1 421P are less fit under experimental conditions.”

Lines 360-362: “The PBP1 421P allele incurred an *in vivo* fitness cost to both FA19 and FA6140 when in competition with the isogenic PBP 421L strain in a female murine infection model”

- 2. Lines 61-70: the authors should consider mentioning other confounding pressures on AMR epidemiology, beyond bacterial factors and antibiotic use. This is particularly relevant for *N. gonorrhoeae*, and they should include things such as host sexual network structure, social determinants (e.g., syndemics; access to testing and treatment; diagnostic pressure; travel).**

We reworded this sentence to explicitly mention host population structures and host behavior (**Lines 73-74**): “... and host factors, such as population structures, and host behavior, including differences in access to diagnostics and treatment ...”

Material and Methods:

- 1. The genomic analysis is pretty standard and robust.**

Thank you.

- 2. Lines 810-812: What was the justification for the 80% LOF cutoff?**

We appreciate the opportunity to explain the basis for this cutoff. We used 80% as a conservative cut-off for LOF in the *mtr* operon previously (PMID 36735316). In that work, we found that the majority of *mtrR* alleles with premature stop codons produce peptides that are 23%-69% of the expected peptide length. In *mtrC*, the 2bp deletion leading to a premature stop codon produces a peptide that is 35% of the expected length. We added text and a citation justifying the 80% cut-off (**Lines 980-983**): “We chose the 80% value as a conservative cutoff based on findings that the majority of *mtrR* alleles with premature stop codons produce peptides that are 23%-69% of the expected peptide length (86). Likewise,

in *mtrC*, the 2bp deletion leading to a premature stop codon produces a peptide that is 35% of the expected length (86).”

3. Lines 841-842: What was the justification for including at least 30 tips? Would this exclude those emerging but epidemiologically important lineages or those with recombination-induced diversity in resistance loci?

The main reason for this threshold was to avoid the inclusion of outbreak clusters and singleton mutants that would adversely affect computational feasibility and potentially bias results.

We agree that emerging lineages may be excluded. However, most if not all emerging lineages will contain little signal to inform estimates of the fitness contributions from resistance determinants, especially when considering the sparsity of sampling.

We updated text in the “Discussion” section to address this point: **Lines 612-614** “... explicitly defining fixed lineages is an approximation and may result in fragmentation of otherwise linked lineages and exclusion of lineages smaller than the threshold size”

4. Lines 904-906: I wasn’t quite clear here – I think the authors are defining lineages as monophyletic groups with stable AMR determinants and assuming no within-lineage recombination. This may not be entirely valid for *N. gonorrhoeae*.

The high recombination rate of *Neisseria gonorrhoeae* poses a challenge for phylogenetic analysis. Over epidemic timescales, however, *Neisseria gonorrhoeae* displays a population structure characterised by strains and lineages and shaped by antimicrobial resistance and sexual networks. See, for example, PMID 27638945 and PMID 31358980. Defining lineages as monophyletic groups would indeed cause issues in light of high recombination rates, due to the presence of (1) recombinants near terminal branches and (2) nested lineage structure. Precisely due to these concerns, we defined lineages as paraphyletic groups – any lines of descent from the lineage’s most recent common ancestor that show evidence of gain or loss of resistance determinants (and thus recombination) are not included as a part of the lineage.

We added the following sentence to highlight the paraphyletic nature of the lineage definition. **Lines 1083-1084**: “Note that the lineages are typically paraphyletic and exclude isolates that show evidence of a change in resistance determinant type.”

5. Line 1119-1120: Do the authors know if introduction of the kanamycin cassette introduces a fitness cost?

To control for the possibility of a kanamycin-associated fitness cost in these experiments, all pairwise competitions were performed with swapped kanamycin markers: that is, for each pair of strains A and B, pairwise competitions were performed with A-kan vs B and with A vs B-kan to ensure that any fitness difference between strains was not attributable to effects from the kanamycin cassette. For example, see Figure 2 (panels B vs C) and Figure S5 (panels A vs B, C vs D). In each case, we found that the relative fitness of the strain pairs in each competition was consistent regardless of which strain carried the kanamycin cassette.

6. Did the authors check for introduction of secondary mutations during the in vitro and in vivo work?

We thank the reviewer for this question. We believe there is very low risk of secondary mutations impacting our results for several reasons.

Because the ciprofloxacin-resistant *gyrA* mutant strains constructed for this work were selected by ciprofloxacin exposure, we analyzed all *gyrA* strains in this manuscript by ONT genome sequencing.

- In the ciprofloxacin-susceptible *gyrA* strain **GCGS0481 *gyrA*91S/95D**, we identified an additional variant in *gyrA* at amino acid position 250 (M250I). This strain was constructed from the parental, ciprofloxacin-resistant GCGS0481 isolate by introducing the *gyrA* allele from the susceptible laboratory strain FA19. The FA19 *gyrA* allele is the source of the variants 91S and 95D, known to confer ciprofloxacin susceptibility, and also the source of the M250I variant. This variant has been reported in other *N. gonorrhoeae* genomes (e.g., PMID 30961546) and does not contribute to fluoroquinolone resistance. The ciprofloxacin-resistance conferring *gyrA* alleles for 91F/95G and 91F/95A natively encode 250M, and so the ciprofloxacin-resistant GCGS0481 *gyrA*91F/95G and *gyrA*91F/95A strains in the manuscript are not matched with the *gyrA*91S/95D at amino acid position 250. We therefore tested whether the M250I substitution impacted our relative fitness measurements. We created a version of the GCGS0481 *gyrA*91F/95G strain with the M250I substitution by transforming the 91S/95D/250I strain with the *gyrA* 91F/95G/250M allele, selecting on ciprofloxacin, and screening transformants to find one with a recombination breakpoint between the 91/95 locus and the polymorphic 250 site to yield a 91F/95G/250I genotype. The 91F/95G/250M and 91F/95G/250I strains had identical ciprofloxacin MICs and performed identically in pairwise competitions with the 91S/95D/250I strain and with the 91F/95A/250M strain, validating that the I250M substitution is neutral. The data and figures throughout the previous version of the manuscript showed results from the GCGS0481 *gyrA*91F/95G/250I strain that we constructed to validate the neutrality of the M250I substitution. These figures are reproduced below for reference. However, because the main text focuses on the comparison between *gyrA*91F/95G and *gyrA*91F/95A, we have revised the manuscript to present only data (previously not shown) from strains with the parental alleles: 91S/95D/250I, 91F/95G/250M, and 91F/95A/250M.

- Comparing the ciprofloxacin-resistant strains **GCGS0481 *gyrA*^{91F/95G}** and **GCGS0481 *gyrA*^{91F/95A}** presented in Figure 3, whole genome sequencing identified only a single variant outside the *gyrA* locus: a 6 bp-deletion in a repetitive region of a predicted phage-associated protein. We then performed genome sequencing of the GCGS0481 *gyrA*^{91F/95G/250I} strain described above and found that it did not carry this deletion. In addition to the expected

M250I substitution in *gyrA*, this strain acquired a phase-variable frameshift mutation in one of the *opa* genes via the described mechanism of varying the number of CTCTT repeats (in this case a one-repeat deletion). Because both the GCGS0481 *gyrA*91F/95G/250M strain shown in the current manuscript and the GCGS0481 *gyrA*91F/95G/250I strain from the previous version have identical phenotypes, and because neither of them shared their single off-target sequence variation with the other, we conclude that the known targeted variants in *gyrA* are responsible for the fitness difference compared to GCGS0481 *gyrA*91F/95A/250M.

- We uploaded the sequencing results to PRJNA1281689, as in the methods section (**Lines 1279-1284**): “Genomic DNA from each strain was purified using an Invitrogen PureLink Genomic DNA Mini Kit (K182001), prepared for sequencing using Oxford Nanopore Technologies Native Barcoding Kit 24 V14 (SQK-NBD114.24), and sequenced on a Oxford Nanopore Technologies R10.4.1 flowcell followed by basecalling with dorado v0.8.1 with super accuracy. Base called reads were uploaded to the NCBI SRA and are available at PRJNA1281689.”

For the *ponA* strains in this manuscript, finding the same fitness effects across multiple strain backgrounds further supported our results and substantially reduced the chance of impactful secondary mutations. The *ponA* mutations were introduced with a spectinomycin resistance cassette. The use of this orthogonal selection pressure reduces the risk of off-target mutations in *ponA* or other penicillin resistance-associated loci. The *in vitro* and *in vivo* fitness effects for the *ponA* alleles investigated here were consistent in two different strain backgrounds (Fig 5B vs Fig S9, Fig S10 A vs B), which would be extremely unlikely if the fitness difference in one pair of isogenic *ponA* strains was attributable to a secondary mutation.

Results:

- 1. Line 114: I suggest modifying this to also note that the effect of sexual networks (composition; behaviours within networks) has an impact on lineages structure.**

We modified the sentence to read “... by antimicrobials and sexual networks ...”

- 2. Lines 157-158: Do the authors have any epidemiological / demographic information on patients in lineage 21 to help explain its dynamics? Were there mutations elsewhere in the genome that may explain its emergence?**

Lineage 21 carries GyrA 91F/95A. Since we found that this allele is associated with a fitness advantage, we believe that this helps explain Lineage 21’s dynamics. We added a sentence highlighting this in the discussion (**lines 556-557**): “Likewise, the fitness advantage of Lineage 21 was associated with the carriage of GyrA 91F/95A (Figure S34).”

- 3. Lines 153-155: The authors make an implicit assumption that changes in treatment guidelines temporally align with changes in $N_e(t)$, which is probably OK, but in practice, changes to guidelines are often delayed or inconsistent across different settings.**

We agree that it can often take time for guideline changes to be realized. However, in this case, the guideline changes align closely with the prescribing patterns reported in CDC data, as shown in Figure S1.

4. Lines 188-203: *gyrA* – did the GyrA 91F/95A lineages that expanded after FQs were no longer recommended have any other mutations that could encourage co-selection?

Linkage between a compensatory or otherwise beneficial mutation and GyrA 91F/95A could explain the expansion of lineages carrying GyrA 91F/95A; however, we saw no obvious candidate mutation. An alternative explanation for the success of these lineages is that GyrA 91F/95A is selected via bystander exposure to fluoroquinolones, consistent with its greater *in vitro* fitness compared to 91F/95G. We plan to investigate the link between population level use of fluoroquinolones and the fitness impact of GyrA 91F/95A in future work.

5. Lines 229-230: Why did the authors only take the competition experiments out to 8 hours? Ideally it would be good to do longer term passage; this may also reveal compensatory mutations.

We limited our *in vitro* competition experiments to 8 hours because our *N. gonorrhoeae* cultures began to die after this time point, consistent with the known phenomenon of stationary phase autolysis in *N. gonorrhoeae*. Serial passaging of *N. gonorrhoeae* can be used to extend the length of competition assays, but we found that 8-hour experiments were sufficient to detect the relatively large-magnitude fitness effects of *gyrA* mutations that we report here.

Extending competition experiments beyond 8 hours to enhance detection of smaller fitness effects and to facilitate the identification of *de novo* compensatory mutations—particularly in the context of the mouse model, where such work has been done by our team (PMID 29615507)—is beyond the scope of this manuscript but is of great interest to us. We are pursuing this strategy for a follow-up study.

6. Lines 228-232: The differential fitness of 91F/95A and 91F/95G should be addressed; although the mutations confer similar high MICs, their fitness difference may indicate non-equivalence in biological cost or compensation. Again, it would be important to confirm there are no secondary mutations that have been introduced during the mutagenesis process which may contribute to observed phenotype.

We appreciate the reviewer's concerns. Please see our answer to the reviewer's comment above (#6 in the Materials and Methods section). We sequenced the isolates to address exactly this point, and, in short, we experimentally confirmed that the single additional amino acid change we identified had no impact on fitness or MIC.

7. Lines 371-372: I note the authors excluded *tetM* because of limited treatment data, but this omits an important contributor to the success of Lineage 2.

We agree that this omits an important contributor to Lineage 2's success; we also recognize that *tetM* is of increasing interest because of the rollout of doxycycline post-exposure prophylaxis. However, as we do not have access to tetracycline prescription data, we could not include it in our model. Nonetheless, despite not including tetracycline prescribing data, we identified *tetM*'s effect through the lineage's growth rate residual term, which we interpret as a success of our modelling approach.

We are very interested in expanding the analysis to include population treatment data to complement available primary treatment data, especially as doing so would enable us to quantify the impact of bystander exposure. We hope to pursue this in future studies.

8. Lines 366-367: The choice of a 10% threshold for identifying meaningful residual deviations should be justified or sensitivity-tested, as it shapes the number of lineages flagged as unexplained.

The purpose of the 10% threshold was to help select lineages for a closer manual inspection, rather than to mark lineages as unexplained. We chose a 10% excess growth or decline in a year, arguing that this threshold was as low as possible while still epidemiologically meaningful. To clarify our approach and this justification, we reworded this paragraph and elaborated on the choice of the threshold (**Lines 449-455**): “While the fitness contributions of the set of resistance determinants in our model accounted for much of the lineage dynamics, some dynamics remained unexplained. We first sought to highlight lineages for which a manual inspection would be beneficial. To do so, we computed the number of years in which the absolute value of the average of the sum of the residual and lineage background terms exceeded a threshold of 0.1 for each lineage. We chose a threshold of 0.1 as it represents approximately 10% growth or decline in a given year, arguing that this is large enough to be epidemiologically meaningful while maintaining sensitivity.”

Discussion - General points:

1. The Discussion underplays other ecological or epidemiological factors (e.g. sexual networks, host immunity, transmission dynamics, diagnostic practices) that may have shaped lineage success. In addition, use of antibiotics for other indications (which this team has previously explored) has not been mentioned – this could obviously impact indirectly on NG lineage structure.

We clarified “bystander exposure” to read (**line 571**) “bystander exposure to antibiotics not used directly to treat gonorrhoea”

2. Some statements imply causal links between antibiotic policies and lineage dynamics, but these are based on temporal correlations without direct causal tests. I think the tone could be slightly modified to frame treatment-lineage relationships as associations or correlations, unless there is direct mechanistic evidence otherwise.

We reworded the following statements to clarify the non-causal nature of these estimates.
Line 517 “The carriage of GyrA 91F/95G was associated with a decrease in lineage fitness after ...”

Lines 529-533 “Similarly, for mosaic *penA* 34, we saw clear evidence of a large growth rate advantage associated with the carriage of this determinant compared to the baseline type that does not carry any of the resistance determinants studied when cephalosporins other than ceftriaxone 250mg were recommended. We estimated that this associated advantage was rapidly lost after the switch to ceftriaxone 250mg, ...”

Line 535 “For PBP1 421P, we estimated a consistent small fitness defect associated with its carriage”

Line 547 “We estimated a large lineage growth rate benefit associated with the carriage of mosaic *mtrR*”

Line 552 “one of mosaic *mtrC* or *mtrD*, both of which are associated with their own distinct fitness”

Lines 562-567 “This revealed an example of apparent hitchhiking, where in Lineage 22 (Figure 7) the fitness cost associated with the carriage of one resistance determinant, GyrA 91F/95G, conferring resistance to fluoroquinolones, was outweighed by the fitness benefit associated with the carriage of another resistance determinant, mosaic *penA* 34, conferring resistance to cephalosporins. Likewise, the fitness advantage of Lineage 21 was associated with the carriage of GyrA 91F/95A (Figure S34).”

3. **My major comment on the Discussion is that it wasn't clear how the findings might inform AMR surveillance, treatment guideline development, or targeted interventions. A few sentences in the closing paragraphs illustrating the direct applicability of this work would greatly help readers fully understand it's utility and potential impact.**

We added the following to the discussion **Lines 655-661**: "In light of co-resistance, anticipating the impact of changing treatment guidelines in the near-term is both challenging and of great importance to public health. A change in guidelines can lead to unintended co-selection for resistance to unrelated antimicrobial classes or produce little effect on the prevalence of target resistance due to it being driven by co-resistance or absence of a fitness cost. Understanding how the combination of genetic elements carried by strains interacts with antimicrobial use can inform these decisions and avert unintended outcomes."

(Remarks on code availability):

1. **I read through the phylodynamic models in detail. While this is not my area of expertise, the rationale for use of the phylodynamic framework seemed methodologically sound.**

Thank you.

Reviewer #3

(Remarks to the Author):

The manuscript "Quantifying the impact of antibiotic use and genetic determinants of resistance on bacterial lineage dynamics" studies the contributions of resistance determinants to strain success in a large collection of *Neisseria gonorrhoeae* isolates. The manuscript is well written and is of great interest to the field, as it explores the fitness effects of different AMR-conferring mutations and validates some of the candidate mutations with experiments.

I was asked to assess the mathematical modelling and phylogenetics-related aspects of the study, so I will focus my review on those points.

Main remarks:

I find the approach of the authors – modelling the effective population size of the different AMR clades – clever and of interest for the field. However, I have a few concerns:

1. **My main concern is the fact that they first reconstruct a time-resolved phylogeny using Bactdating, which assumes a constant effective population size, and then model the changes in effective population size on this tree, for particular clades. To me, these two modelling assumptions contradict each other. Have the authors tested how these model choices impact their analysis? If the initial time-resolved phylogeny was reconstructed using a varying effective population size, would the results change? Using a skygrid/skyline model in the time-tree reconstruction step would then not contradict the modelling of the changes in effective population size in the second part of the analysis. I realise that skygrid/skyline models are not implemented in Bactdating at present, but this point should be at the very least discussed in depth as a limitation of the study.**

The reviewer raises an important point. In the absence of phylogenetic signal in the data, using BactDating would produce a date phylogeny with limited evidence of fluctuations in

effective population. However, in such a scenario there would be next to no signal to conduct any form of phylodynamic analysis. A two-step approach to phylodynamic analysis, where a dated phylogeny is reconstructed first and then used as a starting point for subsequent analysis, is fairly common when studying bacterial pathogens. See PMID 35989600 for an overview. While there may be a loss of statistical power due to such an approach, it is typically very mild (PMID 35989600), and due to the computational challenges of working with bacterial phylogenies, it is often the only scalable approach available.

It is worth pointing out that the assumptions of a skygrid/skyline approach are also not quite in line with the assumptions of the downstream analysis. A skygrid/skyline model would assume the same (time-inhomogeneous) effective population size across the tree; however, our analysis is applied to distinct subtrees. In an ideal setting, one would aim to jointly estimate the individual time trees and the effective population sizes, while pooling clock parameters across trees or estimate the baseline tree under a skyline model that can accommodate multiple subpopulations (e.g., under a version of the skyline model presented in PMID 34893904). Unfortunately, there aren't any scalable methods available for either of these approaches. Developing such an approach is of great interest to us, albeit not within the scope of this manuscript.

We have highlighted the two-step approach as a limitation of this study within the discussion. **Lines 600-609:** “Second, we have used a two-step approach for the phylodynamic analysis, whereby a dated phylogeny is reconstructed first, and then used as a fixed input for downstream phylodynamic analysis. This approach neglects the propagation of uncertainty from the phylogenetic reconstruction (and phylogenetic dating) through to the phylodynamic analysis while potentially reducing temporal signal due to over-regularization (or under-regularization) by the model used for reconstructing the dated phylogeny. This approach is commonly used in bacterial phylodynamics (19, 22, 23) due to the computational challenges present. Relaxing this approximation would require the development of a scalable approach to jointly infer the ancestral dates of multiple lineage trees along with the parameters of the phylodynamic model.”

- 2. The authors use a two-step approach to perform their analysis (tree reconstruction and then N_e modelling). I might have missed it, but did the authors consider the uncertainty in the reconstructed timed phylogeny from Bactdating? (i.e. the posterior of trees generated by Bactdating) In Bactdating, the tree topology is fixed, but the branch lengths can have wide credible intervals.**

The uncertainty in the time tree is only considered for the purposes of lineage assignment (when determining age cutoff; see **lines 1014-1019**) but not used during any subsequent analysis.

- 3. The authors base their study on a bacterial time-resolved phylogeny, yet they do not present the clock signal (root-to-tip regression) or the substitution rate estimated during the tree reconstruction. This is important to assess the reliability of the phylogenetic signal and compare the tree obtained to the existing literature on *N. gonorrhoeae*.**

We added the following text to the Lineage Assignment & Phylogenetic Reconstruction subsection (**lines 996-999**): “The estimated substitution rate was 5.46×10^{-6} substitutions per site per year with 95% credible interval of (5.29×10^{-6} , 5.65×10^{-6}) substitutions per genome per year, while the root date was estimated to be 1645 with 95% credible interval of (1621, 1667). These values are largely in agreement with prior estimates (27).”

- 4. The hierarchical phylodynamic model developed by the authors is highly parameterized, taking into account “multiple lineages, multiple pathways, and multiple time-varying covariates”. While it is highly valuable to be able to build such a model to test hypotheses, I wonder what a model comparison analysis, removing in turn some parameters (genetic markers/antibiotics), would yield. Are all the parameters critical to explain the N_e changes of the multiple lineages? How much does the model currently explain the fitness dynamics?**

The constant lineage background terms, along with the lineage specific $r(t)$ variations are necessary to ensure that the model is properly specified and has the capacity to capture overdispersion in the lineage trajectories. It is worth noting that without any covariates the model collapses into a hierarchical skygrid-type model with a shared “mean” trajectory following a second-order random walk. As such, the model used cannot do any worse at explaining lineage N_e changes than the skygrid prior.

While removing covariates is a valuable exercise in terms of forecasting and variable selection, the aim of our modelling was to estimate the effects of genetic markers that have been described as major drivers of resistance in *N. gonorrhoeae* on lineage growth rates, along with quantifying the associated uncertainty as opposed to forecasting. Removing covariates may increase statistical power, but doing so is also effectively guaranteed to introduce unobserved confounding due to failing to adjust for covariates that we know are very likely true covariates. We agree that variable selection to determine a minimal set of determinants and treatment data necessary for forecasting AMR fitness is of general interest. In future work, we aim to investigate the use of shrinkage priors (e.g., <https://doi.org/10.48550/arXiv.1707.01694>), or model reduction methods such as projection predictive inference (e.g., <https://doi.org/10.1214/20-EJS1711>) for this purpose. We further note that this task non-trivial. Shrinkage priors are notoriously difficult to sample from and are not straightforward to implement when considering the structure of the model. However, approaches such as projection predictive inference typically rely on factorizable likelihoods – a property that the coalescent likelihood fails to satisfy.

Other remarks:

- 1. The dataset comprises isolates from 2000 to 2019, but most figures show dynamics estimated before this – can the author discuss why they chose this? For clarity, it would be good to add a mark on the figures showing for which time period there is sequence data.**

Phylodynamic inference allows us to extrapolate beyond the sampling frame (as long as there is sufficient signal in the lineages-through-time function). We chose the year 1993 as the lower bound for our analysis because this marks when fluoroquinolones were first included on the list of recommended treatments for *N. gonorrhoeae*.

We have highlighted the span of the treatment data in introduction (**lines 85-86**) “Data on primary treatment in the US over this period, and going back to 1988 (figure S1), have been reported by the CDC ...”

We modified the final sentence of the introduction to highlight the span of the study and the significance of the year 1993 (**lines 92-112**) “We analyzed the period from 1993, the year fluoroquinolones were recommended as first-line therapy for *N. gonorrhoeae* in the US (figure S1), to 2019, the year in which the most recent sample in this dataset was collected.”

- 2. Figure 1: what are the colors indicating?**

The coloring was simply to increase the legibility of the plot. We updated the legend to clarify this. **Figure 1 legend:** “Colors are for legibility purposes only.”

- 3. Figure 2A: The legend indicates “The grey transparent tips correspond to isolates that have diverged from the ancestral determinant combination of the parental lineage.” It would be good to mention what is the ancestral determinant combination to increase the readability of the figure.**

We added the ancestral types to the figure legend. **Figure 2A legend** “Ancestral types for Lineage 20: GyrA 91F/95G, PBP1 421P, mtr promoter A-del, 23s rRNA 2611C. Ancestral types for Lineage 21: GyrA 91F/95A, PBP1 421P, mtr promoter: non-mosaic, 23s rRNA 2611C. Ancestral types for Lineage 22: GyrA 91F/95G, PBP1 421P, penA 34 mosaic, mtr promoter non-mosaic, 23s rRNA 2611C.”

- 4. Line 817: “4; Less than 12% of positions were missing in pseudogenomes”, how did the authors come up with this threshold?**

This threshold was chosen on based on prior analysis (see PMID 37138444) of the distribution of missing sites among *N. gonorrhoeae* isolates with high quality genomic data to account for differences among the isolates in the accessory genome and repeat sequences. The reference genome used in our pipeline encodes the gonococcal genetic island (GGI), which is not present in ~50% of gonococcal isolates, and the gonococcus genome encodes several repetitive loci (e.g., *opa* and *pil* genes).

(Remarks on code availability):

I successfully ran the code provided on the GitHub repository. I did not go through all the codes, but only run_model.R file, and the stan model briefly. I was able to reproduce the main figures. It is nicely organised and easy to run. However, the code is sparsely commented, which can make it hard to use for future applications on other datasets. It would be of particular interest to have a description of the data format used in the model. Additionally, it would be good to have a clear list of packages and versions that are needed to run the code.

We updated the README file within the github repository to list all the required dependencies and where they can be installed from.

Reviewer #4

(Remarks to the Author):

This is a wonderful, though dense, piece of work. It is exactly what is needed for the field of AMR and microbial population dynamics. The authors have done an incredible amount of work to integrate laboratory phenotypes with public health scale population dynamic modelling, leveraging genomic surveillance data to guide exquisitely targeted confirmatory experimentation for better understanding AMR and the influence of genetic context (in a WHO AMR priority pathogen). Moreover, they have made an admirably succinct narration for what is a very complex, interdisciplinary study. I have several probing questions below which mostly reflect my enthusiasm for the manuscript and subject area, rather than major criticisms that detract from the overall quality of the work. However, I do feel some require further explanation (and potentially further analyses) could improve the manuscript.

1. Line 51 – “neutral or deleterious” perhaps – deleterious assumes there is still a fitness cost associated with AMR carriage (which is often ameliorated under sustained selection)

We made reworded the sentence change: (Line 55-56): “As such alleles and genes that were previously fitness conferring may become neutral or deleterious.”

2. Line 78 – “demographics of the infected individuals” Any info on ab use by those individuals?

Unfortunately, the data on individual antibiotic use were not available.

3. Figure 1. I feel like this could be more informative for evaluating the study - perhaps truncating to 1980 to allow better visualisation of the relevant time period and room to show uncertainty (in both topology and ASR node support) in the tree structure and maybe some raw AMR data (not sure why this was separated into Table 1 and hard to assess how well the ASR has worked for lineage assignment as is). It’s hard to get a feel for co-existence/circulation of the lineages in its current form (since presumably real world competition dynamics are what we are trying to capture/test in the lab here).

Thank you for this suggestion. We truncated the tree in Figure 1 at 1950 to increase legibility while retaining most of the overall structure during the bulk of the antibiotic era.

Our main intention for this figure is to give the reader an idea of the overall population structure of *Neisseria gonorrhoeae* in the US. We use 27 unique AMR determinants to define lineages, making it infeasible to visualise the diversity of AMR determinants in single figure. Consequently, we split the lineage determinant types into Table 1 and provided a snapshot of determinant diversity in Figure 2. However, we agree with the reviewer’s suggestion that phenotypic data can be used as a form of validation for the lineage assignment step. As such, we modified figure 1 to include the distribution of MIC values for ciprofloxacin, cefixime, and azithromycin (see above).

We updated the figure and **Figure 1 legend** text to reflect this: “Panel A: The phylogenetic tree annotated according to the lineage assignment of each node. Gray nodes in the phylogenetic tree denote lines of descent that are not in an assigned lineage. Lineage numbering was determined by post-order traversal of the tree. Colors are for legibility purposes only. Panel B: The distribution of phenotypic resistance for ciprofloxacin, cefixime, and azithromycin. Gray represents missing values. MICs are reported in units of $\mu\text{g}/\text{mL}$. MICs are truncated for ciprofloxacin above $32\mu\text{g}/\text{mL}$ and below $0.005\mu\text{g}/\text{mL}$; for cefixime above $1\mu\text{g}/\text{mL}$ and below $0.003\mu\text{g}/\text{mL}$, and for azithromycin MICs above $16\mu\text{g}/\text{mL}$ and below $0.06\mu\text{g}/\text{mL}$.”

We made the following changes to the main text referencing Figure 1B (Lines 175-179): “While we did not use phenotypic resistance data in the lineage assignment or subsequent analysis, we noted that the resulting lineage assignment (Figure 1A) was congruent with the distribution of phenotypic resistance to ciprofloxacin, cefixime, and azithromycin (Figure 1B). This provided us with a form of validation for the lineage assignment approach.”

4. Line 131 – are the phenotypes the same?

For both lineages 21 and 23, the median cefixime MIC was $0.25\ \mu\text{g}/\text{mL}$

5. e.g. in Table S3 – resistances are presented as ≥ 32 – is there a higher MIC for one set of mutations, was this information available? Or were absolute MIC laboratory phenotypes at least determined and compared for the competing strains? [and same question for other antimicrobial class phenotypes and allelic variants]

Ciprofloxacin MICs above $32\ \mu\text{g}/\text{mL}$ were not evaluated, as there is no evidence that MICs above this threshold are associated with significantly different clinical outcomes upon fluoroquinolone treatment. While some *gyrA* alleles may confer MICs above this threshold, it is unlikely that these differences contribute to lineage dynamics, given the lack of evidence that higher MICs are meaningfully selected for under typical fluoroquinolone dosage regimens.

While the PonA L421P contributes to multilocus chromosomal penicillin resistance, this mutation in isolation has a negligible effect on penicillin MIC (see PMID 11850260 and PMID 40631333).

6. Table 1- maybe add the corresponding antimicrobial classes for the uninitiated?

We updated Table 1 legend to list the associations between the loci listed and relevant antimicrobials. **Table 1 legend:** “Genes and loci associated with fluoroquinolone resistance: *gyrA*, *parC*, *mtr*. Genes and loci associated with cephalosporin resistance: *ponA*, *penA*, *mtr*. Genes and loci associated with macrolide resistance: *mtr*, 23S rRNA.”

7. Figure 2 is compelling, but what does this look like across the tree? If they are partitioned into resistant and non-resistant (or determinant-containing vs not) irrespective of lineage? Do the population dynamics shown in this select set of sub-lineages hold?

Figure 2 is effectively a form of exploratory analysis. We developed the hierarchical phylogenetic precisely to formalize and extend this type of analysis to the entire tree. We made the following change to the manuscript to highlight this (Lines 208-210): “We next sought to quantify the fitness contributions of the genetic determinants of resistance, how

these varied over time, and whether this was consistent across the phylogeny, in effect extending and quantifying the analysis presented in (Figure 2).”

8. It’s unclear why further detail (rightmost bars) are only provided for Lineage 20?

Lineage 20 is unique in that after 2007 nearly all descendant isolates changed type in at least one of the lineage-defining determinants and thus are no longer considered a part of the lineage.

9. Figure 3 - The modelling results are borderline on their own with wide uncertainty and little convincing variation from the baseline, but these estimates have natural limitations borne from the actual evolutionary dynamics and polyphyly of phenotypes in the bacterial populations. The results are rescued convincingly by the juxtaposed laboratory data in panel B which confirms the relative fitness benefits of these alleles in vitro.

We appreciate the reviewer’s point. We note that the large uncertainty present in some of the estimates in the **Figure 3A -“Mean”** ribbon may reflect genuine heterogeneity of effects across different *parC* contexts. The uncertainty for estimates on the *parC* 86D/87R/91E background is much reduced. We added the following sentence to highlight this (**Lines 251-254**): “It is also worth noting that we were unable to make meaningful inferences about the impact of GyrA 91F/95A and GyrA 91F/95G across all *parC* contexts (Figure 3A; “Mean”). This may be due either to heterogeneity of fitness effects across *parC* contexts or to the lack of sufficient polyphyly to inform this estimate.”

10. Figure 4 – This decrease in fitness appears to be mostly driven by the descent of lineage 22 (which is most of the mosaic allele containing isolates according to Table 1 and Figure 1 – e.g. Lineage 23 appears relatively small) Is there anything to be learned from the Lineage 23 dynamics that would support this? And the descent of Lineage 22 is really a clonal replacement (of 22 by 21) at the time of treatment recommendation change. So, is the ‘mysterious’ expansion of Lineage 21 now meant to be explained by the decreased fitness of the mosaic *penA* allele? Again, these are limitations of the natural dynamics, but it needs exploring – why no head-to-head laboratory competition for these alleles?

Lineage 23 is indeed relatively small, and we note that lineage 23 has no sampled descendants after 2012. This observation is in line with the hypothesis, suggested by our modelling, that the mosaic *penA* 34 allele does not provide any fitness benefit after the guideline switch to ceftriaxone 250mg.

Based on our modelling, the expansion of lineage 21 is best explained by the presence of GyrA 91F/95A, which is predicted to be associated with a fitness advantage. We added the following sentence highlighting it to the discussion (**Lines 566-567**): “Likewise, the fitness advantage of Lineage 21 was associated with the carriage of GyrA 91F/95A (Figure S34).”

11. Figure 5 – This is a nice negative control for confirming the observed changes are the influence of fluctuating treatment guidance, but it’s not explicit why this progressed to mouse experiments whereas the fluoroquinolone determinants did not. Is this because of the negative in vitro growth results? And why wasn’t it observed in vitro? Just a time effect? Could it be influenced by different starting ratios? It would have been nice to see the contrast with the *gyrA/parC* work discussed more e.g. with Line 433 – 442.

While we saw fitness effects of *gyrA* mutations in *in vitro* competition assays that were in line with our expectations based on phylodynamic work, the *ponA* mutations we examined here had no fitness effect *in vitro* despite the phylodynamic prediction that the PBP1 L421P mutation is likely deleterious. Likewise, the fitness estimate obtained through phylodynamic modelling was also small. We therefore hypothesized that the *ponA* mutation may cause a conditional fitness cost that is not replicated in the highly artificial growth environment of rich media and moved to test the fitness of these strains *in vivo* in an attempt to investigate the fitness of these mutations in a more physiologically relevant environment. Verifying the fitness of *gyrA* mutants in mice is beyond the scope of this project, especially because we already see concordant results for these genotypes in *in vitro* competition assays and in the phylodynamic analysis of natural *N. gonorrhoeae* populations. We updated the corresponding results section to reflect this (**Lines 357-360**): “Because the predicted fitness effect of the PBP1 421P variant is small and may not be detectable on the timeline of the *in vitro* studies, we also performed competition assays of these strains in a murine infection model to measure relative fitness in the presence of physiologically relevant stressors and over a longer period.”

- 12. Figure 6 – given the need to collapse AZM genetic determinants, I wonder whether the model have more signal if different phenotype bins (e.g. bounds of MIC value) were compared here. I’m unsure of the underlying MIC data distribution, and of course the relative influence of genotype and phenotype is not the main focus of the paper, but it would be interesting. Also perhaps specify ‘co-treatment’ with what for the AZM in legend**

We agree that utilizing phenotypic data may lead to increased power to detect fitness changes, especially the beneficial effect of resistance in the presence of antimicrobial use and of great interest. However, incorporating phenotypic data is outside of the scope of this study due to the inherent modelling challenges associated with such an approach. We also note that the AZM determinants aren’t collapsed in the sense of them being combined into one category in the regression model, rather the posterior is being evaluated to obtain the predicted effect of their sum. We added the following sentence to the legend of figure 6 explaining AZM co-treatment (**Figure 6 legend**): “Azithromycin co-treatment refers to the guidance recommending the co-administration of azithromycin and any recommended cephalosporin.”

- 13. Figure 7 – again Abx class (both for years of rec and corresponding genes) could be added to the Table for clarity/readability. It’s not intuitively clear from the text what AMR total is – is this having phenotypic resistance against the recommended treatment? Does the tailing off of a fitness benefit suggest there are other resistance alleles emerging? Or is this the influence of tetracycline use and non-target abxs?**

The AMR total category refers to the fitness effect of all the lineage determinants carried by a given lineage. We amended the legend of **figure 7** and supplemental **figures S14-S41** to better reflect this: “The bottom panel depicts a table summarizing the median total growth rate effect across 4 treatment periods, as well as the 95% credible interval around the median in brackets. AMR Total denotes the combined contribution across all AMR determinants studied carried by the lineage. Total denotes the combined growth rate effect of all AMR determinants carried as well as the residual and lineage background terms.”

Our modelling suggests that the tailing off of the fitness benefit is likely caused by the loss of fitness advantage of the mosaic *penA* 34 allele on the shift to ceftriaxone 250mg + azithromycin as the only recommended treatment. We do not estimate a fitness benefit associated with the mosaic *penA* 34 allele in that landscape. We have amended the following paragraph to clarify this. **Lines 426-429** “... however, the loss of fitness benefit of

the mosaic *penA* 34 after the shift to ceftriaxone 250mg plus azithromycin as the only recommended treatment in 2012 led to the accumulation of fitness costs from other resistance determinants and an overall negative growth rate.”

14. L393 – 416 – what about the possibility of different (non-binary) resistance phenotypes?

We did not see any discernible difference between the ciprofloxacin MICs conferred by the 91F/95A and 91F/95G alleles. See (lines 259-263).

15. L428 – again, this is potentially driven by clonal replacement perhaps consider/discuss differences in cefixime/ceftriaxone phenotypes for the different alleles here

We appreciate that the reviewer is positing an alternate hypothesis, but it is not clear to us what precisely this hypothesis is. We respectfully ask the reviewer to elaborate. We also note that the observed loss of fitness of *penA*34 after cefixime was discontinued as a primary treatment for gonorrhea is congruent with estimates based on incidence data from England (see PMID 29088226, cited also in main text).

16. L433- L442 – again, as above, some discussion on why this effect wasn’t detectable in vitro would add value

We added a note on this to the discussion. Lines 542-543 “however, we did not see such an effect *in vitro*, potentially due to the absence of physiological stressors.”

17. L466 – ‘incorporating population-wide antimicrobial use would ...’ (I assume individual level use data was not associated with this surveillance program?)

Correct, there are no data available on individual-level use.

18. L477 – ‘reducing the uncertainty’ ... the inherent bacterial population dynamics may also be a natural limiter here (see above) e.g. lack of polyphyly for some elements, etc

We agree that there are inherent limits imposed by bacterial dynamics and have amended the corresponding sentence to reflect this (Lines 595-596): “... although there are inherent limits to reduction in uncertainty due to frequent lack of polyphyly in bacteria”

19. L494 – re: ignoring spatial separation – for at least one analysis transmission network (i.e. MSM or MSW) was considered and could act as a proxy for spatial separation throughout – why was this not more systematically considered across lineages? Also, is there much/anything known about the spatial structure of gonno in the US that would support ignoring this?

Resolving spatial heterogeneities requires more detailed data than what was available. Unfortunately, we have neither the access to treatment data stratified by location nor the sampling resolution to meaningfully consider spatial dynamics. We amended the corresponding paragraph to elaborate on this (Lines 623-628): “Seventh, we ignored spatial heterogeneity in transmission and treatment. Relaxing this assumption would require access to spatially resolved treatment data as well as denser sampling. As the data collection is overall sparse with just over 4400 sequences retained for phylodynamic analysis, compared to over 640,000 reported cases of gonorrhea in 2022 (85), heterogeneity within the US in transmission and treatment is unlikely to impact the results.”

(Remarks on code availability):

I have had a look though and it appears comprehensive, is well commented, and having read through the methods, much of it is drawn from previous manuscripts/analyses so I'm confident or reproducibility here.

We thank the editor and reviewers for their continued constructive feedback and for the opportunity to respond to the comments here. Please see our point-by-point responses as below.

Reviewer #1

(Remarks to the Author):

In general the authors have thoroughly and thoughtfully addressed the reviewer comments. The new Snakemake pipeline on github is well documented and will be extremely helpful for reproducibility. My only remaining minor comment, is that although the authors now specify where to retrieve packages and dependencies, they still do not specify the version used for each. As both Reviewer 1 and 3 requested this, I think this should be addressed as it is important for reproducibility.

We thank the reviewer for this point, and we have now updated the GitHub documentation to include the versions.

(Remarks on code availability):

The code is well documented and runs well. I was able to reproduce the figures.

Great—thank you.

Reviewer #3

(Remarks to the Author):

Thank you for your careful consideration of the reviewers' remarks, including mine. The changes made to the manuscript add precision to both the results and the discussion. It remains a great, well-written study and is of large interest to the field.

We thank the reviewer for the kind words.

Last two remarks:

1. I agree with the authors that two-step approaches, while not ideal, are currently necessary when considering large datasets such as theirs. Thank you for adding this point in the discussion. However, given that the authors' results focus on the timings of fitness changes with respect to antibiotic usage/implementation, it would be good to have special care with the use of the "one" timed tree. Some examples from the text: "Lineage 21 expanded after the 2010 switch" Line 172, "with clear support for a growth rate benefit only in 2010-2011" : Line 348 It is unclear how much the results from a single timed tree reflects what would be obtained from the full BactDating posterior (which already does not take into account topology uncertainty). I understand that using the full posterior of trees is tricky; however, the authors could replicate their analysis on a few trees to analyse how consistent their results are. A difference of a few years in some node dating could impact their conclusions. I would also encourage the authors to plot a phylogeny with confidence intervals on node dates in the supplementary material.

We thank the reviewer for these points. We have added a supplemental figure (Figure S3) that includes credible intervals for the origin nodes for each lineage. We now cite this figure when referencing the credible intervals for time estimates in **Line 172**.

[Lines 173-176]: Lineage 21 expanded around the 2010 switch in recommended treatment to dual therapy with azithromycin plus ceftriaxone median lineage MRCA time: 2011.46, 95% credible interval: [2009.66, 2012.66]; see Figure S3 for credible intervals indicating dating uncertainty for lineage ancestor nodes.

Figure S3: Lineage assignment based on AMR determinants with dating uncertainty. The phylogenetic tree annotated according to the lineage assignment of each node. Orange bars indicate 95% credible intervals for the dating of lineage ancestors. Gray nodes in the phylogenetic tree denote lines of descent that are not in an assigned lineage. Lineage numbering was determined by post-order traversal of the tree. Colors are for legibility purposes only.

To the reviewer's concern that a difference of a few years in the node dating could impact the conclusions, we note that our results focus on phylodynamic estimates of fluctuating growth rates and regression of these rates against patterns of antibiotic use. As such, our results depend on the overall distribution of ancestral lines through time, rather than on precise timing of single nodes.

We also note that use of a single fixed tree is a limitation of the available methods. While we agree with the reviewer that it would be ideal to simultaneously consider tree inference and phylodynamic model estimation, current methods cannot scale to a dataset as large and a model as complex as used here. We follow the standard for the field in using this simplification in the context of complex models; see, for example [PMID: 24743590], [PMID: 39527637], [PMID: 28204593], [PMID: 37273682]. This simplification is especially common in the context of studies of bacterial pathogens [PMID: 35989600], as phylogenetic reconstruction for bacterial pathogens is often much more challenging than for viral pathogens.

The reviewer suggests using a few samples from the time tree posterior as a workaround. However, we are unaware of a principled way to select a few trees and derive from them interpretable inferences. A meaningful sensitivity analysis would require on the order of hundreds of samples, which would then have to be integrated as independent observations into the hierarchical phylodynamic model. This would make the model prohibitively computationally expensive, as the memory cost and computational cost would scale at least linearly with each additional sample.

We agree with the reviewer that methods capable of integrating phylogenetic inference with phylodynamic estimation and efficiently scaling to large datasets and complex models would be of great interest. In the absence of such methods, we have followed the standards of the field.

- 2. Clock signal: Thank you for providing the estimates of the substitution rate and root time. I note that the reference added [PMID 31358980] estimated 3.74E-06 [3.39E-06 – 4.07E-06] substitutions/site/year, which is ~20% lower than the one they estimated. Such a difference could be due to the fact that the datasets are different. As in the article the authors cited, it would be good to present a root-to-tip regression to show how much signal there is in the authors' dataset (see Fig S2 in 31358980). Again, as there is uncertainty in the estimated rate, it would be sensible to analyse/provide a sense of the impact of these on the results (see point above).**

We appreciate the reviewer's concern about the substitution rate. The choice of a clock model is one of the most important decisions of an analysis that relies on time-scaled phylogenies. Since clock rate heterogeneity is extremely common in bacteria [PMID: 25887947] and relaxed clocks can capture this heterogeneity, we chose a relaxed clock model as implemented in BactDating. In contrast, strict clock models assume a linear relationship between the number of substitutions and evolutionary distances. The strict clock assumption is also common to approximate tree timing methods, such as root-to-tip regression and least squares dating (LSD).

The substitution rate estimate reported in the main text of the paper cited by the reviewer [PMID 31358980] was obtained using LSD [PMID 26424727]. However, estimates of substitution rates based on a strict-clock assumption (as in LSD) display a

strong downwards bias in the presence of clock-rate heterogeneity [PMID: 2741209]. Further, LSD underestimates substitutions rates in scenarios that warrant the use of a relaxed clock [PMID: 2741209]. This bias has been independently observed in several other papers, e.g., Figures S2 and S4 in [PMID: 30184106] and Figure S3 in [<https://doi.org/10.1093/ve/vex025>].

To test whether this explains the difference between our reported substitution rate and the one cited by the reviewer, we ran LSD on the same ML tree that was used as an input to BactDating in our study. This resulted in a substitution rate estimate of **3.56e-06** [**3.44e-06 – 3.63e-06**]. This estimate is indistinguishable from that of **3.74e-06** [**3.39e-06 – 4.07e-06**] obtained in [PMID 31358980]. This demonstrates that the discrepancy noted by the reviewer can be entirely attributed to the use of a relaxed clock in our study as opposed to the strict clock assumed in [PMID 31358980].

Reviewer #4

(Remarks to the Author):

Thank you to the authors for providing comprehensive responses to my queries.

We thank the reviewer for their contributions to improving the manuscript.

Additional notes

We have corrected a several typos in the methods section. We have also ensured that the notation for probability distributions remains consistent throughout the methods section, namely on **lines 1156, 1158** where a variance parametrization was used as opposed to the standard deviation parametrization used throughout the remained of the text. This has now been changed to a standard deviation parametrization.

We have also clarified that the parametrization for the distribution on **line 1174** is analogous: **Lines 1175-1178** “Where *MVN* stands for a multi-variate normal distribution parametrized using a mean and a square root of the covariance matrix, **1** represents the matrix with unit entries, *I* is the identity matrix and $U([0,1])$ is the uniform distribution on the unit interval. ϕ is a pooling factor that determines the level of correlation between the individual antimicrobial specific coefficients.”

Response to reviewers

We thank the editor and the reviewer for their feedback on the revision and the opportunity to submit an updated version of the manuscript, which we believe improved in clarity and messaging.

Reviewer #3:

Remarks to the Author:

I thank the authors for carefully addressing my comments and for the detailed clarifications provided.

Although I agree that the use of a single fixed tree reflects a limitation of currently available methods, this should not preclude at least some exploration of tree uncertainty. I appreciate the inclusion of the new Figure S3 in the supplementary material and find the authors' reasoning and explanations satisfactory.

While there remains scope for further methodological improvement in future work, I believe the current version appropriately acknowledges the relevant limitations. I have no further comments.

We thank the reviewer for their continued engagement with the manuscript and feedback and appreciate the reviewer's finding that our reasoning and explanations satisfactorily address the reviewer's concerns. We agree that the field needs new and improved methods to address the impact of tree uncertainty, a challenging and interesting problem.